# PCA Initialization for Approximate Message Passing in Rotationally Invariant Models

**Marco Mondelli**
IST Austria
marco.mondelli@ist.ac.at

**Ramji Venkataramanan**
University of Cambridge
rv285@cam.ac.uk

## Abstract

We study the problem of estimating a rank-1 signal in the presence of rotationally invariant noise—a class of perturbations more general than Gaussian noise. Principal Component Analysis (PCA) provides a natural estimator, and sharp results on its performance have been obtained in the high-dimensional regime. Recently, an Approximate Message Passing (AMP) algorithm has been proposed as an alternative estimator with the potential to improve the accuracy of PCA. However, the existing analysis of AMP requires an initialization that is both correlated with the signal and independent of the noise, which is often unrealistic in practice. In this work, we combine the two methods, and propose to initialize AMP with PCA. Our main result is a rigorous asymptotic characterization of the performance of this estimator. Both the AMP algorithm and its analysis differ from those previously derived in the Gaussian setting: at every iteration, our AMP algorithm requires a specific term to account for PCA initialization, while in the Gaussian case, PCA initialization affects only the first iteration of AMP. The proof is based on a two-phase artificial AMP that first approximates the PCA estimator and then mimics the true AMP. Our numerical simulations show an excellent agreement between AMP results and theoretical predictions, and suggest an interesting open direction on achieving Bayes-optimal performance.

## 1 Introduction

We consider the problem of estimating a rank-1 signal from a noisy data matrix. In the square symmetric case, the data matrix is modeled as

$$X = \frac{\alpha}{n} u^* u^{*\mathsf{T}} + W \in \mathbb{R}^{n \times n}, \tag{1.1}$$

where $u^* \in \mathbb{R}^n$ is the unknown rank-1 signal, $W \in \mathbb{R}^{n \times n}$ is a symmetric noise matrix, and $\alpha > 0$ captures the signal-to-noise ratio (SNR). In the rectangular case, we observe the data matrix

$$X = \frac{\alpha}{m} u^* v^{*\mathsf{T}} + W \in \mathbb{R}^{m \times n}, \tag{1.2}$$

where $u^* \in \mathbb{R}^m$ and $v^* \in \mathbb{R}^n$ are the unknown signals, and $W \in \mathbb{R}^{m \times n}$ is a rectangular noise matrix. A natural estimator of the signal in the symmetric case is the principal eigenvector of $X$ (singular vectors, in the rectangular case). The performance of this principal component analysis (PCA) estimator and, more generally, the behavior of eigenvalues and eigenvectors of models like (1.1)-(1.2) has been widely studied in statistics [27, 49] and random matrix theory [2, 3, 10, 11, 15, 22, 30].

If $u^*, v^*$ are unstructured (e.g., they are uniformly distributed on a sphere), then it is not generally possible to improve on the PCA estimator. However, in a broad range of applications, the unknown signals have some underlying structure, e.g., they may be sparse, their entries may belong to a certain

35th Conference on Neural Information Processing Systems (NeurIPS 2021).

set, or they may be modelled using a prior distribution. Examples of structured matrix estimation problems include sparse PCA [17, 28, 61], non-negative PCA [32, 43], community detection under the stochastic block model [1, 16, 45], and group synchronization [50]. Since PCA is ill-equipped to capture the structure of the signal, we aim to improve on it using a family of iterative algorithms known as approximate message passing (AMP). AMP algorithms have two particularly attractive features: *(i)* they can be tailored to take advantage of prior information on the structure of the signal; and *(ii)* under suitable model assumptions, their performance in the high-dimensional limit is precisely characterized by a succinct deterministic recursion called *state evolution* [7, 13, 26]. AMP algorithms have been applied to a wide range of inference problems: estimation in linear models [8, 7, 19, 31, 39], generalized linear models [5, 37, 38, 42, 51, 53, 55], and low-rank matrix estimation with Gaussian noise [6, 17, 23, 29, 34, 44]. The survey [21] provides a unified description of AMP for these applications. Using the state evolution analysis, it has been proved that AMP achieves Bayes-optimal performance in some Gaussian models [17, 18, 44], and a bold conjecture from statistical physics posits that AMP is optimal among polynomial-time algorithms.

We study rank-1 matrix estimation in the setting where the noise matrix $W$ is rotationally invariant. This is a much milder assumption than $W$ being Gaussian: it only imposes that the orthogonal matrices in the spectral decomposition of $W$ are uniformly random, and allows for arbitrary eigenvalues/singular values. Hence, $W$ can capture a more complex correlation structure, which is typical in applications. For the models (1.1)-(1.2) with rotationally invariant noise, AMP algorithms were derived in [14, 48] and generalized in [20]. In particular, the AMP algorithm of [20] for the problem (1.1) produces estimates $u^t \in \mathbb{R}^n$ as follows:

$$u^t = \mathsf{u}_t(f^{t-1}), \quad f^t = Xu^t - \sum_{i=1}^{t} \mathsf{b}_{t,i}u^i, \qquad t \geq 2. \tag{1.3}$$

The iteration is initialized with a pilot estimate $u^1$. We can interpret (1.3) as a generalized power method. Recall that the power method approximates the principal eigenvector of $X$ using the iterative updates $\bar{u}^t = X\bar{u}^{t-1}/\|X\bar{u}^{t-1}\|$. For each $t$, the function $\mathsf{u}_t$ can be chosen to exploit any structural information known about the signal (e.g., sparsity). The "memory" coefficients $\{\mathsf{b}_{t,1}, \ldots, \mathsf{b}_{t,t}\}$ have a specific form to ensure that the iterates $(f^t, u^{t+1})$ have desirable statistical properties captured by state evolution. A rigorous state evolution result for the iteration (1.3) is established in [20], but the algorithm and its analysis require an initialization $u^1$ that is correlated with the unknown signal and independent of the noise $W$. In practice, one typically does not have access to such an initialization.

**Main contribution.** In this paper, we propose an AMP algorithm initialized via the PCA estimator, namely, the principal eigenvector of $X$ for the square case (1.1) and the left singular vector of $X$ for the rectangular case (1.2). Our main technical contribution is a state evolution result for this AMP algorithm, which gives a rigorous characterization of its performance in the high-dimensional limit. The challenge is that, as the PCA initialization depends on the noise matrix $W$, one cannot apply the state evolution machinery of [20]. To circumvent this issue, our key idea is to construct and analyze a *two-phase artificial AMP* algorithm. In the first phase, the artificial AMP performs a power method approaching the PCA estimator; and in the second phase, it mimics the behavior of the true AMP. We remark that the artificial AMP only serves as a proof technique. Thus, we can initialize it with a vector correlated with the signal $u^*$ and independent of the noise matrix $W$, which allows us to analyze it using the existing state evolution result.

Our analysis is tight in the sense that our AMP algorithm can be initialized with PCA whenever the PCA estimate has strictly positive correlation with the signal. This requires showing that, when PCA is effective, the state evolution of the first phase of the artificial AMP has a unique fixed point. To obtain such a result, we exploit free probability tools developed in [10, 11]. The agreement between the practical performance of AMP and the theoretical predictions of state evolution is demonstrated via numerical results for different spectral distributions of $W$. Our simulations also show that the performance of AMP—as well as its ability to improve upon the PCA initialization—crucially depends on the choice of the denoising functions $\mathsf{u}_t$ in the algorithm. Thus, the design of a Bayes-optimal AMP remains an exciting avenue for future research.

**Related work.** The asymptotic Bayes-optimal error for low-rank matrix estimation has been precisely characterized for Gaussian noise [4, 33], but remains an open problem for rotationally invariant noise. An AMP algorithm with PCA initialization was proposed in [44] for the Gaussian setting, and it was shown to be Bayes-optimal for some signal priors. A recent paper by Zhong et

al. [60] shows how AMP with PCA initialization can be used for estimating the top-$k$ principal components in applications such as high-dimensional genomics datasets. The authors use an empirical Bayes method to determine a joint prior distribution for the $k$ principal components, and assuming a Gaussian noise model, employ an AMP algorithm tailored to the prior to improve the PCA estimates of the principal components.

Both our AMP algorithm and proof technique differ significantly from those for Gaussian noise. When $\boldsymbol{W}$ is Gaussian, the PCA initialization affects only the first iteration of AMP. In contrast, for more general noise distributions, the AMP algorithm and its associated state evolution require a correction term at every iteration to account for the PCA initialization. This is due to the fact that, while AMP has a single memory term in the Gaussian case, more general noise distributions lead to a more involved memory structure, as in (1.3). As regards the proof technique, the argument of [44] consists of decoupling the PCA estimate from the bulk of the spectrum of $\boldsymbol{X}$. In contrast, our approach is based on a two-phase artificial AMP algorithm. This technique has proved successful in the context of generalized linear models [41, 42], albeit for Gaussian measurements. Other extensions of AMP beyond the Gaussian setting include Orthogonal AMP [36, 56], Vector AMP [24, 25, 52, 54], convolutional AMP [57] and Memory AMP [35]. These algorithms have been derived specifically for linear or generalized linear models, and extending them (with a practical initialization method) to low-rank matrix estimation is an interesting research direction.

Finally, we mention the recent independent work of Zhong et al. [59], which appeared after the original submission of our paper. This work generalizes AMP with PCA initialization to the problem of estimating rank-$k$ matrices in rotationally invariant noise, for $k \geq 1$. We remark that, in order to prove a state evolution result for AMP initialized with PCA, in [59] it is assumed that the signal strength is sufficiently large. In contrast, our result holds for any signal strength such that the PCA method is effective, but we require the free cumulants of the noise matrix to be non-negative. We also note that, when the signal strength is large, the assumption on the free cumulants can be automatically satisfied (see the footnote on page 6).

## 2 Preliminaries

**Notation and definitions.** Given $a \in \mathbb{R}$, we define $(a)_+ = \max(a, 0)$. Given two integers $i \leq j$, we define $[i, j] = \{i, \ldots, j\}$. If $i > j$, then $[i, j]$ denotes the empty set; products over the empty set are taken to be 1. Given a vector $\boldsymbol{x} \in \mathbb{R}^n$, we denote by $\|\boldsymbol{x}\|$ its Euclidean norm and by $\langle \boldsymbol{x} \rangle$ its empirical mean, i.e., $\langle \boldsymbol{x} \rangle = \frac{1}{n} \sum_{i=1}^n x_i$. The empirical distribution of $\boldsymbol{x} = (x_1, \ldots, x_n)^\mathsf{T}$ is given by $\frac{1}{n} \sum_{i=1}^n \delta_{x_i}$, where $\delta_{x_i}$ denotes a Dirac delta mass on $x_i$. The notation $\boldsymbol{x} \xrightarrow{W} X$ denotes convergence of the empirical distribution of $\boldsymbol{x}$ to the random variable $X$ in Wasserstein distance at all orders. Given a symmetric square matrix $\boldsymbol{A} \in \mathbb{R}^{n \times n}$, we denote by $\lambda_1(\boldsymbol{A}) \geq \lambda_2(\boldsymbol{A}) \geq \ldots \geq \lambda_n(\boldsymbol{A})$ its eigenvalues sorted in decreasing order. Given a rectangular matrix $\boldsymbol{A} \in \mathbb{R}^{m \times n}$, with $m < n$, we denote by $\sigma_1(\boldsymbol{A}) \geq \sigma_2(\boldsymbol{A}) \geq \ldots \sigma_m(\boldsymbol{A})$ its singular values sorted in decreasing order.

**Rank-$1$ estimation – Symmetric square matrices.** Consider the problem of estimating the signal $\boldsymbol{u}^* \in \mathbb{R}^n$ from the data matrix $\boldsymbol{X}$ in (1.1). We assume that $\boldsymbol{W}$ is rotationally invariant in law, i.e., $\boldsymbol{W} = \boldsymbol{O}^\mathsf{T} \boldsymbol{\Lambda} \boldsymbol{O}$, where $\boldsymbol{\Lambda} = \mathrm{diag}(\boldsymbol{\lambda})$ is a diagonal matrix containing the eigenvalues of $\boldsymbol{W}$ and $\boldsymbol{O}$ is a Haar orthogonal matrix independent of $\boldsymbol{\Lambda}$. As $n \to \infty$, we assume that the empirical distributions of $\boldsymbol{\lambda}$ and $\boldsymbol{u}^*$ satisfy

$$\boldsymbol{\lambda} \xrightarrow{W} \Lambda \quad \text{and} \quad \boldsymbol{u}^* \xrightarrow{W} U_*, \tag{2.1}$$

where $\Lambda$ and $U_*$ represent the limiting spectral distribution of the noise and the prior on the signal, respectively. We take $\|\boldsymbol{u}\| = \sqrt{n}$ so that $\mathbb{E}\{U_*^2\} = \lim_{n \to \infty} \frac{1}{n} \|\boldsymbol{u}^*\|^2 = 1$. We assume that the moment $\mathbb{E}\{U_*^{2+\varepsilon}\} < \infty$ for some $\varepsilon > 0$. We also assume that $\Lambda$ has compact support, and denote by $b$ the supremum of this support. We denote by $\{\kappa_k\}_{k \geq 1}$ the free cumulants corresponding to the moments $\{m_k\}_{k \geq 1}$ of the empirical eigenvalue distribution of $\boldsymbol{X}$ excluding its largest eigenvalue, i.e., $m_k = \frac{1}{n} \sum_{i=2}^n \lambda_i(\boldsymbol{X})^k$ (for details, see (A.1)-(A.2) in Appendix A). The assumption (2.1) implies that, as $n \to \infty$, $m_k \to m_k^\infty = \mathbb{E}\{\Lambda^k\}$ and $\kappa_k \to \kappa_k^\infty$, where $\{m_k^\infty\}_{k \geq 1}$ and $\{\kappa_k^\infty\}_{k \geq 1}$ are respectively moments and free cumulants of $\Lambda$.

**PCA – Symmetric square matrices.** Let $\boldsymbol{u}_{\mathrm{PCA}}$ be the principal eigenvector of $\boldsymbol{X}$, and define $\alpha_\mathrm{s} = 1/G(b^+)$, where $G(z) = \mathbb{E}\{(z - \Lambda)^{-1}\}$ is the Cauchy transform of $\Lambda$, and $G(b^+) = \lim_{z \to b^+} G(z)$.

Then, for $\alpha > \alpha_{\mathrm{s}}$, $\lambda_1(\boldsymbol{X}) \xrightarrow{\text{a.s.}} G^{-1}(1/\alpha)$ and $\lambda_2(\boldsymbol{X}) \xrightarrow{\text{a.s.}} b$, where $G^{-1}$ is the inverse of $G$; see Theorem 2.1 in [10]. Furthermore, Theorem 2.2 in [10] gives that, for $\alpha > \alpha_{\mathrm{s}}$,

$$\frac{\langle \boldsymbol{u}_{\mathrm{PCA}}, \boldsymbol{u}^* \rangle^2}{n} \xrightarrow{\text{a.s.}} \rho_\alpha^2 = \frac{-1}{\alpha^2 G'(G^{-1}(1/\alpha))} > 0. \tag{2.2}$$

In words, above the spectral threshold $\alpha_{\mathrm{s}}$, the principal eigenvalue of $\boldsymbol{X}$ escapes the bulk of the spectrum and its associated eigenvector becomes strictly correlated with the signal $\boldsymbol{u}^*$.

**Rank-1 estimation – Rectangular matrices.**  Consider now the problem of estimating the signals $\boldsymbol{u}^* \in \mathbb{R}^m$ and $\boldsymbol{v}^* \in \mathbb{R}^n$ given the rectangular data matrix $\boldsymbol{X}$ in (1.2). Without loss of generality, we assume that $m \leq n$ (if $m > n$, one can just exchange the role of $\boldsymbol{u}^*$ and $\boldsymbol{v}^*$ and consider $\boldsymbol{X}^{\mathsf{T}}$ in place of $\boldsymbol{X}$). We assume that $W$ is bi-rotationally invariant in law, i.e., $\boldsymbol{W} = \boldsymbol{O}^{\mathsf{T}} \boldsymbol{\Lambda} \boldsymbol{Q}$, where $\boldsymbol{\Lambda} = \mathrm{diag}(\boldsymbol{\lambda})$ is a $m \times n$ diagonal matrix containing the singular values of $\boldsymbol{W}$, and $\boldsymbol{O}, \boldsymbol{Q}$ are Haar orthogonal matrices independent of one another and also of $\boldsymbol{\Lambda}$. As $n \to \infty$, we assume that $\boldsymbol{\lambda} \xrightarrow{W} \Lambda$, $\boldsymbol{u}^* \xrightarrow{W} U_*$, $\boldsymbol{v}^* \xrightarrow{W} V_*$ and $m/n \to \gamma$, for some constant $\gamma \in (0, 1]$. We take $\|\boldsymbol{u}\| = \sqrt{m}$ and $\|\boldsymbol{v}\| = \sqrt{n}$ so that $\mathbb{E}\{U_*^2\} = \mathbb{E}\{V_*^2\} = 1$. As before, $b < \infty$ is the supremum of the compact support of $\Lambda$, and $U_*, V_*$ are assumed to have finite $(2 + \varepsilon)$-th moment for some $\varepsilon > 0$. To analyze PCA using the framework in [11], we also assume that the entries of $\boldsymbol{u}^*$ and $\boldsymbol{v}^*$ are i.i.d., and their law has zero mean and satisfies a log-Sobolev inequality. We denote by $\{\kappa_{2k}\}_{k \geq 1}$ the rectangular free cumulants associated to the even moments $\{m_{2k}\}_{k \geq 1}$, with $m_{2k} = \frac{1}{m} \sum_{i=2}^m \sigma_i(\boldsymbol{X})^{2k}$ (for details, see (A.11)-(A.12) in Appendix A). Furthermore, as $n, m \to \infty$, $m_{2k} \to m_{2k}^\infty = \mathbb{E}\{\Lambda^{2k}\}$ and $\kappa_{2k} \to \kappa_{2k}^\infty$, where $\{m_{2k}^\infty\}_{k \geq 1}$ and $\{\kappa_{2k}^\infty\}_{k \geq 1}$ are respectively even moments and rectangular free cumulants of $\Lambda$.

**PCA – Rectangular matrices.**  Denote by $\boldsymbol{u}_{\mathrm{PCA}}$ and $\boldsymbol{v}_{\mathrm{PCA}}$ the left and right principal singular vectors of $\boldsymbol{X}$, and define $\tilde{\alpha}_{\mathrm{s}} = 1/\sqrt{D(b^+)}$, where $D(z) = \phi(z)\bar{\phi}(z)$, $\phi(z) = \mathbb{E}\{z/(z^2 - \Lambda^2)\}$, $\bar{\phi}(z) = \gamma \phi(z) + (1-\gamma)/z$, and $D(b^+) = \lim_{z \to b^+} D(z)$. Note that the singular value of the rank-one signal $\frac{\alpha}{m} \boldsymbol{u}^* \boldsymbol{v}^{*\mathsf{T}}$ is $\tilde{\alpha} \triangleq \alpha/\sqrt{\gamma}$. Then, for $\tilde{\alpha} > \tilde{\alpha}_{\mathrm{s}}$, $\sigma_1(\boldsymbol{X}) \xrightarrow{\text{a.s.}} D^{-1}(1/\tilde{\alpha}^2)$ and $\sigma_2(\boldsymbol{X}) \xrightarrow{\text{a.s.}} b$; see Theorem 2.8 in [11]. Furthermore, Theorem 2.9 in [11] gives that, for $\tilde{\alpha} > \tilde{\alpha}_{\mathrm{s}}$,

$$\frac{\langle \boldsymbol{u}_{\mathrm{PCA}}, \boldsymbol{u}^* \rangle^2}{m} \xrightarrow{\text{a.s.}} \Delta_{\mathrm{PCA}} = \frac{-2\phi(D^{-1}(1/\tilde{\alpha}^2))}{\tilde{\alpha}^2 D'(D^{-1}(1/\tilde{\alpha}^2))} > 0, \tag{2.3}$$

$$\frac{\langle \boldsymbol{v}_{\mathrm{PCA}}, \boldsymbol{v}^* \rangle^2}{n} \xrightarrow{\text{a.s.}} \Gamma_{\mathrm{PCA}} = \frac{-2\bar{\phi}(D^{-1}(1/\tilde{\alpha}^2))}{\tilde{\alpha}^2 D'(D^{-1}(1/\tilde{\alpha}^2))} > 0. \tag{2.4}$$

In words, above the spectral threshold $\tilde{\alpha}_{\mathrm{s}}$, the principal singular value escapes from the bulk of the spectrum and the left/right principal singular vectors become correlated with the signal $\boldsymbol{u}^*/\boldsymbol{v}^*$.

## 3  PCA Initialization for Approximate Message Passing

### 3.1  Symmetric Square Matrices

We consider a family of Approximate Message Passing (AMP) algorithms to estimate $\boldsymbol{u}^*$ from $\boldsymbol{X} = \frac{\alpha}{n} \boldsymbol{u}^* \boldsymbol{u}^{*\mathsf{T}} + \boldsymbol{W}$. We initialize using the PCA estimate $\boldsymbol{u}_{\mathrm{PCA}}$:

$$\boldsymbol{u}^1 = \sqrt{n} \boldsymbol{u}_{\mathrm{PCA}}, \quad \boldsymbol{f}^1 = \boldsymbol{X} \boldsymbol{u}^1 - \mathsf{b}_{1,1} \boldsymbol{u}^1, \tag{3.1}$$

with $\mathsf{b}_{1,1} = \sum_{i=0}^\infty \kappa_{i+1} \alpha^{-i}$. Then, for $t \geq 2$, the algorithm computes

$$\boldsymbol{u}^t = \mathsf{u}_t(\boldsymbol{f}^{t-1}), \quad \boldsymbol{f}^t = \boldsymbol{X} \boldsymbol{u}^t - \sum_{i=1}^t \mathsf{b}_{t,i} \boldsymbol{u}^i, \tag{3.2}$$

where the memory coefficients $\{\mathsf{b}_{t,i}\}_{i \in [1,t]}$ are given by $\mathsf{b}_{t,t} = \kappa_1$, and

$$\mathsf{b}_{t,1} = \prod_{\ell=2}^t \langle \mathsf{u}'_\ell(\boldsymbol{f}^{\ell-1}) \rangle \sum_{i=0}^\infty \kappa_{i+t} \alpha^{-i}, \quad \mathsf{b}_{t,t-j} = \kappa_{j+1} \prod_{i=t-j+1}^t \langle \mathsf{u}'_i(\boldsymbol{f}^{i-1}) \rangle, \text{ for } (t-j) \in [2, t-1]. \tag{3.3}$$

Here, the function $\mathsf{u}_t : \mathbb{R} \to \mathbb{R}$ is continuously differentiable and Lipschitz, it is applied component-wise to vectors, i.e., $\mathsf{u}_t(\boldsymbol{f}^{t-1}) = (\mathsf{u}_t(f_1^{t-1}), \ldots, \mathsf{u}_t(f_n^{t-1}))$, and $\mathsf{u}_t'$ denotes its derivative. The AMP algorithm in (3.1)-(3.3) is similar to the one in [20, Sec. 3.1] (and the ones in [14, 48]), with the main differences being the initialization $\boldsymbol{u}^1$ and the formula for the memory term $\mathsf{b}_{t,1}$. We highlight that the algorithm does not require the knowledge of $\alpha$ or of the noise distribution. In fact, $\alpha$ can be consistently estimated from the principal eigenvalue of $\boldsymbol{X}$ via $\hat{\alpha} = (G(\lambda_1(\boldsymbol{X})))^{-1}$. Furthermore, one can compute the moments $\{m_k\}_{k \geq 1}$ of the empirical eigenvalue distribution of $\boldsymbol{X}$ (excluding its largest one) and, from these, deduce the free cumulants $\{\kappa_k\}_{k \geq 1}$.

The asymptotic empirical distribution of the iterates $\boldsymbol{u}^t, \boldsymbol{f}^t$, for $t \geq 1$, can be succinctly characterized via a deterministic recursion, called *state evolution*, and expressed via a sequence of mean vectors $\boldsymbol{\mu}_K = (\mu_t)_{t \in [1,K]}$ and covariance matrices $\boldsymbol{\Sigma}_K = (\sigma_{s,t})_{s,t \in [1,K]}$. For $K = 1$, set $\mu_1 = \alpha \rho_\alpha$ and $\sigma_{11} = \alpha^2(1 - \rho_\alpha^2)$, with $\rho_\alpha$ given in (2.2). Then define $\boldsymbol{\mu}_{K+1}, \boldsymbol{\Sigma}_{K+1}$ from $\boldsymbol{\mu}_K, \boldsymbol{\Sigma}_K$ as follows. Let

$$(F_1, \ldots, F_K) = \boldsymbol{\mu}_K U_* + (Z_1, \ldots, Z_K), \text{ where } (Z_1, \ldots, Z_K) \sim \mathcal{N}(\boldsymbol{0}, \boldsymbol{\Sigma}_K), \tag{3.4}$$

$$U_t = \mathsf{u}_t(F_{t-1}) \text{ for } 2 \leq t \leq K+1, \quad \text{and} \quad U_t = \frac{F_1}{\alpha} \text{ for } -\infty < t \leq 1. \tag{3.5}$$

Then, the entries of $\boldsymbol{\mu}_{K+1}$ are given by $\mu_t = \alpha \mathbb{E}\{U_t U_*\}$ for $t \in [1, K+1]$. Furthermore, the entries of $\boldsymbol{\Sigma}_{K+1}$ can be expressed via the following formula, for $s, t \in [1, K+1]$:

$$\sigma_{s,t} = \sum_{j=0}^{\infty} \sum_{k=0}^{\infty} \kappa_{j+k+2}^{\infty} \left(\frac{1}{\alpha}\right)^{(k-t+1)_+ + (j-s+1)_+} \cdot \mathbb{E}\{U_{s-j} U_{t-k}\} \cdot \left(\prod_{i=\max(2,s+1-j)}^{s} \mathbb{E}\{\mathsf{u}_i'(F_{i-1})\}\right) \cdot \left(\prod_{i=\max(2,t+1-k)}^{t} \mathbb{E}\{\mathsf{u}_i'(F_{i-1})\}\right). \tag{3.6}$$

Our main result, Theorem 1, shows that for $t \geq 1$, the empirical joint distribution of the entries of $(\boldsymbol{u}^*, \boldsymbol{f}^1, \ldots, \boldsymbol{f}^t)$ converges in Wasserstein distance $W_2$ to the law of the random vector $(U_*, F_1, \ldots, F_t)$. We provide a proof sketch in Section 5, and the complete proof is deferred to Appendix B. This result is stated in terms of *pseudo-Lipschitz* test functions. A function $\psi : \mathbb{R}^m \to \mathbb{R}$ is pseudo-Lipschitz of order 2, i.e., $\psi \in \mathrm{PL}(2)$, if there is a constant $C > 0$ such that

$$\|\psi(\boldsymbol{x}) - \psi(\boldsymbol{y})\| \leq C(1 + \|\boldsymbol{x}\| + \|\boldsymbol{y}\|) \|\boldsymbol{x} - \boldsymbol{y}\|. \tag{3.7}$$

The equivalence between convergence in terms of $\mathrm{PL}(2)$ functions and convergence in $W_2$ distance follows from [58, Definition 6.7 and Theorem 6.8].

**Theorem 1.** *In the square symmetric model* (1.1)*, assume that $\alpha > \alpha_s$, and that the free cumulants of order 2 and higher are non-negative, i.e., $\kappa_k^\infty \geq 0$ for $k \geq 2$. Consider the AMP algorithm with PCA initialization in* (3.1)-(3.2)*, with continuously differentiable and Lipschitz functions $\mathsf{u}_t : \mathbb{R} \to \mathbb{R}$. (Without loss of generality, assume that $\langle \boldsymbol{u}^*, \boldsymbol{u}^{\mathrm{PCA}} \rangle \geq 0$.)*

*Then, for $t \geq 1$ and any* $\mathrm{PL}$*(2) function $\psi : \mathbb{R}^{2t+2} \to \mathbb{R}$, we almost surely have:*

$$\lim_{n \to \infty} \frac{1}{n} \sum_{i=1}^{n} \psi(u_i^*, u_i^1, \ldots, u_i^{t+1}, f_i^1, \ldots f_i^t) = \mathbb{E}\left\{\psi(U_*, U_1, \ldots, U_{t+1}, F_1, \ldots, F_t)\right\}, \tag{3.8}$$

*where $U_1, \ldots, U_{t+1}$ and $F_1, \ldots, F_t$ are defined in* (3.4)*.*

**Assumptions of the theorem.** The basic assumption that the noise matrix is rotationally invariant is rather mild as it allows for arbitrary eigenvalue distributions. The assumption $\alpha > \alpha_s$ ensures that the PCA initialization is correlated with the signal. This condition is necessary and sufficient for PCA to be effective: under the additional requirement that $G'(b) = -\infty$, we have that, if $\alpha < \alpha_s$, then the normalized correlation between $\boldsymbol{u}_{\mathrm{PCA}}$ and $\boldsymbol{u}^*$ vanishes almost surely; see Theorem 2.3 of [10]. Conversely for $\alpha > \alpha_s$, the asymptotic correlation is strictly non-zero and given by (2.2).

*Non-negativity of free cumulants*: The assumption that $\kappa_k^\infty \geq 0$ for $k \geq 2$ appears to be an artifact of the proof technique. As detailed in the proof sketch in Section 5, this assumption is needed to show that the state evolution of the artificial AMP in the first phase has a unique fixed point. We expect our approach to generalize to any limiting noise distribution $\Lambda$ with compact support, and defer such a generalization to future work. In support of this view, the simulations of Section 4 verify the

claim of Theorem 1 in a setting where the free cumulants of $\Lambda$ have alternating signs (corresponding to an eigenvalue distribution $\Lambda \sim \text{Uniform}[-1/2, 1/2]$; see Figs. 1b-1d and 2b–2d). Finally, we remark that, if $W$ follows a Marcenko-Pastur distribution ($W = AA^\mathsf{T}$, where $A$ has i.i.d. Gaussian entries), then the free cumulants of $\Lambda$ are all equal and strictly positive; see [40, Chap. 2, Exercise 11]. Thus, the assumption of Theorem 1 holds for noise distributions that are sufficiently close to the Marcenko-Pastur one, or for sufficiently large values of the signal-to-noise ratio $\alpha$.[1]

*Continuous differentiability and other technical assumptions*: The assumption that $u_t$ is continuously differentiable can be weakened to: *(i)* $u_t$ being differentiable almost everywhere, and *(ii)* satisfying a mild non-degeneracy condition (Assumption 4.2(e) in [20]). In this way, we can cover most practically relevant choices of $u_t$ such as soft thresholding and ReLU. Theorem 1 also requires the technical assumptions in (2.1) and the text below it: convergence of the empirical distributions of the signal and of the eigenvalues of the noise matrix; boundedness of the $(2 + \varepsilon)$-moment of the signal; and compact support of the spectrum of the noise matrix. We regard these technical assumptions as minor, and remark that they are quite standard in the literature. For the rectangular case, we also need the additional assumption that the law of the signal is zero mean and satisfies a log-Sobolev inequality, which is necessary to apply the framework in [10].

**How PCA initialization influences AMP.** The form of the memory coefficient $b_{t,1}$ in (3.3) reflects the PCA initialization of the AMP iteration. PCA initialization can be interpreted as the result of a first AMP phase with linear denoisers (see the proof sketch in Sec. 5). The coefficient $b_{t,1}$ multiplying the initialization $u_1$ represents the cumulative effect of this first AMP phase leading to the PCA estimate. The main differences from the AMP algorithm in [20] (where the initialization is independent of $W$) are the expressions for the coefficient $b_{t,1}$ and the state evolution parameters $\sigma_{s,t}$ (compare (3.3) and (3.6) in this paper with (1.15) and (1.17) in [20]). One can interpret the new form of $b_{t,1}$ and $\sigma_{s,t}$ as a memory of the PCA initialization. For the special case of Gaussian noise, the spectral initialization only affects the first iteration of AMP [44]. This is due to the fact that, while in a rotationally invariant model the AMP iterate at step $t$ depends on *all* previous iterates, in the Gaussian case it depends only on the iterate at step $t-1$.

**Choice of $u_t(\cdot)$.** Theorem 1 holds for any choice of denoisers $\{u_t\}$ that are Lipschitz and continuously differentiable. Indeed, our analysis shows that by picking $u_t(f) = f/\alpha$, AMP just returns the PCA estimate; see the proof sketch in Section 5. If some structural information about the signal is available (e.g., sparsity), denoisers that take advantage of this structure can give substantial improvements over PCA. Thus, a key question is how to optimally select the $u_t$'s. Theorem 1 tells us that the empirical distribution of $f^t$ converges to the law of $\mu_t U_* + \sqrt{\sigma_{t,t}} Z$, for $Z \sim \mathcal{N}(0,1)$ and independent of $U_*$. Hence, the quality of the estimate at each iteration $t$ is governed by the SNR $\rho_t := \mu_t^2 / \sigma_{t,t}$. Consider running the algorithm for $\bar{t}$ iterations, and let $u^{\bar{t}+1} = u_{\bar{t}+1}(f^{\bar{t}})$ be the final estimate. Then, for each $t \in [2, \bar{t}]$, the Bayes-optimal choice for $u_t$ is the one that maximizes $\rho_t$, i.e., the SNR for the next iteration. In the case of Gaussian noise [44], the maximum is achieved by the posterior mean $u_t(f) = \mathbb{E}\{U_* \mid \mu_t U_* + \sqrt{\sigma_{t,t}} Z = f\}$. For rotationally invariant noise, this choice minimizes the mean-squared error $\frac{1}{n}\|u^t - u^*\|^2$ (for fixed $u_1, \ldots, u_{t-1}$), but it *does not* necessarily maximize the SNR $\rho_t$. We provide an example of this behavior in the simulations reported in Section 4. Therefore, the optimal strategy would be to choose functions $u_2, \ldots, u_{\bar{t}}$ to maximize the SNRs $\rho_2, \ldots, \rho_{\bar{t}}$, and then in the final iteration, to pick $u_{\bar{t}+1}$ to minimize the desired loss. Note that $u_t$ depends on the previously chosen functions $u_1, \ldots, u_{t-1}$ in a complicated way, due to the definition of $\sigma_{t,t}$ in (3.6). Thus, finding $u_t$ that maximizes the SNR $\rho_t$ remains an outstanding challenge. Finally, we remark that though we only consider one-step denoisers in this paper, Theorem 1 can be readily extended to cover denoisers with memory, i.e., those of the form $u_t(f^1, \ldots, f^{t-1})$.

## 3.2 Rectangular Matrices

We now present an AMP algorithm to estimate $u^*$ and $v^*$ from the $m \times n$ data matrix $X = \frac{\alpha}{m} u^* v^{*\mathsf{T}} + W$. We initialize the algorithm using the PCA estimate $u_{\text{PCA}}$:

---

[1]One can add an independent artificial noise matrix with Marcenko-Pastur distribution to the data in order to make the required free cumulants non-negative, and the result would hold for $\alpha$ greater than the new spectral threshold.

$$\boldsymbol{u}^1 = \sqrt{m}\,\boldsymbol{u}_{\mathrm{PCA}}, \quad \boldsymbol{g}^1 = \left(1 + \gamma\sum_{i=1}^{\infty}\kappa_{2i}\Big(\frac{\gamma}{\alpha^2}\Big)^i\right)^{-1}\boldsymbol{X}^{\mathsf{T}}\boldsymbol{u}^1, \quad \boldsymbol{v}^1 = \mathsf{v}_1(\boldsymbol{g}^1) = \frac{\gamma}{\alpha}\boldsymbol{g}^1. \qquad (3.9)$$

Then, for $t \geq 1$, we iteratively compute:

$$\boldsymbol{f}^t = \boldsymbol{X}\boldsymbol{v}^t - \sum_{i=1}^{t}\mathsf{a}_{t,i}\boldsymbol{u}^i, \quad \boldsymbol{u}^{t+1} = \mathsf{u}_{t+1}(\boldsymbol{f}^t), \quad \boldsymbol{g}^{t+1} = \boldsymbol{X}\boldsymbol{u}^{t+1} - \sum_{i=1}^{t}\mathsf{b}_{t+1,i}\boldsymbol{v}^i, \quad \boldsymbol{v}^{t+1} = \mathsf{v}_{t+1}(\boldsymbol{g}^{t+1}).$$

$$(3.10)$$

Here, $\mathsf{u}_{t+1}, \mathsf{v}_{t+1} : \mathbb{R} \to \mathbb{R}$ are continuously differentiable Lipschitz functions that act component-wise on vectors. We define $\mathsf{a}_{1,1} = \alpha\sum_{i=1}^{\infty}\kappa_{2i}\big(\frac{\gamma}{\alpha^2}\big)^i$, and for $t \geq 2$:

$$\mathsf{a}_{t,1} = \langle\mathsf{v}_t'(\boldsymbol{g}^t)\rangle\prod_{i=2}^{t}\langle\mathsf{u}_i'(\boldsymbol{f}^{i-1})\rangle\langle\mathsf{v}_{i-1}'(\boldsymbol{g}^{i-1})\rangle\left(\sum_{i=0}^{\infty}\kappa_{2(i+t)}\Big(\frac{\gamma}{\alpha^2}\Big)^i\right), \qquad (3.11)$$

$$\mathsf{a}_{t,t-j} = \langle\mathsf{v}_t'(\boldsymbol{g}^t)\rangle\prod_{i=t-j+1}^{t}\langle\mathsf{u}_i'(\boldsymbol{f}^{i-1})\rangle\langle\mathsf{v}_{i-1}'(\boldsymbol{g}^{i-1})\rangle\kappa_{2(j+1)}, \qquad \text{for } (t-j) \in [2,t]. \qquad (3.12)$$

Furthermore, for $t \geq 1$,

$$\mathsf{b}_{t+1,1} = \gamma\langle\mathsf{u}_{t+1}'(\boldsymbol{f}^t)\rangle\prod_{i=2}^{t}\langle\mathsf{v}_i'(\boldsymbol{g}^i)\rangle\langle\mathsf{u}_i'(\boldsymbol{f}^{i-1})\rangle\left(\kappa_{2t} + \sum_{i=1}^{\infty}\kappa_{2(i+t)}\Big(\frac{\gamma}{\alpha^2}\Big)^i\right), \qquad (3.13)$$

$$\mathsf{b}_{t+1,t+1-j} = \gamma\langle\mathsf{u}_{t+1}'(\boldsymbol{f}^t)\rangle\prod_{i=t+2-j}^{t}\langle\mathsf{v}_i'(\boldsymbol{g}^i)\rangle\langle\mathsf{u}_i'(\boldsymbol{f}^{i-1})\rangle\,\kappa_{2j}, \qquad \text{for } (t+1-j) \in [2,t]. \quad (3.14)$$

Similarly to the square case, $\alpha$ can be consistently estimated from the largest singular value of $\boldsymbol{X}$ via $\alpha = \sqrt{\gamma(D(\sigma_1(\boldsymbol{X})))^{-1}}$, and the rectangular free cumulants $\{\kappa_{2k}\}_{k\geq 1}$ can be obtained from the even moments of the empirical distribution of the singular values of $\boldsymbol{X}$ (excluding its largest one).

The asymptotic empirical distributions of the iterates $(\boldsymbol{f}^t, \boldsymbol{g}^t)$ can be characterized via a state evolution recursion, which specifies a sequence of mean vectors $\boldsymbol{\mu}_K = (\mu_t)_{t\in[0,K]}$, $\boldsymbol{\nu}_K = (\nu_t)_{t\in[1,K]}$ and covariance matrices $\boldsymbol{\Sigma}_K = (\sigma_{s,t})_{s,t\in[0,K]}$, $\boldsymbol{\Omega}_K = (\omega_{s,t})_{s,t\in[1,K]}$. These are iteratively defined, starting with the initialization $\mu_0 = \alpha\sqrt{\Delta_{\mathrm{PCA}}}$ and $\sigma_{0,0} = \alpha^2(1 - \Delta_{\mathrm{PCA}})$, where $\Delta_{\mathrm{PCA}}$ is given by (2.3). Having defined $\boldsymbol{\mu}_K, \boldsymbol{\Sigma}_K, \boldsymbol{\nu}_K, \boldsymbol{\Omega}_K$, let

$$(F_0, \ldots, F_K) = \boldsymbol{\mu}_K U_* + (Y_0, \ldots, Y_K), \text{ where } (Y_0, \ldots, Y_K) \sim \mathcal{N}(\boldsymbol{0}, \boldsymbol{\Sigma}_K), \qquad (3.15)$$

$$U_t = \mathsf{u}_t(F_{t-1}) \text{ for } 2 \leq t \leq K+1, \quad \text{and} \quad U_t = \frac{F_0}{\alpha} \text{ for } -\infty < t \leq 1, \qquad (3.16)$$

$$(G_1, \ldots, G_K) = \boldsymbol{\nu}_K V_* + (Z_1, \ldots, Z_K), \text{ where } (Z_1, \ldots, Z_K) \sim \mathcal{N}(\boldsymbol{0}, \boldsymbol{\Omega}_K), \qquad (3.17)$$

$$V_t = \mathsf{v}_t(G_t) \text{ for } 2 \leq t \leq K+1, \quad \text{and} \quad V_t = \frac{\gamma}{\alpha}G_1 \text{ for } -\infty < t \leq 1. \qquad (3.18)$$

Given $\boldsymbol{\mu}_K$ and $\boldsymbol{\Sigma}_K$, the entries of $\boldsymbol{\nu}_{K+1}$ are given by $\nu_t = \alpha\mathbb{E}\{U_t U_*\}$ (for $t \in [1, K+1]$), and the entries of $\boldsymbol{\Omega}_{K+1}$ (for $s+1, t+1 \in [1, K+1]$) are given by

$$\omega_{s+1,t+1} = \sum_{j=0}^{\infty}\sum_{k=0}^{\infty}\gamma\Big(\frac{\gamma}{\alpha^2}\Big)^{(j-s)_+ + (k-t)_+}\Big(\prod_{i=\max(2,s+2-j)}^{s+1}\mathsf{x}_i\cdot\mathsf{y}_{i-1}\Big)\cdot\Big(\prod_{i=\max(2,t+2-k)}^{t+1}\mathsf{x}_i\cdot\mathsf{y}_{i-1}\Big)$$

$$\cdot\Big[\kappa_{2(j+k+1)}^{\infty}\mathbb{E}\{U_{s+1-j}U_{t+1-k}\} + \kappa_{2(j+k+2)}^{\infty}\mathbb{E}\{V_{s-j}V_{t-k}\}\mathsf{x}_{s+1-j}\cdot\mathsf{x}_{t+1-k}\Big]. \quad (3.19)$$

Here, we define $\mathsf{x}_i = \mathbb{E}\{\mathsf{u}_i'(F_{i-1})\}$ if $i \geq 2$, and $\mathsf{x}_i = 1/\alpha$ otherwise; $\mathsf{y}_i = \mathbb{E}\{\mathsf{v}_i'(G_i)\}$ if $i \geq 2$, and $\mathsf{y}_i = \gamma/\alpha$ otherwise. We note that $\omega_{11}$ is computed by solving the linear equation obtained by setting $s = t = 0$ in (3.19) (see (C.96)). Next, given $\boldsymbol{\nu}_{K+1}$ and $\boldsymbol{\Omega}_{K+1}$ for some $K \geq 1$, the entries of $\boldsymbol{\mu}_{K+1}$ are $\mu_t = \frac{\alpha}{\gamma}\mathbb{E}\{V_t V_*\}$ (for $t \in [0, K+1]$), and the entries of $\boldsymbol{\Sigma}_{K+1}$ (for $s, t \in [0, K+1]$) are

$$\sigma_{s,t} = \sum_{j=0}^{\infty}\sum_{k=0}^{\infty}\Big(\frac{\gamma}{\alpha^2}\Big)^{(j-s+1)_+ + (k-t+1)_+}\Big(\prod_{i=\max(2,s+1-j)}^{s}\mathsf{x}_i\cdot\mathsf{y}_i\Big)\cdot\Big(\prod_{i=\max(2,t+1-k)}^{t}\mathsf{x}_i\cdot\mathsf{y}_i\Big)$$

$$\cdot\Big[\kappa_{2(j+k+1)}^{\infty}\mathbb{E}\{V_{s-j}V_{t-k}\} + \kappa_{2(j+k+2)}^{\infty}\mathbb{E}\{U_{s-j}U_{t-k}\}\mathsf{y}_{s-j}\cdot\mathsf{y}_{t-k}\Big]. \qquad (3.20)$$

Our main result for the rectangular case, Theorem 2, shows that for $t \geq 1$, the empirical joint distribution of the entries of $(\boldsymbol{u}^*, \boldsymbol{f}^1, \ldots, \boldsymbol{f}^t)$ converges in Wasserstein distance $W_2$ to the law of the random vector $(U_*, F_1, \ldots, F_t)$. Similarly, the empirical joint distribution of the entries of $(\boldsymbol{v}^*, \boldsymbol{g}^1, \ldots, \boldsymbol{g}^t)$ converges to the law of $(V_*, G_1, \ldots, G_t)$. The proof is given in Appendix C. As in the square case, we state this result in terms of pseudo-Lipschitz test functions.

**Theorem 2.** *In the rectangular model* (1.2)*, assume that $\tilde{\alpha} > \tilde{\alpha}_s$ and that $\kappa_{2k}^\infty \geq 0$ for $k \geq 1$. Consider the AMP algorithm with PCA initialization in* (3.9)-(3.10)*, with continuously differentiable and Lipschitz functions* $\mathsf{u}_t, \mathsf{v}_t : \mathbb{R} \to \mathbb{R}$*. (Assume without loss of generality that $\langle \boldsymbol{u}^*, \boldsymbol{u}^{\mathrm{PCA}} \rangle \geq 0$.)*

*Then, for $t \geq 1$ and any* PL(2) *functions $\psi : \mathbb{R}^{2t+2} \to \mathbb{R}$ and $\varphi : \mathbb{R}^{2t+1} \to \mathbb{R}$, we almost surely have:*

$$\lim_{m \to \infty} \frac{1}{m} \sum_{i=1}^m \psi(u_i^*, u_i^1, \ldots, u_i^{t+1}, f_i^1, \ldots f_i^t) = \mathbb{E} \left\{ \psi(U_*, U_1, \ldots, U_{t+1}, F_1, \ldots, F_t) \right\}, \quad (3.21)$$

$$\lim_{n \to \infty} \frac{1}{n} \sum_{i=1}^n \varphi(v_i^*, v_i^1, \ldots, v_i^t, g_i^1, \ldots g_i^t) = \mathbb{E} \left\{ \varphi(V_*, V_1, \ldots, V_t, G_1, \ldots, G_t) \right\}, \quad (3.22)$$

*where $(U_1, \ldots, U_{t+1})$, $(F_1, \ldots, F_t)$, $(V_1, \ldots, V_t)$ and $(G_1, \ldots, G_t)$ are defined as in* (3.15)-(3.18)*.*

The condition $\tilde{\alpha} > \tilde{\alpha}_s$ is necessary and sufficient for PCA to be effective: under the additional requirement that $\phi'(b^+) = -\infty$, if $\tilde{\alpha} < \tilde{\alpha}_s$, then the normalized correlation between $\boldsymbol{u}_{\mathrm{PCA}}$ and $\boldsymbol{u}^*$ vanishes almost surely, see [11, Theorem 2.10]. Comments similar to those at the end of Section 3.1 can be made about *(i)* the requirement that the rectangular free cumulants are non-negative, *(ii)* the effect of the PCA initialization on AMP, and *(iii)* the choice of the denoisers $\mathsf{u}_t, \mathsf{v}_t$.

## 4 Numerical Simulations

We consider the following settings: *(i)* square model (1.1) with Marcenko-Pastur noise, i.e., $\boldsymbol{W} = \frac{1}{n} \boldsymbol{A} \boldsymbol{A}^\mathsf{T} \in \mathbb{R}^{n \times n}$, where the entries of $\boldsymbol{A} \in \mathbb{R}^{n \times p}$ are i.i.d. standard Gaussian, see (a) in the figures; *(ii)* square model (1.1) with uniform noise, i.e., $\boldsymbol{W} = \boldsymbol{O}^\mathsf{T} \boldsymbol{\Lambda} \boldsymbol{O} \in \mathbb{R}^{n \times n}$, where $\boldsymbol{O}$ is a Haar orthogonal matrix and the entries of $\boldsymbol{\Lambda}$ are i.i.d. and uniformly distributed in the interval $[-1/2, 1/2]$, see (b) in the figures; *(iii)* rectangular model (1.2) with uniform noise, i.e., $\boldsymbol{W} = \boldsymbol{O}^\mathsf{T} \boldsymbol{\Lambda} \boldsymbol{Q} \in \mathbb{R}^{m \times n}$, where $\boldsymbol{O}, \boldsymbol{Q}$ are Haar orthogonal matrices and the entries of $\boldsymbol{\Lambda}^2$ are i.i.d. and uniformly distributed in the interval $[0, 1]$, see (c)-(d) in the figures.

In the simulations, $\alpha$ is estimated from the largest eigenvalue/singular value of $\boldsymbol{X}$. Furthermore, the free cumulants $\kappa_k$ ($\kappa_{2k}$ in the rectangular case) are replaced by their limits $\kappa_k^\infty$ ($\kappa_{2k}^\infty$ resp.), which are obtained as follows. For (a), all the free cumulants of $\Lambda$ are equal to $c \triangleq p/n$, i.e., $\kappa_k^\infty = c$ for $k \geq 1$, see [40, Chap. 2, Exercise 11]. For (b), the odd free cumulants of $\Lambda$ are 0 and the even ones are given by $\kappa_{2n}^\infty = B_{2n}/(2n!)$, where $B_{2n}$ denotes the $2n$-th Bernoulli number. For details, see the derivation of (A.10) in Appendix A. For (c)-(d), the even moments of $\Lambda$ are given by $m_{2k}^\infty = 1/(k+1)$ and, from these, we numerically compute the rectangular free cumulants via (A.12) in Appendix A. Furthermore, the spectral threshold for the setting in (a) is $\alpha_s = 1 + \sqrt{c}$ ; for (b), $\alpha_s = 0$ ; and for (c)-(d), $\tilde{\alpha}_s = 0$. In (a), we set $n = 8000$ and $c = 2$; in (b), we set $n = 4000$; and in (c)-(d), we set $n = 8000$ and $\gamma = 1/2$. The signal $\boldsymbol{u}^*$ has a Rademacher prior, i.e., its entries are i.i.d. and uniform in $\{-1, 1\}$. In the rectangular case, the signal $\boldsymbol{v}^*$ has a Gaussian prior, i.e., it is uniformly distributed on the sphere of radius $\sqrt{n}$. Given these priors, $\mathsf{u}_t$ is chosen to be the single-iterate posterior mean denoiser given by $\mathsf{u}_t(x) = \tanh(\mu_t x/\sigma_{t,t})$, where $\mu_t$ and $\sigma_{t,t}$ are the state evolution parameters; these are replaced by consistent estimates in the simulations. For the rectangular case, we choose $\mathsf{v}_t(x) = x$. Each experiment is repeated for $n_{\mathrm{trials}} = 100$ independent runs. We report the average and error bars at 1 standard deviation.

Figure 1 compares the performance between the proposed AMP algorithm with PCA initialization (PCA+AMP) and the theoretical predictions of state evolution (SE), for two different values of $\alpha$. On the $x$-axis, we have the number of iterations of AMP, and on the $y$-axis the normalized squared correlation between the iterate and the signal. As a reference, we also plot the performance of PCA as a horizontal line. We observe an excellent agreement of AMP with state evolution, even in the settings (b)-(c)-(d) where the free cumulants (resp. rectangular free cumulants) are alternating in sign. This supports our conjecture that Theorems 1-2 hold for more general noise distributions.

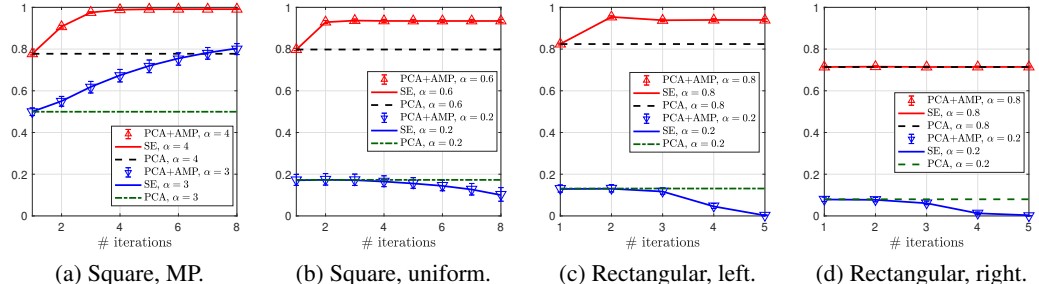

(a) Square, MP.     (b) Square, uniform.     (c) Rectangular, left.     (d) Rectangular, right.

Figure 1: Comparison between AMP with PCA initialization and the related state evolution (SE). The plots show the normalized squared correlation between iterate and signal, as a function of the number of iterations.

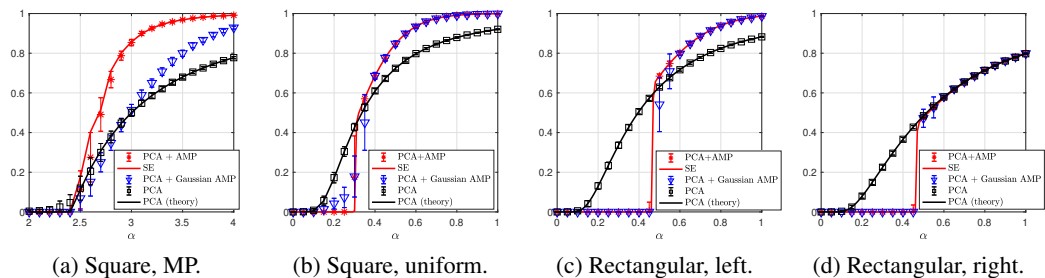

(a) Square, MP.     (b) Square, uniform.     (c) Rectangular, left.     (d) Rectangular, right.

Figure 2: Comparison between AMP with PCA initialization and the PCA method alone. The plots show the normalized squared correlation between the signal and the estimate (PCA, or AMP+PCA), as a function of $\alpha$.

In Figure 2, we run PCA+AMP until the algorithm converges, and we compare the results with *(i)* the AMP with PCA initialization developed in [44] which assumes that the noise matrix is Gaussian (with the correct variance), and *(ii)* the PCA method alone, as a function of the SNR $\alpha$. For Marchenko-Pastur noise (setting (a)), PCA+AMP always improves upon the PCA initialization. However, this is not the case when the eigenvalues/singular values of the noise are uniformly distributed (settings (b), (c) and (d)). In fact, we observe a phase transition phenomenon: below a certain critical $\alpha$, AMP converges to a trivial fixed point at $0$, while PCA shows positive correlation with the signal; above the critical $\alpha$, PCA+AMP is no worse than PCA. This is due to the sub-optimal choice of $\mathsf{u}_t$; recall the discussion on p.6. We observe no improvement for the estimation of the right singular vector (setting (d)), as the prior of $\boldsymbol{v}^*$ is Gaussian, in which case we expect the PCA estimate to be optimal. The interesting behavior demonstrated in Figure 2 motivates the study of the optimal choice for $\mathsf{u}_t, \mathsf{v}_t$ in future work. We also note that, in settings (c)-(d), $\tilde{\alpha}_{\mathrm{s}} = 0$, which means that the PCA estimator has non-zero correlation with the signal for all $\alpha > 0$. However, for $\alpha < 0.1$, this correlation remains rather small. Finally, we highlight that our proposed rotationally invariant PCA+AMP always improves upon the Gaussian PCA+AMP. In general, this performance gap will be significant unless the sequence of free cumulants $\kappa_k^\infty$ ($\kappa_{2k}^\infty$ in the rectangular case) decays quickly. For Marchenko-Pastur noise, the free cumulants are all equal, and thus the performance gap is significant. If the eigenvalues/singular values of the noise are uniform, then the sequence of free cumulants decays rapidly and the performance gap is small.

## 5 Proof Sketch: Symmetric Square Matrices

We consider the following artificial AMP algorithm, whose iterates are denoted by $\tilde{\boldsymbol{u}}^t, \tilde{\boldsymbol{f}}^t$ for $t \geq 1$. We initialize with $\tilde{\boldsymbol{u}}^1 = \rho_\alpha \boldsymbol{u}^* + \sqrt{1 - \rho_\alpha^2}\, \boldsymbol{n}$ and $\tilde{\boldsymbol{f}}^1 = \boldsymbol{X}\tilde{\boldsymbol{u}}^1 - \kappa_1 \tilde{\boldsymbol{u}}^1$. Here, $\boldsymbol{n}$ is standard Gaussian and $\rho_\alpha$ is the normalized (limit) correlation of the PCA estimate given in (2.2). We note that this initialization is impractical, as it requires the knowledge of the unknown signal $\boldsymbol{u}^*$. However, this is not an issue since the artificial AMP serves only as a proof technique. (The *true AMP* (3.2) used for estimation uses the PCA initialization in (3.1).) The subsequent iterates of the artificial AMP are defined in two phases. In the first phase, which lasts up to iteration $(T + 1)$, the functions defining

the artificial AMP are chosen so that $\tilde{\boldsymbol{u}}^{T+1}$ is closely aligned with the eigenvector $\boldsymbol{u}_{\mathrm{PCA}}$ as $T \to \infty$. In the second phase, the functions are chosen to match those in the true AMP.

The artificial AMP initialization $\tilde{\boldsymbol{u}}^1$ is chosen such that it has non-zero asymptotic correlation with the signal $\boldsymbol{u}^*$. Indeed, when the signal prior has zero mean, a random initialization (independent of $\boldsymbol{u}^*$) would be asymptotically uncorrelated with the signal; consequently, the first phase of the artificial AMP would get stuck at a trivial fixed point and the iterates would not be guaranteed to converge to the principal eigenvector. We ensure that this does not happen by defining the initialization $\tilde{\boldsymbol{u}}^1$ to be a linear combination of the signal and Gaussian noise.

**First phase.** For $2 \le t \le (T+1)$, the artificial AMP iterates are

$$\tilde{\boldsymbol{u}}^t = \tilde{\boldsymbol{f}}^{t-1}/\alpha, \qquad \tilde{\boldsymbol{f}}^t = \boldsymbol{X}\tilde{\boldsymbol{u}}^t - \sum_{i=1}^{t} \tilde{\mathsf{b}}_{t,i}\tilde{\boldsymbol{u}}^i, \tag{5.1}$$

where $\tilde{\mathsf{b}}_{t,t-j} = \kappa_{j+1}\alpha^{-j}$, for $(t-j) \in [1,t]$. We claim that, for sufficiently large $T$, $\tilde{\boldsymbol{u}}^{T+1}$ approaches the PCA estimate $\boldsymbol{u}_{\mathrm{PCA}}$, that is, $\lim_{T\to\infty} \lim_{n\to\infty} \frac{1}{\sqrt{n}}\|\tilde{\boldsymbol{u}}^{T+1} - \sqrt{n}\boldsymbol{u}_{\mathrm{PCA}}\| = 0$. This result is proved in Lemma B.7 in Appendix B.3. We give a heuristic sanity check here. Assume that the iterate $\tilde{\boldsymbol{u}}^{T+1}$ converges to a limit $\tilde{\boldsymbol{u}}^\infty$ in the sense that $\lim_{T\to\infty} \lim_{n\to\infty} \frac{1}{\sqrt{n}}\|\tilde{\boldsymbol{u}}^{T+1} - \tilde{\boldsymbol{u}}^\infty\| = 0$. Then, from (5.1), the limit $\tilde{\boldsymbol{u}}^\infty$ satisfies

$$\tilde{\boldsymbol{u}}^\infty = \frac{1}{\alpha}\boldsymbol{X}\tilde{\boldsymbol{u}}^\infty - \sum_{i=1}^{\infty} \kappa_i \left(\frac{1}{\alpha}\right)^i \tilde{\boldsymbol{u}}^\infty \iff \left(\alpha + \sum_{i=1}^{\infty} \kappa_i \left(\frac{1}{\alpha}\right)^{i-1}\right)\tilde{\boldsymbol{u}}^\infty = \boldsymbol{X}\tilde{\boldsymbol{u}}^\infty, \tag{5.2}$$

which means that $\tilde{\boldsymbol{u}}^\infty$ is an eigenvector of $\boldsymbol{X}$. Furthermore, by using known identities in free probability (see (A.4) and (A.6)), the eigenvalue $\alpha + \sum_{i=1}^{\infty} \kappa_i \left(\frac{1}{\alpha}\right)^{i-1}$ can be re-written as $G^{-1}(1/\alpha)$. Recall that, for $\alpha > \alpha_\mathrm{s}$, $\boldsymbol{X}$ exhibits a spectral gap and its largest eigenvalue converges to $G^{-1}(1/\alpha)$. Thus, $\boldsymbol{u}^\infty$ must be aligned with the principal eigenvector of $\boldsymbol{X}$, as desired.

A key step in our analysis is to show that, as $T \to \infty$, the state evolution of the artificial AMP in the first phase has the unique fixed point $(\tilde{\mu} = \alpha\rho_\alpha, \tilde{\sigma} = \alpha^2(1-\rho_\alpha^2))$. This is established in Lemma B.2 proved in Appendix B.2. The proof follows the approach developed in Section 7 of [20]. However, the analysis of [20] requires that $\alpha$ is sufficiently large, while our result holds for all $\alpha > \alpha_\mathrm{s}$. Our idea is to exploit the expression of the limit correlation between the PCA estimate and the signal. In particular, we prove that, when the PCA estimate is correlated with the signal, state evolution is close to a limit map which is a contraction. The price to pay for this approach is the requirement that the free cumulants are non-negative.

**Second phase.** The second phase of the artificial AMP is designed so that its iterates $(\tilde{\boldsymbol{u}}^{T+k}, \tilde{\boldsymbol{f}}^{T+k})$ are close to $(\boldsymbol{u}^k, \boldsymbol{f}^k)$, for $k \ge 2$. For $t \ge (T+2)$, the artificial AMP computes:

$$\tilde{\boldsymbol{u}}^t = \mathsf{u}_{t-T}(\tilde{\boldsymbol{f}}^{t-1}), \qquad \tilde{\boldsymbol{f}}^t = \boldsymbol{X}\tilde{\boldsymbol{u}}^t - \sum_{i=1}^{t} \tilde{\mathsf{b}}_{t,i}\tilde{\boldsymbol{u}}^i. \tag{5.3}$$

Here, the functions $\{u_k\}_{k\ge 2}$, are the ones used in the true AMP (3.2). The coefficients $\{\tilde{\mathsf{b}}_{t,i}\}$ for $t \ge (T+2)$ are given by:

$$\tilde{\mathsf{b}}_{tt} = \kappa_1, \quad \tilde{\mathsf{b}}_{t,t-j} = \kappa_{j+1}\left(\frac{1}{\alpha}\right)^{(T+1-(t-j))_+} \prod_{i=\max\{t-j+1,T+2\}}^{t} \langle \mathsf{u}'_{i-T}(\tilde{\boldsymbol{f}}^{i-1})\rangle, \quad (t-j) \in [1,t-1].$$
$$\tag{5.4}$$

Since the artificial AMP is initialized with $\tilde{\boldsymbol{u}}^1$ that is correlated with $\boldsymbol{u}^*$ and independent of the noise matrix $\boldsymbol{W}$, a state evolution result for it can be obtained directly from [20, Theorem 1.1]. We then show in Lemma B.8 in Appendix B.4 that the second phase iterates in (5.3) are close to the true AMP iterates in (3.2), and that their state evolution parameters are also close. This result yields Theorem 1, as shown in Appendix B.5. The complete proof of Theorem 2 (rectangular case) is given in Appendix C. We describe the artificial AMP for this case along with a proof sketch in Appendix C.1.

## Acknowledgements

M. Mondelli would like to thank László Erdös for helpful discussions. M. Mondelli was partially supported by the 2019 Lopez-Loreta Prize. R. Venkataramanan was partially supported by the Alan Turing Institute under the EPSRC grant EP/N510129/1.

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
