# Supplementary Material (Appendix)

# PCA Initialization for Approximate Message Passing in Rotationally Invariant Models

## A  Free Probability Background

### A.1  Symmetric Square Matrices

Let $X$ be a random variable of finite moments of all orders, and denote its moments by $m_k = \mathbb{E}\{X^k\}$. In this paper, $X$ represents either the empirical eigenvalue distribution of the noise matrix $\boldsymbol{W} \in \mathbb{R}^{n \times n}$, or its limit law $\Lambda$ (in the latter case, the moments and free cumulants are denoted by $\{m_k^\infty\}_{k \geq 1}$ and $\{\kappa_k^\infty\}_{k \geq 1}$, respectively). For the model (1.1), note that the empirical eigenvalue distribution of $\boldsymbol{W}$ coincides with the empirical eigenvalue distribution of $\boldsymbol{X}$ after excluding the largest eigenvalue of $\boldsymbol{X}$, since we consider the case $\alpha > \alpha_{\mathrm{s}}$. The free cumulants $\{\kappa_k\}_{k \geq 1}$ of $X$ are defined recursively by the moment-cumulant relations

$$m_k = \sum_{\pi \in \mathrm{NC}(k)} \prod_{S \in \pi} \kappa_{|S|}, \tag{A.1}$$

where $\mathrm{NC}(k)$ is the set of all non-crossing partitions of $\{1, \dots, k\}$, and $|S|$ denotes the cardinality of $S$. Furthermore, by exploiting the connection between the formal power series with coefficients $\{m_k\}_{k \geq 1}$ and $\{\kappa_k\}_{k \geq 1}$, each free cumulant $\kappa_k$ can be computed from $m_1, \dots, m_k$ and $\kappa_1, \dots, \kappa_{k-1}$ as [47, Section 2.5]

$$\kappa_k = m_k - [z^k] \sum_{j=1}^{k-1} \kappa_j \left( z + m_1 z^2 + m_2 z^3 + \dots + m_{k-1} z^k \right)^j, \tag{A.2}$$

where $[z^k](q(z))$ denotes the coefficient of $z^k$ in the polynomial $q(z)$.

Consider now the random variable $\Lambda$ representing the limiting spectral distribution of $\boldsymbol{W}$, and recall that $b < \infty$ denotes the supremum of the support of $\Lambda$. Then, for $z > b$, the Cauchy transform $G(z)$ of $\Lambda$ is given by

$$G(z) = \mathbb{E} \left\{ \frac{1}{z - \Lambda} \right\}. \tag{A.3}$$

Another transform that will be useful in our analysis is the $R$-transform $R(z)$ of $\Lambda$, which can be defined by the convergent series:

$$R(z) = \sum_{i=0}^{\infty} \kappa_{i+1}^\infty z^i, \tag{A.4}$$

where $\{\kappa_k^\infty\}_{k \geq 1}$ are the free cumulants of $\Lambda$. The derivative of the $R$-transform is denoted by $R'(z)$ and given by

$$R'(z) = \sum_{i=0}^{\infty} (i+1) \kappa_{i+2}^\infty z^i = \sum_{j=0}^{\infty} \sum_{k=0}^{\infty} \kappa_{j+k+2}^\infty z^{j+k}, \tag{A.5}$$

where the second equality follows from a double-counting argument. The series in (A.4) and (A.5) are well-defined and converge to a finite value for $z < 1/\alpha_{\mathrm{s}}$, where $\alpha_{\mathrm{s}} = 1/G(b^+)$ is the spectral threshold [10]. The $R$-transform can also be expressed in terms of the Cauchy transform, see e.g. Theorem 12.7 of [46]:

$$R(z) = G^{-1}(z) - \frac{1}{z}. \tag{A.6}$$

By taking the derivative on both sides of (A.5), we have

$$R'(z) = \frac{1}{G'(G^{-1}(z))} + \frac{1}{z^2}. \tag{A.7}$$

If $\boldsymbol{W}$ follows a Marcenko-Pastur distribution (i.e., $\boldsymbol{W} = \frac{1}{n} \boldsymbol{G}_n \boldsymbol{G}_n^\mathsf{T} \in \mathbb{R}^{n \times n}$, where the entries of $\boldsymbol{G}_n \in \mathbb{R}^{n \times p}$ are i.i.d. standard Gaussian), then it is well known that $\kappa_k^\infty = c \triangleq p/n$ for $k \geq 1$, see

e.g. [40, Chap. 2, Exercise 11]. This corresponds to the setting (a) in the numerical results of Section 4. If the eigenvalues of $W$ are i.i.d. and uniformly distributed in the interval $[-1/2, 1/2]$, the free cumulants $\kappa_k^\infty$ have also a simple form. In fact, by explicitly computing the expectation in (A.3), we have that

$$G(z) = \log \frac{2z + 1}{2z - 1}. \tag{A.8}$$

Thus, by applying (A.6), we deduce that

$$R(z) = \frac{1}{2} \coth\left(\frac{z}{2}\right) - \frac{1}{z}. \tag{A.9}$$

By comparing the series expansion (A.4) with that of the hyperbolic cotangent, we conclude that

$$\kappa_k = \begin{cases} 0, & \text{if } k \text{ is odd,} \\ \dfrac{B_k}{k!}, & \text{if } k \text{ is even,} \end{cases} \tag{A.10}$$

where $B_k$ denotes the $k$-th Bernoulli number. This corresponds to the setting (b) in the numerical results of Section 4.

## A.2 Rectangular Matrices

Let $X$ be a random variable of finite moments of all orders, and denote its even moments by $m_{2k} = \mathbb{E}\{X^{2k}\}$. In this paper, $X^2$ represents either the empirical eigenvalue distribution of $WW^\top \in \mathbb{R}^{m \times m}$, or its limit law $\Lambda^2$ (in the latter case, the moments and rectangular free cumulants are denoted by $\{m_{2k}^\infty\}_{k\geq 1}$ and $\{\kappa_{2k}^\infty\}_{k\geq 1}$, respectively). For the model (1.2), note that the empirical eigenvalue distribution of $WW^\top$ coincides with the empirical eigenvalue distribution of $XX^\top$ after excluding the largest eigenvalue of $XX^\top$, since we consider the case $\tilde{\alpha} > \tilde{\alpha}_s$. The rectangular free cumulants $\{\kappa_{2k}\}_{k\geq 1}$ of $X$ are defined recursively by the moment-cumulant relations [9, Section 3]

$$m_{2k} = \gamma \sum_{\substack{\pi \in \mathrm{NC}'(2k)}} \prod_{\substack{S \in \pi \\ \min S \text{ is odd}}} \kappa_{|S|} \prod_{\substack{S \in \pi \\ \min S \text{ is even}}} \kappa_{|S|}, \tag{A.11}$$

where $\mathrm{NC}'(2k)$ is the set of non-crossing partitions $\pi$ of $\{1, \ldots, 2k\}$ such that each set $S \in \pi$ has even cardinality. Furthermore, by exploiting the connection between the formal power series with coefficients $\{m_{2k}\}_{k\geq 1}$ and $\{\kappa_{2k}\}_{k\geq 1}$, each rectangular free cumulant $\kappa_{2k}$ can be computed from $m_2, \ldots, m_{2k}$ and $\kappa_2, \ldots, \kappa_{2(k-1)}$ as [9, Lemma 3.4]

$$\kappa_{2k} = m_{2k} - [z^k] \sum_{j=1}^{k-1} \kappa_{2j} \left(z(\gamma M(z) + 1)(M(z) + 1)\right)^j, \tag{A.12}$$

where $M(z) = \sum_{k=1}^\infty m_{2k} z^k$ and $[z^k](q(z))$ denotes again the coefficient of $z^k$ in the polynomial $q(z)$.

Consider now the random variable $\Lambda$ representing the limiting distribution of the singular values of $W$, and recall that $b < \infty$ denotes the supremum of the support of $\Lambda$. Then, for $z > b$, the $D$-transform $D(z)$ of $\Lambda$ is given by

$$D(z) = \phi(z) \cdot \bar{\phi}(z), \tag{A.13}$$

where

$$\phi(z) = \mathbb{E}\left\{\frac{z}{z^2 - \Lambda^2}\right\}, \quad \bar{\phi}(z) = \gamma\phi(z) + \frac{1 - \gamma}{z}. \tag{A.14}$$

Another transform that will be useful in our analysis is the rectangular $R$-transform $R(z)$ of $\Lambda$, which can be defined by the convergent series:

$$R(z) = \sum_{i=1}^\infty \kappa_{2i}^\infty z^i, \tag{A.15}$$

where $\{\kappa_{2k}^\infty\}_{k\geq 1}$ are the rectangular free cumulants of $\Lambda$. The derivative of the rectangular $R$-transform is denoted by $R'(z)$ and given by

$$R'(z) = \sum_{i=0}^{\infty}(i+1)\kappa_{2(i+1)}^\infty z^i = \sum_{j=0}^{\infty}\sum_{k=0}^{\infty}\kappa_{2(j+k+1)}^\infty z^{j+k}, \tag{A.16}$$

where the second equality follows from a double-counting argument. By combining (A.15) and (A.16), we also obtain the useful identities

$$\sum_{j=0}^{\infty}\sum_{k=0}^{\infty}\kappa_{2(j+k+2)}z^{j+k+2} = zR'(z) - R(z), \tag{A.17}$$

$$\sum_{i=0}^{\infty}(i+1)\kappa_{2(i+2)}^\infty z^i = z^{-1}R'(z) - z^{-2}R(z). \tag{A.18}$$

The series in (A.15)-(A.18) are well-defined and converge to a finite value for $z < 1/(\tilde{\alpha}_s)^2$, where $\tilde{\alpha}_s = 1/\sqrt{D(b^+)}$ is the spectral threshold [11]. The rectangular $R$-transform can also be expressed in terms of the $D$-transform, see e.g. [11, Section 2.5]:

$$\gamma R^2(z) + (\gamma + 1)R(z) + 1 = z(D^{-1}(z))^2. \tag{A.19}$$

# B    Proof of Theorem 1

This appendix is organized as follows. In Appendix B.1, we present the state evolution recursion associated to the artificial AMP iteration defined in (5.1) and (5.3). In Appendix B.2, we prove that the first phase of this state evolution admits a unique fixed point. Using this fact, in Appendix B.3, we prove that the artificial AMP iterate at the end of the first phase approaches the PCA estimator. Then, in Appendix B.4, we show that *(i)* the iterates in the second phase of the artificial AMP are close to the true AMP iterates, and *(ii)* the related state evolution parameters also remain close. Finally, in Appendix B.5, we give the proof of Theorem 1.

## B.1    State Evolution for the Artificial AMP

Consider the artificial AMP iteration defined in (5.1) and (5.3), with initialization

$$\tilde{u}^1 = \rho_\alpha u^* + \sqrt{1-\rho_\alpha^2}\,n, \qquad \tilde{f}^1 = X\tilde{u}^1 - \kappa_1\tilde{u}^1. \tag{B.1}$$

Then, its associated state evolution recursion is expressed in terms of a sequence of mean vectors $\tilde{\mu}_K = (\tilde{\mu}_t)_{t\in[0,K]}$ and covariance matrices $\tilde{\Sigma}_K = (\tilde{\sigma}_{s,t})_{s,t\in[0,K]}$ defined recursively as follows. We initialize with

$$\tilde{\mu}_0 = \alpha\rho_\alpha, \qquad \tilde{\sigma}_{0,0} = \alpha^2(1-\rho_\alpha^2), \quad \tilde{\sigma}_{0,t} = \tilde{\sigma}_{t,0} = 0, \quad \text{for } t \geq 1. \tag{B.2}$$

Given $\tilde{\mu}_K$ and $\tilde{\Sigma}_K$, let

$$(\tilde{F}_0, \ldots, \tilde{F}_K) = \tilde{\mu}_K U_* + (\tilde{Z}_0, \ldots, \tilde{Z}_K), \quad \text{where } (\tilde{Z}_0, \ldots, \tilde{Z}_K) \sim \mathcal{N}(0, \tilde{\Sigma}_K), \quad \text{and}$$

$$\tilde{U}_t = \tilde{u}_t(\tilde{F}_{t-1}), \quad \text{where } \tilde{u}_t(x) = \begin{cases} x/\alpha, & 1 \leq t \leq T+1, \\ u_{t-T}(x), & t \geq T+2. \end{cases} \tag{B.3}$$

Then, the entries of $\tilde{\mu}_{K+1}$ are given by $\tilde{\mu}_t = \alpha\mathbb{E}\{\tilde{U}_t U_*\}$ (for $t \in [1, K+1]$), and the entries of $\tilde{\Sigma}_{K+1}$ (for $s,t \in [1, K+1]$) are given by

$$\tilde{\sigma}_{s,t} = \sum_{j=0}^{s-1}\sum_{k=0}^{t-1}\kappa_{j+k+2}^\infty\left(\prod_{i=s-j+1}^{s}\mathbb{E}\{\tilde{u}_i'(\tilde{F}_{i-1})\}\right)\left(\prod_{i=t-k+1}^{t}\mathbb{E}\{\tilde{u}_i'(\tilde{F}_{i-1})\}\right)\mathbb{E}\{\tilde{U}_{s-j}\tilde{U}_{t-k}\}. \tag{B.4}$$

**Proposition B.1** (State evolution for artificial AMP – symmetric square matrices)**.** *Consider the setting of Theorem 1, the artificial AMP iteration described in* (5.1) *and* (5.3) *with the initialization given in* (B.1)*, and the corresponding state evolution parameters defined in* (B.2)-(B.4)*. Then, for $t \geq 1$ and any* PL(2) *function $\psi : \mathbb{R}^{2t+2} \to \mathbb{R}$, the following holds almost surely:*

$$\lim_{n\to\infty}\frac{1}{n}\sum_{i=1}^{n}\psi(u_i^*, \tilde{u}_i^1, \ldots, \tilde{u}_i^{t+1}, \tilde{f}_i^1, \ldots\tilde{f}_i^t) = \mathbb{E}\left\{\psi(U_*, \tilde{U}_1, \ldots, \tilde{U}_{t+1}, \tilde{F}_1, \ldots, \tilde{F}_t)\right\}. \tag{B.5}$$

The proposition follows directly from Theorem 1.1 in [20] since the initialization $\tilde{\boldsymbol{u}}^1$ of the artificial AMP is independent of $\boldsymbol{W}$.

## B.2 Fixed Point of State Evolution for the First Phase

From (B.2)-(B.4), we note that the state evolution recursion for the first phase $(t \in [1, T+1])$ has the following form:

$$
\begin{aligned}
\tilde{\mu}_t &= \alpha \rho_\alpha, \quad \text{for } t \in [1, T+1], \\
\tilde{\sigma}_{s,t} &= \sum_{j=0}^{s-1} \sum_{k=0}^{t-1} \kappa_{j+k+2}^\infty \left(\frac{1}{\alpha}\right)^{j+k+2} \left((\alpha \rho_\alpha)^2 + \tilde{\sigma}_{s-j-1,t-k-1}\right), \quad \text{for } s, t \in [1, T+1].
\end{aligned}
\tag{B.6}
$$

In this section, we prove the following result concerning the fixed point of the recursion (B.6).

**Lemma B.2** (Fixed point of state evolution for first phase – Square matrices). *Consider the state evolution recursion for the first phase given by* (B.6), *initialized according to* (B.2). *Assume that* $\kappa_i^\infty \geq 0$ *for all* $i \geq 2$, *and that* $\alpha > \alpha_{\mathrm{s}}$. *Pick any* $\xi < 1$ *such that* $\alpha \xi > \alpha_{\mathrm{s}}$. *Then,*

$$
\lim_{T \to \infty} \max_{s,t \in [0,T]} \xi^{\max(s,t)} |\tilde{\sigma}_{T+1-s, T+1-t} - \alpha^2 (1 - \rho_\alpha^2)| = 0.
\tag{B.7}
$$

To prove the claim, we consider the space of infinite matrices $\boldsymbol{x} = (x_{s,t} : s, t \leq 0)$ indexed by the non-positive integers and equipped with the weighted $\ell_\infty$-norm:

$$
\|\boldsymbol{x}\|_\xi = \sup_{s,t \leq 0} \xi^{\max(|s|,|t|)} |x_{s,t}|.
\tag{B.8}
$$

We define $\mathcal{X} = \{\boldsymbol{x} : \|\boldsymbol{x}\|_\xi < \infty\}$, and note that $\mathcal{X}$ is complete under $\|\cdot\|_\xi$. For any compact set $I \subset \mathbb{R}$, we also define

$$
\mathcal{X}_I = \{\boldsymbol{x} : x_{s,t} \in I \text{ for all } s, t \leq 0\} \subset \mathcal{X}.
\tag{B.9}
$$

Then, $\mathcal{X}_I$ is closed in $\mathcal{X}$ and therefore it is also complete under $\|\cdot\|_\xi$. We embed the matrix $\tilde{\boldsymbol{\Sigma}}_{\bar{T}}$ as an element $\boldsymbol{x} \in \mathcal{X}$ with the following coordinate identification:

$$
\begin{aligned}
\tilde{\sigma}_{s,t} &= x_{s-\bar{T}, t-\bar{T}}, \\
x_{s,t} &= 0, \quad \text{if } s < -\bar{T} \text{ or } t < -\bar{T}.
\end{aligned}
$$

The idea is to approximate the map $\tilde{\boldsymbol{\Sigma}}_{\bar{T}-1} \mapsto \tilde{\boldsymbol{\Sigma}}_{\bar{T}}$ with the *limit* map $h^\Sigma$ defined as

$$
h_{s,t}^\Sigma(\boldsymbol{x}) = \sum_{j=0}^\infty \sum_{k=0}^\infty \kappa_{j+k+2}^\infty \left(\frac{1}{\alpha}\right)^{j+k+2} \left((\alpha \rho_\alpha)^2 + x_{s-j,t-k}\right).
\tag{B.10}
$$

The map $h^\Sigma$ has a similar structure to the embedding of the map $\tilde{\boldsymbol{\Sigma}}_{\bar{T}-1} \mapsto \tilde{\boldsymbol{\Sigma}}_{\bar{T}}$ into $\mathcal{X}$. However, comparing (B.6) and (B.10), we highlight two important differences. First, the indices of $x_{s-j,t-k}$ are shifted with respect to the indices of $\tilde{\sigma}_{s-j-1,t-k-1}$. This difference is purely technical and it simplifies the proof of the subsequent Lemma B.6, which shows that $h^\Sigma$ is close to the map $\tilde{\boldsymbol{\Sigma}}_{\bar{T}-1} \mapsto \tilde{\boldsymbol{\Sigma}}_{\bar{T}}$. Second, the map $h^\Sigma$ is *fixed*, in the sense that it does not depend on $s, t$. In fact, note that the sums over $j$ and $k$ run from 0 to $\infty$ in (B.10). This is in contrast with (B.6) where the two sums run until $j = s - 1$ and $k = t - 1$.

The approach of approximating the state evolution map with a fixed limit map was first developed in [20]. The key difference is that, in [20], it is assumed that $\alpha$ is sufficiently large, which allows to simplify the analysis. On the contrary, our result holds for all $\alpha > \alpha_{\mathrm{s}}$, $\alpha_{\mathrm{s}}$ being the spectral threshold for PCA. This is because of two main reasons. First, the expressions for the state evolution recursion are simplified by considering linear denoisers in the first phase of the artificial AMP. Second, we crucially exploit the form (and the strict positivity) of the correlation between the signal and the PCA estimate, in order to prove that the limit map (B.10) is a contraction (cf. (B.14) in Lemma B.5).

First, we show that $h^\Sigma(\mathcal{X}_{I^*}) \subseteq \mathcal{X}_{I^*}$ for a suitably defined compact set $I^*$.

**Lemma B.3** (Image of limit map – Square matrices). *Consider the map* $h^\Sigma$ *defined in* (B.10). *Assume that* $\kappa_i^\infty \geq 0$ *for all* $i \geq 2$, *and that* $\alpha > \alpha_{\mathrm{s}}$. *Then, there exists* $I^* = [-a^*, a^*]$ *such that, if* $\boldsymbol{x} \in \mathcal{X}_{I^*}$, *then* $h^\Sigma(\boldsymbol{x}) \in \mathcal{X}_{I^*}$.

*Proof.* Let $\boldsymbol{x} \in \mathcal{X}_{I^*}$. Then, the following chain of inequalities holds:

$$|h_{s,t}^{\Sigma}(\boldsymbol{x})| \overset{(a)}{=} \left| \rho_\alpha^2 R'\left(\frac{1}{\alpha}\right) + \sum_{j=0}^{\infty} \sum_{k=0}^{\infty} \kappa_{j+k+2}^{\infty} \left(\frac{1}{\alpha}\right)^{j+k+2} x_{s-j,t-k} \right|$$

$$\overset{(b)}{\leq} \rho_\alpha^2 \left| R'\left(\frac{1}{\alpha}\right) \right| + \sum_{j=0}^{\infty} \sum_{k=0}^{\infty} \kappa_{j+k+2}^{\infty} \left(\frac{1}{\alpha}\right)^{j+k+2} |x_{s-j,t-k}|$$

$$\overset{(c)}{\leq} \rho_\alpha^2 \left| R'\left(\frac{1}{\alpha}\right) \right| + a^* R'\left(\frac{1}{\alpha}\right)\left(\frac{1}{\alpha}\right)^2.$$

Here, (a) follows from (B.10) and (A.5); (b) follows from the hypothesis that $\kappa_i^{\infty} \geq 0$ for $i \geq 2$; and (c) uses again (A.5) and the fact that $\boldsymbol{x} \in \mathcal{X}_{I^*}$.

Now, recall from (2.2) that above the spectral threshold, namely, when $\alpha > \alpha_{\mathrm{s}}$, the PCA estimator $\boldsymbol{u}_{\mathrm{PCA}}$ has strictly positive correlation with the signal $\boldsymbol{u}^*$:

$$\frac{\langle \boldsymbol{u}_{\mathrm{PCA}}, \boldsymbol{u}^* \rangle^2}{n} \overset{\mathrm{a.s.}}{\longrightarrow} \rho_\alpha^2 = \frac{-1}{\alpha^2 G'(G^{-1}(1/\alpha))},$$

which immediately implies that

$$\frac{1}{\alpha^2 G'(G^{-1}\left(\frac{1}{\alpha}\right))} < 0. \tag{B.11}$$

Thus, by combining (B.11) with (A.7), we deduce that

$$R'\left(\frac{1}{\alpha}\right)\left(\frac{1}{\alpha}\right)^2 < 1. \tag{B.12}$$

Hence, as $R'\left(\frac{1}{\alpha}\right) < \infty$, there exists an $a^*$ such that

$$\rho_\alpha^2 \left| R'\left(\frac{1}{\alpha}\right) \right| + a^* R'\left(\frac{1}{\alpha}\right)\left(\frac{1}{\alpha}\right)^2 \leq a^*,$$

which implies the desired claim. $\qquad\square$

Next, we compute a fixed point of $h^{\Sigma}$.

**Lemma B.4** (Fixed point of limit map – Square matrices). *Consider the map $h^{\Sigma}$ defined in (B.10), and let $\boldsymbol{x}^* = (x_{s,t}^* : s, t \leq 0)$ with $x_{s,t}^* = \alpha^2(1 - \rho_\alpha^2)$. Assume that $\alpha > \alpha_{\mathrm{s}}$. Then, $\boldsymbol{x}^*$ is a fixed point of $h^{\Sigma}$.*

*Proof.* Note that, for $x = 1/\alpha$, the power series expansion (A.5) of $R'$ converges to a finite limit as $\alpha > \alpha_{\mathrm{s}}$. Hence, by using the definition (B.10), we have that

$$h_{s,t}^{\Sigma}(\boldsymbol{x}^*) = R'\left(\frac{1}{\alpha}\right).$$

Then, the claim follows from (A.7) and the definition $\rho_\alpha = \sqrt{\frac{-1}{\alpha^2 G'(G^{-1}(1/\alpha))}}$, which together show that $R'\left(\frac{1}{\alpha}\right) = \alpha^2(1 - \rho_\alpha^2)$. $\qquad\square$

Let $I^*$ be such that $h^{\Sigma} : \mathcal{X}_{I^*} \to \mathcal{X}_{I^*}$ (the existence of such a set $I^*$ is guaranteed by Lemma B.3). Then, the next step is to show that $h^{\Sigma} : \mathcal{X}_{I^*} \to \mathcal{X}_{I^*}$ is a contraction. We remark that, by the Banach fixed point theorem, this result implies that the fixed point $\boldsymbol{x}^*$ defined in Lemma B.4 is unique.

**Lemma B.5** (Limit map is a contraction). *Consider the map $h^{\Sigma} : \mathcal{X}_{I^*} \to \mathcal{X}_{I^*}$ defined in (B.10) and where $I^*$ is given by Lemma B.3. Assume that $\kappa_i^{\infty} \geq 0$ for all $i \geq 2$, and let $\xi < 1$ be such that $\alpha\xi > \alpha_{\mathrm{s}}$. Then, for any $\boldsymbol{x}, \boldsymbol{y} \in \mathcal{X}_{I^*}$,*

$$\|h^{\Sigma}(\boldsymbol{x}) - h^{\Sigma}(\boldsymbol{y})\|_\xi \leq R'\left(\frac{1}{\xi\alpha}\right)\left(\frac{1}{\xi\alpha}\right)^2 \|\boldsymbol{x} - \boldsymbol{y}\|_\xi, \tag{B.13}$$

*where*

$$R'\left(\frac{1}{\xi\alpha}\right)\left(\frac{1}{\xi\alpha}\right)^2 < 1. \tag{B.14}$$

*Proof.* First of all, for any $s, t \le 0$, we have that

$$
|h_{s,t}^{\Sigma}(\boldsymbol{x}) - h_{s,t}^{\Sigma}(\boldsymbol{y})| \overset{(a)}{=} \left| \sum_{j=0}^{\infty} \sum_{k=0}^{\infty} \kappa_{j+k+2}^{\infty} \left(\frac{1}{\alpha}\right)^{j+k+2} (x_{s-j,t-k} - y_{s-j,t-k}) \right|
$$

$$
\overset{(b)}{\le} \sum_{j=0}^{\infty} \sum_{k=0}^{\infty} \kappa_{j+k+2}^{\infty} \left(\frac{1}{\alpha}\right)^{j+k+2} |x_{s-j,t-k} - y_{s-j,t-k}|.
$$

(B.15)

Here, (a) follows from (B.10), and (b) follows from the hypothesis that $\kappa_i^{\infty} \ge 0$ for $i \ge 2$. Furthermore, we have that

$$
|x_{s-j,t-k} - y_{s-j,t-k}| \le \|\boldsymbol{x} - \boldsymbol{y}\|_{\xi} \xi^{-\max(|s-j|,|t-k|)}.
$$

(B.16)

Thus, by using (B.15) and (B.16), we obtain

$$
\|h^{\Sigma}(\boldsymbol{x}) - h^{\Sigma}(\boldsymbol{y})\|_{\xi} = \sup_{s,t \le 0} \xi^{\max(|s|,|t|)} |h_{s,t}^{\Sigma}(\boldsymbol{x}) - h_{s,t}^{\Sigma}(\boldsymbol{y})|
$$

$$
\le \sup_{s,t \le 0} \xi^{\max(|s|,|t|)} \|\boldsymbol{x} - \boldsymbol{y}\|_{\xi} \sum_{j=0}^{\infty} \sum_{k=0}^{\infty} \kappa_{j+k+2}^{\infty} \left(\frac{1}{\alpha}\right)^{j+k+2} \xi^{-\max(|s-j|,|t-k|)}.
$$

(B.17)

Note that, as $\xi < 1$,

$$
\xi^{-\max(|s-j|,|t-k|)} \le \xi^{-\max(|s|,|t|)-j-k-2},
$$

which implies that the RHS of (B.17) is bounded above by

$$
\|\boldsymbol{x} - \boldsymbol{y}\|_{\xi} \sum_{j=0}^{\infty} \sum_{k=0}^{\infty} \kappa_{j+k+2}^{\infty} \left(\frac{1}{\xi\alpha}\right)^{j+k+2} = R' \left(\frac{1}{\xi\alpha}\right) \left(\frac{1}{\xi\alpha}\right)^2 \|\boldsymbol{x} - \boldsymbol{y}\|_{\xi},
$$

(B.18)

where the equality follows from (A.5). This shows that (B.13) holds. The proof of (B.14) follows the same argument as (B.12), since $\xi\alpha > \alpha_{\mathrm{s}}$. $\qquad \square$

At this point, we show that the state evolution of $\tilde{\boldsymbol{\Sigma}}_{\bar{T}}$ can be approximated via the fixed map $h^{\Sigma}$.

**Lemma B.6** (Limit map approximates state evolution map – Square matrices). *Consider the map $h^{\Sigma} : \mathcal{X}_{I^*} \to \mathcal{X}_{I^*}$ defined in (B.10), where $I^*$ is given by Lemma B.3. Assume that $\kappa_i^{\infty} \ge 0$ for all $i \ge 2$, and let $\xi < 1$ be such that $\alpha\xi > \alpha_{\mathrm{s}}$. Then, for any $\boldsymbol{x} \in \mathcal{X}_{I^*}$,*

$$
\|\tilde{\boldsymbol{\Sigma}}_{\bar{T}} - h^{\Sigma}(\boldsymbol{x})\|_{\xi} \le R' \left(\frac{1}{\xi\alpha}\right) \left(\frac{1}{\xi\alpha}\right)^2 \|\tilde{\boldsymbol{\Sigma}}_{\bar{T}-1} - \boldsymbol{x}\|_{\xi} + F(\bar{T}),
$$

(B.19)

*where*

$$
\lim_{\bar{T} \to \infty} F(\bar{T}) = 0.
$$

(B.20)

*Proof.* Throughout the proof, we consider $\tilde{\boldsymbol{\Sigma}}_{\bar{T}}, \tilde{\boldsymbol{\Sigma}}_{\bar{T}-1}$ as embedded in $\mathcal{X}$. First, we write

$$
\|\tilde{\boldsymbol{\Sigma}}_{\bar{T}} - h^{\Sigma}(\boldsymbol{x})\|_{\xi} = \sup_{s,t \le 0} \xi^{\max(|s|,|t|)} |(\tilde{\boldsymbol{\Sigma}}_{\bar{T}})_{s,t} - h_{s,t}^{\Sigma}(\boldsymbol{x})|
$$

$$
= \max \left( \sup_{\substack{s,t \le 0 \\ \max(|s|,|t|) < \bar{T}}} \xi^{\max(|s|,|t|)} |(\tilde{\boldsymbol{\Sigma}}_{\bar{T}})_{s,t} - h_{s,t}^{\Sigma}(\boldsymbol{x})|, \right.
$$

$$
\left. \sup_{\substack{s,t \le 0 \\ \max(|s|,|t|) \ge \bar{T}}} \xi^{\max(|s|,|t|)} |(\tilde{\boldsymbol{\Sigma}}_{\bar{T}})_{s,t} - h_{s,t}^{\Sigma}(\boldsymbol{x})| \right),
$$

(B.21)

where $(\tilde{\boldsymbol{\Sigma}}_{\bar{T}})_{s,t} = \tilde{\sigma}_{s+\bar{T},t+\bar{T}}$ if $s \ge -\bar{T}$ and $t \ge -\bar{T}$, and $(\tilde{\boldsymbol{\Sigma}}_{\bar{T}})_{s,t} = 0$ otherwise.

Let us look at the case $\max(|s|,|t|) < \bar{T}$, and define $I_1 = \{(j,k) : j \geq s + \bar{T} \text{ or } k \geq t + \bar{T}\}$. Then,

$$
\begin{aligned}
|(\tilde{\boldsymbol{\Sigma}}_{\bar{T}})_{s,t} - h_{s,t}^{\Sigma}(\boldsymbol{x})| = \Bigg| & \sum_{j=0}^{s+\bar{T}-1} \sum_{k=0}^{t+\bar{T}-1} \kappa_{j+k+2}^{\infty} \left(\frac{1}{\alpha}\right)^{j+k+2} \left(\alpha^2 \rho_{\alpha}^2 + \tilde{\sigma}_{s-j+\bar{T}-1,t-k+\bar{T}-1}\right) \\
& - \sum_{j=0}^{\infty} \sum_{k=0}^{\infty} \kappa_{j+k+2}^{\infty} \left(\frac{1}{\alpha}\right)^{j+k+2} \left(\alpha^2 \rho_{\alpha}^2 + x_{s-j,t-k}\right) \Bigg| \\
\leq \Bigg| & \sum_{j=0}^{s+\bar{T}-1} \sum_{k=0}^{t+\bar{T}-1} \kappa_{j+k+2}^{\infty} \left(\frac{1}{\alpha}\right)^{j+k+2} \left(\tilde{\sigma}_{s-j+\bar{T}-1,t-k+\bar{T}-1} - x_{s-j,t-k}\right) \Bigg| \\
& + \Bigg| \sum_{j,k\in I_1} \kappa_{j+k+2}^{\infty} \left(\frac{1}{\alpha}\right)^{j+k+2} \left(\alpha^2 \rho_{\alpha}^2 + x_{s-j,t-k}\right) \Bigg| := T_1 + T_2.
\end{aligned}
$$

(B.22)

The term $T_1$ can be upper bounded as follows:

$$
\begin{aligned}
T_1 & \overset{(a)}{\leq} \sum_{j=0}^{s+\bar{T}-1} \sum_{k=0}^{t+\bar{T}-1} \kappa_{j+k+2}^{\infty} \left(\frac{1}{\alpha}\right)^{j+k+2} \left|\tilde{\sigma}_{s-j+\bar{T}-1,t-k+\bar{T}-1} - x_{s-j,t-k}\right| \\
& \leq \sum_{j=0}^{s+\bar{T}-1} \sum_{k=0}^{t+\bar{T}-1} \kappa_{j+k+2}^{\infty} \left(\frac{1}{\alpha}\right)^{j+k+2} \|\tilde{\boldsymbol{\Sigma}}_{\bar{T}-1} - \boldsymbol{x}\|_{\xi} \xi^{-\max(|s-j|,|t-k|)} \\
& \overset{(b)}{\leq} \sum_{j=0}^{s+\bar{T}-1} \sum_{k=0}^{t+\bar{T}-1} \kappa_{j+k+2}^{\infty} \left(\frac{1}{\xi\alpha}\right)^{j+k+2} \|\tilde{\boldsymbol{\Sigma}}_{\bar{T}-1} - \boldsymbol{x}\|_{\xi} \xi^{-\max(|s|,|t|)} \\
& \overset{(c)}{\leq} \sum_{j=0}^{\infty} \sum_{k=0}^{\infty} \kappa_{j+k+2}^{\infty} \left(\frac{1}{\xi\alpha}\right)^{j+k+2} \|\tilde{\boldsymbol{\Sigma}}_{\bar{T}-1} - \boldsymbol{x}\|_{\xi} \xi^{-\max(|s|,|t|)} \\
& \overset{(d)}{=} R' \left(\frac{1}{\xi\alpha}\right) \left(\frac{1}{\xi\alpha}\right)^2 \|\tilde{\boldsymbol{\Sigma}}_{\bar{T}-1} - \boldsymbol{x}\|_{\xi} \xi^{-\max(|s|,|t|)}.
\end{aligned}
$$

(B.23)

Here, (a) and (c) follows from the hypothesis that $\kappa_i^{\infty} \geq 0$ for $i \geq 2$; (b) uses that $\xi < 1$; and (d) uses (A.5). The term $T_2$ can be upper bounded as follows:

$$
\begin{aligned}
T_2 & \leq \left(\alpha^2 \rho_{\alpha}^2 + a^*\right) \sum_{j,k\in I_1} \kappa_{j+k+2}^{\infty} \left(\frac{1}{\alpha}\right)^{j+k+2} \\
& \leq \frac{\alpha^2 \rho_{\alpha}^2 + a^*}{\alpha^2} \sum_{i=-\max(|s|,|t|)+\bar{T}}^{\infty} \kappa_{i+2}^{\infty}(i+1) \left(\frac{1}{\alpha}\right)^i,
\end{aligned}
$$

(B.24)

where the first inequality uses that $\boldsymbol{x} \in \mathcal{X}_{I^*}$ and the second inequality uses that, if $(j,k) \in I_1$, then $j + k \geq -\max(|s|,|t|) + \bar{T}$. By combining (B.22), (B.23) and (B.24), we obtain that

$$
\begin{aligned}
& \sup_{\substack{s,t\leq 0 \\ \max(|s|,|t|)<\bar{T}}} \xi^{\max(|s|,|t|)} |(\tilde{\boldsymbol{\Sigma}}_{\bar{T}})_{s,t} - h_{s,t}^{\Sigma}(\boldsymbol{x})| \\
& \leq R' \left(\frac{1}{\xi\alpha}\right) \left(\frac{1}{\xi\alpha}\right)^2 \|\tilde{\boldsymbol{\Sigma}}_{\bar{T}-1} - \boldsymbol{x}\|_{\xi} + \frac{\alpha^2 \rho_{\alpha}^2 + a^*}{\alpha^2} \sup_{0\leq t\leq \bar{T}} \xi^t \sum_{i=\bar{T}-t}^{\infty} \kappa_{i+2}^{\infty}(i+1) \left(\frac{1}{\alpha}\right)^i.
\end{aligned}
$$

(B.25)

Let us now look at the case $\max(|s|,|t|) \geq \bar{T}$. Recall that $|h_{s,t}^{\Sigma}(\boldsymbol{x})| \leq a^*$, $\tilde{\sigma}_{0,0} = (1 - \rho_{\alpha}^2)\alpha^2$ and $\tilde{\sigma}_{0,t} = 0$ for $t \in [1, \bar{T}]$. Thus,

$$
|(\tilde{\boldsymbol{\Sigma}}_{\bar{T}})_{s,t} - h_{s,t}^{\Sigma}(\boldsymbol{x})| \leq c_1,
$$

where $c_1$ is a constant independent of $s, t, \bar{T}$. This immediately implies that

$$\sup_{\substack{s,t\leq 0 \\ \max(|s|,|t|)\geq \bar{T}}} \xi^{\max(|s|,|t|)} |(\tilde{\boldsymbol{\Sigma}}_{\bar{T}})_{s,t} - h_{s,t}^{\Sigma}(\boldsymbol{x})| \leq c_1 \xi^{\bar{T}},$$

which combined with (B.25) allows us to conclude that

$$
\begin{aligned}
\|\tilde{\boldsymbol{\Sigma}}_{\bar{T}} - h^{\Sigma}(\boldsymbol{x})\|_{\xi} \leq{}& R'\left(\frac{1}{\xi\alpha}\right)\left(\frac{1}{\xi\alpha}\right)^2 \|\tilde{\boldsymbol{\Sigma}}_{\bar{T}-1} - \boldsymbol{x}\|_{\xi} \\
&+ \frac{\alpha^2\rho_\alpha^2 + a^*}{\alpha^2} \sup_{0\leq t\leq \bar{T}} \xi^t \sum_{i=\bar{T}-t}^{\infty} \kappa_{i+2}^{\infty}(i+1)\left(\frac{1}{\alpha}\right)^i + c_1\xi^{\bar{T}}.
\end{aligned}
\tag{B.26}
$$

As $\alpha > \alpha_{\mathrm{s}}$ and the series in (A.5) is convergent for $z < 1/\alpha_{\mathrm{s}}$, one readily verifies that

$$\lim_{\bar{T}\to\infty} \sup_{0\leq t\leq \bar{T}} \xi^t \sum_{i=\bar{T}-t}^{\infty} \kappa_{i+2}^{\infty}(i+1)\left(\frac{1}{\alpha}\right)^i = 0, \tag{B.27}$$

which concludes the proof. $\qquad\square$

Finally, we can put everything together and prove Lemma B.2.

*Proof of Lemma B.2.* Fix $\epsilon > 0$ and denote by $\left(h^{\Sigma}\right)^{T_0}$ the $T_0$-fold composition of $h^{\Sigma}$. Recall from Lemmas B.4 and B.5 that $\boldsymbol{x}^*$ is the unique fixed point of $h^{\Sigma} : X_{I^*} \to X_{I^*}$. Then, for any $\boldsymbol{x} \in \mathcal{X}_{I^*}$,

$$\| \left(h^{\Sigma}\right)^{T_0}(\boldsymbol{x}) - \boldsymbol{x}^*\|_{\xi} = \| \left(h^{\Sigma}\right)^{T_0}(\boldsymbol{x}) - \left(h^{\Sigma}\right)^{T_0}(\boldsymbol{x}^*)\|_{\xi} \leq \left(R'\left(\frac{1}{\xi\alpha}\right)\left(\frac{1}{\xi\alpha}\right)^2\right)^{T_0} \|\boldsymbol{x} - \boldsymbol{x}^*\|_{\xi}, \tag{B.28}$$

where the inequality follows from Lemma B.5. Note that $R'\left(\frac{1}{\xi\alpha}\right)\left(\frac{1}{\xi\alpha}\right)^2 < 1$ (see (B.14)) and that $\boldsymbol{x}, \boldsymbol{x}^* \in \mathcal{X}_{I^*}$. Thus, we can make the RHS of (B.28) smaller than $\epsilon/2$ by choosing a sufficiently large $T_0$. Furthermore, an application of Lemma B.6 gives that, for all sufficiently large $\bar{T}$,

$$\|\boldsymbol{\Sigma}_{\bar{T}+T_0} - \left(h^{\Sigma}\right)^{T_0}(\boldsymbol{x})\|_{\xi} \leq \left(R'\left(\frac{1}{\xi\alpha}\right)\left(\frac{1}{\xi\alpha}\right)^2\right)^{T_0} \|\boldsymbol{\Sigma}_{\bar{T}} - \boldsymbol{x}\|_{\xi} + \frac{\epsilon}{4}. \tag{B.29}$$

Note that $\boldsymbol{x} \in \mathcal{X}_{I^*}$ implies that $\|\boldsymbol{x}\|_{\xi} \leq a^*$. In addition, by following the same argument as in Lemma B.3, one can show that $|\tilde{\sigma}_{s,t}| \leq a^*$ for all $s, t$, which in turn implies that $\|\boldsymbol{\Sigma}_{\bar{T}}\|_{\xi} \leq a^*$. As a result, we can make the RHS of (B.29) is smaller than $\epsilon/2$ by choosing sufficiently large $T_0$. As the RHS of both (B.28) and (B.29) can be made smaller than $\epsilon/2$, an application of the triangle inequality gives that

$$\limsup_{\bar{T}\to\infty} \|\boldsymbol{\Sigma}_{\bar{T}} - \boldsymbol{x}^*\|_{\xi} \leq \epsilon, \tag{B.30}$$

which, after setting $\bar{T} = T + 1$, implies the desired result. $\qquad\square$

## B.3 Convergence to PCA Estimator for the First Phase

In this section, we prove that the artificial AMP iterate at the end of the first phase converges to the PCA estimator in normalized $\ell_2$-norm.

**Lemma B.7** (Convergence to PCA estimator – Square matrices). *Consider the setting of Theorem 1, and the first phase of the artificial AMP iteration described in* (5.1)*, with the initialization given in* (B.1)*. Assume that $\kappa_i^{\infty} \geq 0$ for all $i \geq 2$, and that $\alpha > \alpha_{\mathrm{s}}$. Then,*

$$\lim_{T\to\infty} \lim_{n\to\infty} \frac{1}{\sqrt{n}} \|\tilde{\boldsymbol{u}}^{T+1} - \sqrt{n}\boldsymbol{u}_{\mathrm{PCA}}\| = 0 \;\; \text{almost surely.} \tag{B.31}$$

*Proof.* Consider the following decomposition of $\tilde{\boldsymbol{u}}^{T+1}$:

$$\tilde{\boldsymbol{u}}^{T+1} = \zeta_{T+1}\boldsymbol{u}_{\mathrm{PCA}} + \boldsymbol{r}^{T+1}, \tag{B.32}$$

where $\zeta_{T+1} = \langle \tilde{\boldsymbol{u}}^{T+1}, \boldsymbol{u}_{\mathrm{PCA}} \rangle$ and $\langle \boldsymbol{r}^{T+1}, \boldsymbol{u}_{\mathrm{PCA}} \rangle = 0$. Define

$$\boldsymbol{e}^{T+1} = \left( \boldsymbol{X} - G^{-1}\left(\frac{1}{\alpha}\right)\boldsymbol{I}_n \right)\tilde{\boldsymbol{u}}^{T+1}, \tag{B.33}$$

where $G^{-1}$ is the inverse of the Cauchy transform of $\Lambda$. Then, using (B.32), (B.33) can be rewritten as

$$\left( \boldsymbol{X} - G^{-1}\left(\frac{1}{\alpha}\right)\boldsymbol{I}_n \right)\boldsymbol{r}^{T+1} = \boldsymbol{e}^{T+1} - \left( \boldsymbol{X} - G^{-1}\left(\frac{1}{\alpha}\right)\boldsymbol{I}_n \right)\zeta_{T+1}\boldsymbol{u}_{\mathrm{PCA}}. \tag{B.34}$$

First, we will show that

$$\left\| \left( \boldsymbol{X} - G^{-1}\left(\frac{1}{\alpha}\right)\boldsymbol{I}_n \right)\boldsymbol{r}^{T+1} \right\| \geq c\|\boldsymbol{r}^{T+1}\|, \tag{B.35}$$

where $c > 0$ is a constant (independent of $n, T$). We start by observing that the matrix $\boldsymbol{X} - G^{-1}\left(\frac{1}{\alpha}\right)\boldsymbol{I}_n$ is symmetric, hence it can be written in the form $\boldsymbol{Q}\tilde{\boldsymbol{\Lambda}}\boldsymbol{Q}^\mathsf{T}$, with $\boldsymbol{Q}$ orthogonal and $\tilde{\boldsymbol{\Lambda}}$ diagonal. Furthermore, the columns of $\boldsymbol{Q}$ are the eigenvectors of $\boldsymbol{X} - G^{-1}\left(\frac{1}{\alpha}\right)\boldsymbol{I}_n$ and the diagonal entries of $\tilde{\boldsymbol{\Lambda}}$ are the corresponding eigenvalues. As $\boldsymbol{r}^{T+1}$ is orthogonal to $\boldsymbol{u}_{\mathrm{PCA}}$, we can write

$$\left( \boldsymbol{X} - G^{-1}\left(\frac{1}{\alpha}\right)\boldsymbol{I}_n \right)\boldsymbol{r}^{T+1} = \boldsymbol{Q}\tilde{\boldsymbol{\Lambda}}'\boldsymbol{Q}^\mathsf{T}\boldsymbol{r}^{T+1}, \tag{B.36}$$

where $\tilde{\boldsymbol{\Lambda}}'$ is obtained from $\tilde{\boldsymbol{\Lambda}}$ by changing the entry corresponding to $\lambda_1(\boldsymbol{X}) - G^{-1}\left(\frac{1}{\alpha}\right)$ to any other value. For our purposes, it suffices to substitute $\lambda_1(\boldsymbol{X}) - G^{-1}\left(\frac{1}{\alpha}\right)$ with $\lambda_2(\boldsymbol{X}) - G^{-1}\left(\frac{1}{\alpha}\right)$. Note that

$$\begin{aligned}
\|\boldsymbol{Q}\tilde{\boldsymbol{\Lambda}}'\boldsymbol{Q}^\mathsf{T}\boldsymbol{r}^{T+1}\|^2 &\geq \|\boldsymbol{r}^{T+1}\|^2 \min_{\boldsymbol{s}:\|\boldsymbol{s}\|=1} \|\boldsymbol{Q}\tilde{\boldsymbol{\Lambda}}'\boldsymbol{Q}^\mathsf{T}\boldsymbol{s}\|^2 \\
&= \|\boldsymbol{r}^{T+1}\|^2 \min_{\boldsymbol{s}:\|\boldsymbol{s}\|=1} \langle \boldsymbol{s}, \boldsymbol{Q}\left(\tilde{\boldsymbol{\Lambda}}'\right)^2\boldsymbol{Q}^\mathsf{T}\boldsymbol{s} \rangle \\
&= \|\boldsymbol{r}^{T+1}\|^2 \lambda_{\min}(\boldsymbol{Q}\left(\tilde{\boldsymbol{\Lambda}}'\right)^2\boldsymbol{Q}^\mathsf{T}),
\end{aligned} \tag{B.37}$$

where $\lambda_{\min}(\boldsymbol{Q}\left(\tilde{\boldsymbol{\Lambda}}'\right)^2\boldsymbol{Q}^\mathsf{T})$ denotes the smallest eigenvalue of $\boldsymbol{Q}\left(\tilde{\boldsymbol{\Lambda}}'\right)^2\boldsymbol{Q}^\mathsf{T}$ and the last equality follows from the variational characterization of the smallest eigenvalue of a symmetric matrix. Note that

$$\lambda_{\min}(\boldsymbol{Q}\left(\tilde{\boldsymbol{\Lambda}}'\right)^2\boldsymbol{Q}^\mathsf{T}) = \lambda_{\min}\left((\tilde{\boldsymbol{\Lambda}}')^2\right) = \min_{i\in\{2,\ldots,n\}}\left(\left(G^{-1}\left(\frac{1}{\alpha}\right) - \lambda_i(\boldsymbol{X})\right)^2\right). \tag{B.38}$$

Recall that, for $\alpha > \alpha_\mathrm{s}$, $\lambda_1(\boldsymbol{X}) \xrightarrow{\text{a.s.}} G^{-1}(1/\alpha)$ and $\lambda_2(\boldsymbol{X}) \xrightarrow{\text{a.s.}} b < G^{-1}(1/\alpha)$, see [10, Theorem 2.1]. Thus, the RHS of (B.38) is lower bounded by a constant independent of $n, T$. By combining this result with (B.36) and (B.37), we deduce that (B.35) holds.

Next, we prove that a.s.

$$\lim_{T\to\infty}\lim_{n\to\infty}\frac{1}{\sqrt{n}}\left\|\boldsymbol{e}^{T+1} - \left( \boldsymbol{X} - G^{-1}\left(\frac{1}{\alpha}\right)\boldsymbol{I}_n \right)\zeta_{T+1}\boldsymbol{u}_{\mathrm{PCA}}\right\| = 0. \tag{B.39}$$

An application of the triangle inequality gives that

$$\left\|\boldsymbol{e}^{T+1} - \left( \boldsymbol{X} - G^{-1}\left(\frac{1}{\alpha}\right)\boldsymbol{I}_n \right)\zeta_{T+1}\boldsymbol{u}_{\mathrm{PCA}}\right\| \leq \|\boldsymbol{e}^{T+1}\| + \left\|\left( \boldsymbol{X} - G^{-1}\left(\frac{1}{\alpha}\right)\boldsymbol{I}_n \right)\zeta_{T+1}\boldsymbol{u}_{\mathrm{PCA}}\right\|. \tag{B.40}$$

The second term on the RHS of (B.40) is equal to

$$|\zeta_{T+1}|\left|\lambda_1(\boldsymbol{X}) - G^{-1}\left(\frac{1}{\alpha}\right)\right|. \tag{B.41}$$

By using Theorem 2.1 of [10], we have that, for $\alpha > \alpha_s$, almost surely,

$$\lim_{n \to \infty} \left| \lambda_1(\boldsymbol{X}) - G^{-1}\left(\frac{1}{\alpha}\right) \right| = 0. \tag{B.42}$$

Furthermore,

$$\frac{1}{\sqrt{n}} |\zeta_{T+1}| \leq \frac{1}{\sqrt{n}} \|\tilde{\boldsymbol{u}}^{T+1}\| = \frac{1}{\alpha\sqrt{n}} \|\tilde{\boldsymbol{f}}^T\|.$$

By Proposition B.1, we have that

$$\lim_{n \to \infty} \frac{1}{\alpha\sqrt{n}} \|\tilde{\boldsymbol{f}}^T\| = \frac{1}{\alpha} \sqrt{\tilde{\mu}_T^2 + \tilde{\sigma}_{T,T}},$$

which, for sufficiently large $T$, is upper bounded by a constant independent of $n, T$, as $\tilde{\mu}_T = \alpha\rho_\alpha$ and $\tilde{\sigma}_{T,T}$ converges to $\alpha^2(1 - \rho_\alpha^2)$ as $T \to \infty$ by Lemma B.2. By combining this result with (B.42), we deduce that

$$\lim_{T \to \infty} \lim_{n \to \infty} \frac{1}{\sqrt{n}} \left\| \left( \boldsymbol{X} - G^{-1}\left(\frac{1}{\alpha}\right) \boldsymbol{I}_n \right) \zeta_{T+1} \boldsymbol{u}_{\mathrm{PCA}} \right\| = 0. \tag{B.43}$$

To bound the first term on the RHS of (B.40), we proceed as follows:

$$\begin{aligned} \lim_{n \to \infty} \frac{1}{n} \|\boldsymbol{e}^{T+1}\|^2 &= \lim_{n \to \infty} \frac{1}{n} \left\| \left( \boldsymbol{X} - G^{-1}\left(\frac{1}{\alpha}\right) \boldsymbol{I}_n \right) \tilde{\boldsymbol{u}}^{T+1} \right\|^2 \\ &\overset{(a)}{=} \lim_{n \to \infty} \frac{1}{n} \left\| \tilde{\boldsymbol{f}}^{T+1} + \sum_{i=1}^{T+1} \kappa_{T-i+2} \left(\frac{1}{\alpha}\right)^{T-i+1} \tilde{\boldsymbol{u}}^i - G^{-1}\left(\frac{1}{\alpha}\right) \tilde{\boldsymbol{u}}^{T+1} \right\|^2 \\ &\overset{(b)}{=} \lim_{n \to \infty} \frac{1}{n} \left\| \tilde{\boldsymbol{f}}^{T+1} + \sum_{i=1}^{T+1} \kappa_{T-i+2}^\infty \left(\frac{1}{\alpha}\right)^{T-i+1} \tilde{\boldsymbol{u}}^i - G^{-1}\left(\frac{1}{\alpha}\right) \tilde{\boldsymbol{u}}^{T+1} \right\|^2 \\ &\overset{(c)}{=} \mathbb{E} \left\{ \left( \tilde{F}_{T+1} + \sum_{i=1}^{T+1} \kappa_{T-i+2}^\infty \left(\frac{1}{\alpha}\right)^{T-i+1} \tilde{U}_i - G^{-1}\left(\frac{1}{\alpha}\right) \tilde{U}_{T+1} \right)^2 \right\}. \end{aligned} \tag{B.44}$$

Here, (a) uses the iteration (5.1) of the first phase of the artificial AMP, and (c) follows from Proposition B.1, where $\tilde{U}_t$ for $t \in [1, T+1]$ and $\tilde{F}_{T+1}$ are defined in (B.3). To obtain (b), we write

$$\begin{aligned} &\lim_{n \to \infty} \frac{1}{n} \left\| \sum_{i=1}^{T+1} (\kappa_{T-i+2} - \kappa_{T-i+2}^\infty) \left(\frac{1}{\alpha}\right)^{T-i+1} \tilde{\boldsymbol{u}}^i \right\|^2 \\ &= \lim_{n \to \infty} \sum_{i,j=1}^{T+1} (\kappa_{T-i+2} - \kappa_{T-i+2}^\infty)(\kappa_{T-j+2} - \kappa_{T-j+2}^\infty) \left(\frac{1}{\alpha}\right)^{2T-i-j+2} \frac{\langle \tilde{\boldsymbol{u}}^i, \tilde{\boldsymbol{u}}^j \rangle}{n}. \end{aligned} \tag{B.45}$$

Using the state evolution result of Proposition B.1 and (B.3), we almost surely have

$$\lim_{n \to \infty} \frac{\langle \tilde{\boldsymbol{u}}^i, \tilde{\boldsymbol{u}}^j \rangle}{n} = \frac{1}{\alpha^2}(\alpha^2\rho_\alpha^2 + \tilde{\sigma}_{i,j}) < 1, \tag{B.46}$$

where the last inequality uses $\tilde{\sigma}_{i,j} < \tilde{\sigma}_{0,0} = \alpha^2(1 - \rho_\alpha^2)$. (This can be deduced from the recursion (B.6) using the formula (2.2) for $\rho_\alpha^2$, and the relations (A.5) and (A.7).) Therefore, since $\kappa_i \overset{n \to \infty}{\longrightarrow} \kappa_i^\infty$ for $i \in [1, T+1]$ (by the model assumptions), we almost surely have that (b) holds.

Next, by the triangle inequality, (B.44) is upper bounded by

$$\begin{aligned} 3 \cdot \mathbb{E} &\left\{ \left( \alpha - G^{-1}\left(\frac{1}{\alpha}\right) + \sum_{i=1}^{T+1} \kappa_{T-i+2}^\infty \left(\frac{1}{\alpha}\right)^{T-i+1} \right)^2 \tilde{U}_{T+1}^2 \right\} \\ &+ 3 \cdot \mathbb{E} \left\{ \left( \sum_{i=1}^{T+1} \kappa_{T-i+2}^\infty \left(\frac{1}{\alpha}\right)^{T-i+1} (\tilde{U}_i - \tilde{U}_{T+1}) \right)^2 \right\} \\ &+ 3 \cdot \mathbb{E} \left\{ (\tilde{F}_{T+1} - \alpha\tilde{U}_{T+1})^2 \right\} := S_1 + S_2 + S_3. \end{aligned} \tag{B.47}$$

The term $S_3$ can be expressed as

$$S_3 = 3 \cdot \mathbb{E}\left\{(\tilde{F}_{T+1} - \tilde{F}_T)^2\right\} = 3(\tilde{\sigma}_{T+1,T+1} - 2\tilde{\sigma}_{T+1,T} + \tilde{\sigma}_{T,T}).$$

Thus, by Lemma B.2, we have that

$$\lim_{T\to\infty} S_3 = 0. \qquad (B.48)$$

The term $S_1$ can be expressed as

$$S_1 = 3 \cdot \left(\alpha - G^{-1}\left(\frac{1}{\alpha}\right) + \sum_{i=1}^{T+1} \kappa_{T-i+2}^\infty \left(\frac{1}{\alpha}\right)^{T-i+1}\right)^2 \frac{\alpha^2 \rho_\alpha^2 + \tilde{\sigma}_{T,T}}{\alpha^2}.$$

By Lemma B.2, we have $\lim_{T\to\infty} \tilde{\sigma}_{T,T} = \alpha^2(1 - \rho_{\alpha^2})$, and hence

$$\lim_{T\to\infty} S_1 = 3 \cdot \left(\alpha - G^{-1}\left(\frac{1}{\alpha}\right) + \sum_{i=0}^{\infty} \kappa_{i+1}^\infty \left(\frac{1}{\alpha}\right)^i\right)^2 = 0, \qquad (B.49)$$

where the last equality follows from (A.4) and (A.6). Finally, consider the term $S_2$, which after expanding the square and some manipulations, can be expressed as

$$S_2 = \frac{3}{\alpha^2} \sum_{i,j=0}^{T} \kappa_{i+1}^\infty \kappa_{j+1}^\infty \left(\frac{1}{\alpha}\right)^{i+j} (\tilde{\sigma}_{T-j,T-i} + \tilde{\sigma}_{T,T} - \tilde{\sigma}_{T,T-i} - \tilde{\sigma}_{T,T-j}). \qquad (B.50)$$

The expression above can be bounded above as

$$S_2 \leq \frac{3}{\alpha^2} \sum_{i,j=0}^{T} \kappa_{i+1}^\infty \kappa_{j+1}^\infty \left(\frac{1}{\alpha}\right)^{i+j} (|\tilde{\sigma}_{T-j,T-i} - \alpha^2(1-\rho_\alpha^2)| + |\tilde{\sigma}_{T,T} - \alpha^2(1-\rho_\alpha^2)| \qquad (B.51)$$

$$+ |\tilde{\sigma}_{T,T-i} - \alpha^2(1-\rho_\alpha^2)| + |\tilde{\sigma}_{T,T-j} - \alpha^2(1-\rho_\alpha^2)|).$$

We now apply Lemma B.2 to bound each of the four absolute values on the RHS of (B.51). Fix any $\xi \in (\frac{\alpha_s}{\alpha}, 1)$. Then, by Lemma B.2, for any $\epsilon > 0$ there exists $T^*(\epsilon)$ such that for $T > T^*(\epsilon)$, we have

$$S_2 \leq \epsilon \cdot \frac{3}{\alpha^2} \sum_{i,j=0}^{T} \kappa_{i+1}^\infty \kappa_{j+1}^\infty \left(\frac{1}{\alpha}\right)^{i+j} \cdot \left(\xi^{-\max(i,j)} + 1 + \xi^{-i} + \xi^{-j}\right)$$

$$\overset{(a)}{\leq} \epsilon \cdot \frac{12}{\alpha^2} \sum_{i,j=0}^{T} \kappa_{i+1}^\infty \kappa_{j+1}^\infty \left(\frac{1}{\xi\alpha}\right)^{i+j} \qquad (B.52)$$

$$\overset{(b)}{\leq} \epsilon \cdot \frac{12}{\alpha^2} \sum_{i,j=0}^{\infty} \kappa_{i+1}^\infty \kappa_{j+1}^\infty \left(\frac{1}{\xi\alpha}\right)^{i+j}$$

$$\overset{(c)}{\leq} \epsilon \cdot \frac{12}{\alpha^2} \left(R\left(\frac{1}{\xi\alpha}\right)\right)^2.$$

Here, (a) uses that $\xi < 1$, (b) uses that $\kappa_i^\infty \geq 0$ for $i \geq 2$, and (c) uses the power series expansion (A.4) of $R(\cdot)$, which converges to a finite limit as $\xi\alpha > \alpha_s$. Since $\epsilon$ can be arbitrarily small, we have

$$\lim_{T\to\infty} S_2 = 0. \qquad (B.53)$$

By combining (B.44), (B.47), (B.48), (B.49) and (B.53), we have that

$$\lim_{T\to\infty} \lim_{n\to\infty} \frac{1}{\sqrt{n}} \left\|e^{T+1}\right\| = 0, \qquad (B.54)$$

which, combined with (B.43), gives (B.39). Finally, by using (B.35) and (B.39), we have that

$$\lim_{T\to\infty} \lim_{n\to\infty} \frac{1}{\sqrt{n}} \left\|r^{T+1}\right\| = 0. \qquad (B.55)$$

Thus, from the decomposition (B.32), we conclude that, as $n \to \infty$ and $T \to \infty$, $\tilde{u}^{T+1}$ is aligned with $u_{\mathrm{PCA}}$. Furthermore, from another application of Proposition B.1, we obtain

$$\lim_{T\to\infty} \lim_{n\to\infty} \frac{1}{\sqrt{n}} \|\tilde{u}^{T+1}\| = \lim_{T\to\infty} \frac{1}{\alpha} \sqrt{\tilde{\mu}_T^2 + \tilde{\sigma}_{T,T}} = 1, \qquad (B.56)$$

which implies that $\lim_{T\to\infty} \lim_{n\to\infty} \zeta_{T+1} = 1$ and concludes the proof. $\qquad \square$

## B.4 Analysis for the Second Phase

We first define a modified version of the true AMP algorithm, in which the memory coefficients $\{b_{t,i}\}_{i\in[1,t]}$ in (3.1)-(3.2) are replaced by deterministic values obtained from state evolution. The iterates of the modified AMP, denoted by $\hat{u}^t$, are given by:

$$\hat{u}^1 = \sqrt{n}u_{\mathrm{PCA}}, \quad \hat{f}^1 = X\hat{u}^1 - \bar{b}_{1,1}\hat{u}^1, \tag{B.57}$$

$$\hat{u}^t = u_t(\hat{f}^{t-1}), \quad \hat{f}^t = X\hat{u}^t - \sum_{i=1}^{t}\bar{b}_{t,i}\hat{u}^i, \qquad t \geq 2, \tag{B.58}$$

where

$$\bar{b}_{1,1} = \sum_{i=0}^{\infty}\kappa_{i+1}^{\infty}\alpha^{-i},$$

$$\bar{b}_{t,t} = \kappa_1^{\infty}, \quad \bar{b}_{t,1} = \sum_{i=0}^{\infty}\kappa_{i+t}^{\infty}\alpha^{-i}\prod_{\ell=2}^{t}\mathbb{E}\{u_{\ell}'(F_{\ell-1})\}, \tag{B.59}$$

$$\bar{b}_{t,t-j} = \kappa_{j+1}^{\infty}\prod_{i=t-j+1}^{t}\mathbb{E}\{u_i'(F_{i-1})\}, \text{ for } (t-j) \in [2, t-1].$$

We recall that $\{\kappa_i^{\infty}\}$ are the free cumulants of the limiting spectral distribution $\Lambda$, and the random variables $\{F_i\}$ are given by (3.4).

The following lemma shows that, as $T$ grows, the iterates of the second phase of the artificial AMP approach those of the modified AMP algorithm above, as do the corresponding state evolution parameters.

**Lemma B.8.** *Consider the setting of Theorem 1. Assume that $\kappa_i^{\infty} \geq 0$ for all $i \geq 2$, and that $\alpha > \alpha_s$. Consider the modified version of the true AMP in* (B.57)-(B.58)*, and the artificial AMP in* (5.1)-(5.3) *along with its state evolution recursion given by* (B.2)-(B.4)*. Then, the following results hold for $s, t \geq 1$:*

*1.*

$$\lim_{T\to\infty}\tilde{\mu}_{T+t} = \mu_t, \qquad \lim_{T\to\infty}\tilde{\sigma}_{T+s,T+t} = \sigma_{s,t}. \tag{B.60}$$

*2. For any* $\mathrm{PL}(2)$ *function* $\psi : \mathbb{R}^{2t+2} \to \mathbb{R}$*, we have*

$$\lim_{T\to\infty}\lim_{n\to\infty}\left|\frac{1}{n}\sum_{i=1}^{n}\psi(u_i^*, \tilde{u}_i^{T+1}, \ldots, \tilde{u}_i^{T+t+1}, \tilde{f}_i^{T+1}, \ldots \tilde{f}_i^{T+t})\right.$$

$$\left. - \frac{1}{n}\sum_{i=1}^{n}\psi(u_i^*, \hat{u}_i^1, \ldots, \hat{u}_i^{t+1}, \hat{f}_i^1, \ldots \hat{f}_i^t)\right| = 0 \quad \text{almost surely.} \tag{B.61}$$

*Proof.* **Proof of** (B.60). We prove by induction. Consider the base case $t = 1$. The formula in (B.6) for $\tilde{\mu}_t$ shows that $\tilde{\mu}_t = \alpha\rho_{\alpha} = \mu_1$ for $t \in [1, T+1]$. Furthermore, Lemma B.2 shows that $\lim_{T\to\infty}\tilde{\sigma}_{T+1,T+1} = \alpha^2(1-\rho_{\alpha}^2)$, which equals $\sigma_{11}$ (defined right before (3.4)).

For $t \geq 2$, assume towards induction that $\lim_{T\to\infty}\tilde{\mu}_{T+\ell} = \mu_{\ell}$ and $\sigma_{T+k,T+\ell} = \sigma_{k,\ell}$, for $k, \ell \in [1, t-1]$. From (B.3)-(B.4), we have

$$\tilde{\mu}_{T+t} = \alpha\mathbb{E}\{u_t(\tilde{\mu}_{T+t-1}U_* + \tilde{Z}_{T+t-1})U_*\}. \tag{B.62}$$

Recalling that $\tilde{Z}_{T+t-1} \sim \mathcal{N}(0, \tilde{\sigma}_{T+t-1,T+t-1})$ and $Z_{t-1} \sim \mathcal{N}(0, \sigma_{t-1,t-1})$, by the induction hypothesis and the continuous mapping theorem, the sequence of random variables $\{u_t(\tilde{\mu}_{T+t-1}U_* + \tilde{Z}_{T+t-1})U_*\}$ converges in distribution as $T \to \infty$ to $u_t(\mu_{t-1}U_* + Z_{t-1})U_*$. We now claim that the sequence $\{u_t(\tilde{\mu}_{T+t-1}U_* + \tilde{Z}_{T+t-1})U_*\}$ is uniformly integrable, from which it follows that [12]

$$\lim_{T\to\infty}\tilde{\mu}_{T+t} = \alpha\mathbb{E}\{u_t(\mu_{t-1}U_* + Z_{t-1})U_*\} = \mu_t. \tag{B.63}$$

We show uniform integrability by showing that $\sup_T \mathbb{E}\{|\mathsf{u}_t(\tilde{\mu}_{T+t-1}U_* + \tilde{Z}_{T+t-1})U_*|^{1+\varepsilon/2}\}$ is bounded, where we recall that $\varepsilon > 0$ is any constant such that $\mathbb{E}\{U_*^{2+\varepsilon}\}$ exists. Using $L_t \geq 1$ to denote a Lipschitz constant of $\mathsf{u}_t$, we have

$$\mathbb{E}\{|\mathsf{u}_t(\tilde{\mu}_{T+t-1}U_* + \tilde{Z}_{T+t-1})U_*|^{1+\varepsilon/2}\}$$

$$\leq L_t^{1+\varepsilon/2}\mathbb{E}\left\{\left||\tilde{\mu}_{T+t-1}|U_*^2 + |\tilde{Z}_{T+t-1}U_*| + |\mathsf{u}_t(0)U_*|\right|^{1+\varepsilon/2}\right\}$$

$$\overset{(a)}{\leq} (3L_t)^{1+\varepsilon/2}\left(|\tilde{\mu}_{T+t-1}|^{1+\varepsilon/2}\,\mathbb{E}\{|U_*|^{2+\varepsilon}\} + \left(\mathbb{E}\{|\tilde{Z}_{T+t-1}|^{1+\varepsilon/2}\} + |\mathsf{u}_t(0)|^{1+\varepsilon/2}\right)\mathbb{E}\{|U_*|^{1+\varepsilon/2}\}\right)$$

$$\overset{(b)}{<} \infty, \tag{B.64}$$

where (a) is obtained using Hölder's inequality, and (b) holds because $\tilde{\mu}_{T+t-1} \to \mu_{t-1}$ and $\tilde{\sigma}_{T+t-1,T+t-1} \to \sigma_{t-1,t-1}$ by the induction hypothesis.

Next, consider $\tilde{\sigma}_{T+s,T+t}$ for $s \in [1,t]$. From (B.4),

$$\tilde{\sigma}_{T+s,T+t} = \sum_{j=0}^{T+s-1}\sum_{k=0}^{T+t-1} \kappa_{j+k+2}^{\infty}\left(\prod_{i=T+s-j+1}^{T+s} \mathbb{E}\{\tilde{\mathsf{u}}_i'(\tilde{F}_{i-1})\}\right)$$

$$\cdot \left(\prod_{i=T+t-k+1}^{T+t} \mathbb{E}\{\tilde{\mathsf{u}}_i'(\tilde{F}_{i-1})\}\right)\mathbb{E}\{\tilde{U}_{T+s-j}\tilde{U}_{T+t-k}\}$$

$$:= A_1 + A_2 + A_3 + A_4, \tag{B.65}$$

where the four terms correspond to the sum over different subsets of the indices $(j,k)$. By using the definition of $\tilde{\mathsf{u}}_i(\cdot)$ in (B.3), those terms can be written as

$$A_1 = \sum_{j=0}^{s-2}\sum_{k=0}^{t-2} \kappa_{j+k+2}^{\infty}\left(\prod_{i=s-j+1}^{s} \mathbb{E}\{\mathsf{u}_i'(\tilde{F}_{T+i-1})\}\right)\left(\prod_{i=t+1-k}^{t} \mathbb{E}\{\mathsf{u}_i'(\tilde{F}_{T+i-1})\}\right)$$

$$\cdot \mathbb{E}\{\tilde{U}_{T+s-j}\tilde{U}_{T+t-k}\}, \tag{B.66}$$

$$A_2 = \sum_{j=0}^{s-2}\sum_{k=t-1}^{T+t-1} \left(\frac{1}{\alpha}\right)^{(k-t+1)}\kappa_{j+k+2}^{\infty}\left(\prod_{i=s-j+1}^{s} \mathbb{E}\{\mathsf{u}_i'(\tilde{F}_{T+i-1})\}\right)\left(\prod_{i=2}^{t} \mathbb{E}\{\mathsf{u}_i'(\tilde{F}_{T+i-1})\}\right)$$

$$\cdot \mathbb{E}\{\tilde{U}_{T+s-j}\tilde{U}_{T+t-k}\}, \tag{B.67}$$

$$A_3 = \sum_{j=s-1}^{T+s-1}\sum_{k=0}^{t-2} \left(\frac{1}{\alpha}\right)^{(j-s+1)}\kappa_{j+k+2}^{\infty}\left(\prod_{i=2}^{s} \mathbb{E}\{\mathsf{u}_i'(\tilde{F}_{T+i-1})\}\right)\left(\prod_{i=k-t+1}^{t} \mathbb{E}\{\mathsf{u}_i'(\tilde{F}_{T+i-1})\}\right)$$

$$\cdot \mathbb{E}\{\tilde{U}_{T+s-j}\tilde{U}_{T+t-k}\}, \tag{B.68}$$

$$A_4 = \sum_{j=s-1}^{T+s-1}\sum_{k=t-1}^{T+t-1} \left(\frac{1}{\alpha}\right)^{(j+k-s-t+2)}\kappa_{j+k+2}^{\infty}\left(\prod_{i=2}^{s} \mathbb{E}\{\mathsf{u}_i'(\tilde{F}_{T+i-1})\}\right)\left(\prod_{i=2}^{t} \mathbb{E}\{\mathsf{u}_i'(\tilde{F}_{T+i-1})\}\right)$$

$$\cdot \mathbb{E}\{\tilde{U}_{T+s-j}\tilde{U}_{T+t-k}\}. \tag{B.69}$$

For $i \in [2,t]$, the induction hypothesis implies that $\tilde{F}_{T+i-1} = \tilde{\mu}_{T+i-1}U_* + \tilde{Z}_{T+i-1} \overset{d}{\to} F_{i-1} = \mu_{i-1}U_* + Z_{i-1}$. Since $u_i$ is Lipschitz and continuously differentiable, Lemma D.1 implies that

$$\lim_{T\to\infty} \mathbb{E}\{\mathsf{u}_i'(\tilde{F}_{T+i-1})\} = \mathbb{E}\{\mathsf{u}_i'(F_{i-1})\}, \qquad i \in [2,t]. \tag{B.70}$$

Next, note that

$$\tilde{U}_{T+s-j} = \begin{cases} \mathsf{u}_{s-j}(\tilde{F}_{T+s-j-1}), & 0 \leq j \leq s-2, \\ \tilde{F}_{T+s-j-1}/\alpha, & s-1 \leq j \leq T+s-1, \end{cases}$$

$$\tilde{U}_{T+t-k} = \begin{cases} \mathsf{u}_{t-k}(\tilde{F}_{T+t-k-1}), & 0 \leq k \leq t-2, \\ \tilde{F}_{T+t-k-1}/\alpha, & t-1 \leq k \leq T+t-1. \end{cases} \tag{B.71}$$

We separately consider $\mathbb{E}\{\tilde{U}_{T+s-j}\tilde{U}_{T+t-k}\}$ for the four cases of $(j,k)$, corresponding to $A_1, A_2, A_3, A_4$. First, for $j \in [0, t-2]$, $k \in [0, s-2]$, we have

$$\mathbb{E}\{\tilde{U}_{T+s-j}\tilde{U}_{T+t-k}\} = \mathbb{E}\{\mathsf{u}_{s-j}(\tilde{F}_{T+s-j-1})\,\mathsf{u}_{t-k}(\tilde{F}_{T+t-k-1})\}. \tag{B.72}$$

By the induction hypothesis and the continuous mapping theorem, the sequence $\{\mathsf{u}_{s-j}(\tilde{F}_{T+s-j-1})\mathsf{u}_{t-k}(\tilde{F}_{T+t-k-1})\}$ converges in distribution to $\mathsf{u}_{s-j}(F_{s-j-1})\mathsf{u}_{t-k}(F_{t-k-1})$ as $T \to \infty$. From an argument similar to (B.64), we also deduce that $\{\mathsf{u}_{s-j}(\tilde{F}_{T+s-j-1})\mathsf{u}_{t-k}(\tilde{F}_{T+t-k-1})\}$ is uniformly integrable, from which it follows that

$$\lim_{T\to\infty} \mathbb{E}\{\mathsf{u}_{s-j}(\tilde{F}_{T+s-j-1})\mathsf{u}_{t-k}(\tilde{F}_{T+t-k-1})\} = \mathbb{E}\{\mathsf{u}_{s-j}(F_{s-j-1})\mathsf{u}_{t-k}(F_{t-k-1})\},$$
$$j \in [0, t-2], \ k \in [0, s-2]. \tag{B.73}$$

Eqs. (B.70) and (B.73) imply that

$$\lim_{T\to\infty} A_1 = \sum_{j=0}^{s-2}\sum_{k=0}^{t-2} \kappa_{j+k+2}^\infty \left( \prod_{i=s-j+1}^{s} \mathbb{E}\{\mathsf{u}_i'(F_{i-1})\} \right) \left( \prod_{i=t+1-k}^{t} \mathbb{E}\{\mathsf{u}_i'(F_{i-1})\} \right) \mathbb{E}\{U_{s-j}U_{t-k}\}. \tag{B.74}$$

Next consider the case where $j \in [s-1, \, T+s-1]$ and $k \in [t-1, \, T+t-1]$. Here,

$$\mathbb{E}\{\tilde{U}_{T+s-j}\tilde{U}_{T+t-k}\} = \frac{1}{\alpha^2}\mathbb{E}\{\tilde{F}_{T-(j+1-s)}\tilde{F}_{T-(k+1-t)}\} = \rho_\alpha^2 + \frac{1}{\alpha^2}\tilde{\sigma}_{T-(j+1-s),T-(k+1-t)}. \tag{B.75}$$

From Lemma B.2, for any $\delta > 0$, for sufficiently large $T$, we have

$$|\tilde{\sigma}_{T-(j+1-s),T-(k+1-t)} - \alpha^2(1-\rho_\alpha^2)| < \delta\xi^{-\max(j+1-s,\,k+1-t)}, \tag{B.76}$$

for some $\xi > 0$ such that $\xi\alpha > \alpha_s$. Combining (B.75)-(B.76) and noting from (3.4) that $\mathbb{E}\{U_{s-j}U_{t-k}\} = \frac{1}{\alpha^2}\mathbb{E}\{F_1^2\} = 1$, we obtain, for sufficiently large $T$:

$$|\mathbb{E}\{\tilde{U}_{T+s-j}\tilde{U}_{T+t-k}\} - \mathbb{E}\{U_{s-j}U_{t-k}\}| < \frac{\delta}{\alpha^2}\xi^{-\max(j+1-s,\,k+1-t)}, \quad \text{for } j \geq (s-1),\ k \geq (t-1). \tag{B.77}$$

Now we write $A_4$ in (B.69) as

$$A_4 = \left(\prod_{i=2}^{s}\mathbb{E}\{\mathsf{u}_i'(\tilde{F}_{T+i-1})\}\right)\left(\prod_{i=2}^{t}\mathbb{E}\{\mathsf{u}_i'(\tilde{F}_{T+i-1})\}\right)$$
$$\cdot \left[\sum_{j=s-1}^{T+s-1}\sum_{k=t-1}^{T+t-1}\left(\frac{1}{\alpha}\right)^{(j+k-s-t+2)}\kappa_{j+k+2}^\infty\,\mathbb{E}\{U_{s-j}U_{t-k}\} + \Delta_4\right], \tag{B.78}$$

where

$$\Delta_4 = \sum_{j=s-1}^{T+s-1}\sum_{k=t-1}^{T+t-1}\left(\frac{1}{\alpha}\right)^{(j+k-s-t+2)}\kappa_{j+k+2}^\infty\,[\mathbb{E}\{\tilde{U}_{T+s-j}\tilde{U}_{T+t-k}\} - \mathbb{E}\{U_{s-j}U_{t-k}\}]. \tag{B.79}$$

Using (B.77), for sufficiently large $T$ we have

$$|\Delta_4| < \frac{\delta}{\alpha^2}\sum_{j=s-1}^{T+s-1}\sum_{k=t-1}^{T+t-1}\left(\frac{1}{\xi\alpha}\right)^{(j+k-s-t+2)}\kappa_{j+k+2}^\infty$$
$$= \frac{\delta}{\alpha^2}(\xi\alpha)^{s+t}\sum_{j=0}^{T}\sum_{k=0}^{T}\left(\frac{1}{\xi\alpha}\right)^{(j+k+s+t)}\kappa_{j+k+s+t}^\infty \tag{B.80}$$
$$< C_{s,t}\delta,$$

for a positive constant $C_{s,t}$ since the double sum is bounded for $\xi\alpha > \alpha_s$ (see (A.5)). Since $\delta > 0$ is arbitrary, this shows that $\Delta_4 \to 0$ as $T \to \infty$. Using this in (B.78) along with (B.70), we obtain

$$\lim_{T\to\infty} A_4 = \left(\prod_{i=2}^{s} \mathbb{E}\{u_i'(F_{i-1})\}\right)\left(\prod_{i=2}^{t} \mathbb{E}\{u_i'(F_{i-1})\}\right)$$
$$\cdot \sum_{j=s-1}^{\infty}\sum_{k=t-1}^{\infty}\left(\frac{1}{\alpha}\right)^{(j+k-s-t+2)}\kappa_{j+k+2}^{\infty}\mathbb{E}\{U_{s-j}U_{t-k}\}. \tag{B.81}$$

Next consider $j \in [0, s-2]$, $k \in [t-1, T+t-1]$. Here

$$\mathbb{E}\{\tilde{U}_{T+s-j}\tilde{U}_{T+t-k}\} = \frac{1}{\alpha}\mathbb{E}\{u_{s-j}(\tilde{F}_{T+s-j-1})\tilde{F}_{T-(k+1-t)}\}$$
$$= \frac{1}{\alpha}\mathbb{E}\{u_{s-j}(\tilde{F}_{T+s-j-1})\tilde{F}_{T+1}\} + \frac{1}{\alpha}\mathbb{E}\{u_{s-j}(\tilde{F}_{T+s-j-1})(\tilde{F}_{T+1} - \tilde{F}_{T-(k+1-t)})\}. \tag{B.82}$$

By the induction hypothesis and the uniform integrability of $\{u_{s-j}(\tilde{F}_{T+s-j-1})\}$, we have

$$\lim_{T\to\infty}\frac{1}{\alpha}\mathbb{E}\{u_{s-j}(\tilde{F}_{T+s-j-1})\tilde{F}_{T+1}\} = \frac{1}{\alpha}\mathbb{E}\{u_{s-j}(F_{s-j-1})F_1\} = \mathbb{E}\{U_{s-j}U_{t-k}\}. \tag{B.83}$$

The second term in (B.82) can be bounded as follows, using the Cauchy-Schwarz inequality:

$$|\mathbb{E}\{u_{s-j}(\tilde{F}_{T+s-j-1})(\tilde{F}_{T+1} - \tilde{F}_{T-(k+1-t)})\}|$$
$$\le L_t(\tilde{\mu}_{T+s-j-1}^2 + \tilde{\sigma}_{T+s-j-1,T+s-j-1} + C)^{1/2}(\mathbb{E}\{(\tilde{F}_{T+1} - \tilde{F}_{T-(k+1-t)})^2\})^{1/2}. \tag{B.84}$$

Using Lemma B.2, for any $\delta > 0$ and $T$ sufficiently large, we have

$$\mathbb{E}\{(\tilde{F}_{T+1} - \tilde{F}_{T-(k+1-t)})^2\} \le |\sigma_{T+1,T+1} - \alpha^2(1 - \rho_\alpha^2)|$$
$$+ |\sigma_{T-(k+1-t),T-(k+1-t)} - \alpha^2(1-\rho_\alpha^2)| + 2|\sigma_{T-(k+1-t),T+1} - \alpha^2(1-\rho_\alpha^2)|$$
$$< \delta\xi^{-(k+1-t)}. \tag{B.85}$$

Combining (B.82)-(B.84), we deduce that for any $\delta > 0$, the following holds for sufficiently large $T$:

$$|\mathbb{E}\{\tilde{U}_{T+s-j}\tilde{U}_{T+t-k}\} - \mathbb{E}\{U_{s-j}U_{t-k}\}| < \delta\xi^{-(k+1-t)}, \quad \text{for } j \in [0, s-2],\ k \in [t-1, T+t-1]. \tag{B.86}$$

We write $A_2$ in (B.67) as

$$A_2 = \left(\prod_{i=2}^{t}\mathbb{E}\{u_i'(\tilde{F}_{T+i-1})\}\right)\sum_{j=0}^{s-2}\left(\prod_{i=s-j+1}^{s}\mathbb{E}\{u_i'(\tilde{F}_{T+i-1})\}\right)$$
$$\cdot\left[\sum_{k=t-1}^{T+t-1}\left(\frac{1}{\alpha}\right)^{(k-t+1)}\kappa_{j+k+2}^{\infty}\mathbb{E}\{U_{s-j}U_{t-k}\} + \Delta_{2,j}\right], \tag{B.87}$$

where

$$\Delta_{2,j} = \sum_{k=t-1}^{T+t-1}\left(\frac{1}{\alpha}\right)^{(k-t+1)}\kappa_{j+k+2}^{\infty}(\mathbb{E}\{\tilde{U}_{T+s-j}\tilde{U}_{T+t-k}\} - \mathbb{E}\{U_{s-j}U_{t-k}\}). \tag{B.88}$$

From (B.86), for any $\delta > 0$ and sufficiently large $T$ we have

$$|\Delta_{2,j}| < \delta(\xi\alpha)^{j+t+1}\sum_{k=1}^{T+1}\left(\frac{1}{\xi\alpha}\right)^{j+k+t}\kappa_{j+k+t}^{\infty} < C_{s,j}\delta, \tag{B.89}$$

for a positive constant $C_{s,j}$ since the sum over $k$ is bounded (see (A.4)). Using this in (B.87) along with (B.70), we obtain

$$\lim_{T\to\infty} A_2 = \sum_{j=0}^{s-2}\sum_{k=t-1}^{\infty}\left(\frac{1}{\alpha}\right)^{(k-t+1)}\kappa_{j+k+2}^{\infty}\mathbb{E}\{U_{s-j}U_{t-k}\}\left(\prod_{i=s-j+1}^{s}\mathbb{E}\{u_i'(F_{i-1})\}\right)$$
$$\cdot\left(\prod_{i=2}^{t}\mathbb{E}\{u_i'(F_{i-1})\}\right). \tag{B.90}$$

Using a similar argument, we also have

$$\lim_{T \to \infty} A_3 = \sum_{j=s-1}^{\infty} \sum_{k=0}^{t-2} \left(\frac{1}{\alpha}\right)^{(j-s+1)} \kappa_{j+k+2}^{\infty} \mathbb{E}\{U_{s-j} U_{t-k}\} \left(\prod_{i=2}^{s} \mathbb{E}\{\mathsf{u}_i'(F_{i-1})\}\right)$$
$$\cdot \left(\prod_{i=t-k+1}^{t} \mathbb{E}\{\mathsf{u}_i'(F_{i-1})\}\right). \tag{B.91}$$

Noting that the sum of the limits in (B.74), (B.81), (B.90) and (B.91) equals $\sigma_{s,t}$ (defined in (3.6)), we have shown that $\lim_{T \to \infty} \tilde{\sigma}_{T+s,T+t} = \sigma_{s,t}$.

**Proof of** (B.61). Since $\psi \in \mathrm{PL}(2)$, for some universal constant $C > 0$ we have

$$\left| \frac{1}{n} \sum_{i=1}^{n} \psi(u_i^*, \tilde{u}_i^{T+1}, \ldots, \tilde{u}_i^{T+t+1}, \tilde{f}_i^{T+1}, \ldots \tilde{f}_i^{T+t}) - \frac{1}{n} \sum_{i=1}^{n} \psi(u_i^*, \hat{u}_i^1, \ldots, \hat{u}_i^{t+1}, \hat{f}_i^1, \ldots \hat{f}_i^t) \right|$$

$$\leq \frac{C}{n} \sum_{i=1}^{n} \left( 1 + |u_i^*| + \sum_{\ell=1}^{t+1} \left( |\tilde{u}_i^{T+\ell}| + |\hat{u}_i^\ell| \right) + \sum_{\ell=1}^{t} \left( |\tilde{f}_i^{T+\ell}| + |\hat{f}_i^\ell| \right) \right)$$

$$\cdot \left( (\tilde{u}_i^{T+1} - \hat{u}_i^1)^2 + \ldots + (\tilde{u}_i^{T+t+1} - \hat{u}_i^{t+1})^2 + (\tilde{f}_i^{T+1} - \hat{f}_i^1)^2 + \ldots + (\tilde{f}_i^{T+t} - \hat{f}_i^t)^2 \right)^{\frac{1}{2}}$$

$$\leq 2C(t+2) \left[ 1 + \frac{\|\boldsymbol{u}^*\|^2}{n} + \sum_{\ell=1}^{t+1} \left( \frac{\|\tilde{\boldsymbol{u}}^{T+\ell}\|^2}{n} + \frac{\|\hat{\boldsymbol{u}}^\ell\|^2}{n} \right) + \sum_{\ell=1}^{t} \left( \frac{\|\tilde{\boldsymbol{f}}^{T+\ell}\|^2}{n} + \frac{\|\hat{\boldsymbol{f}}^\ell\|^2}{n} \right) \right]^{\frac{1}{2}}$$

$$\cdot \left( \frac{\|\tilde{\boldsymbol{u}}^{T+1} - \hat{\boldsymbol{u}}^1\|^2}{n} + \ldots + \frac{\|\tilde{\boldsymbol{u}}^{T+t+1} - \hat{\boldsymbol{u}}^{t+1}\|^2}{n} + \frac{\|\tilde{\boldsymbol{f}}^{T+1} - \hat{\boldsymbol{f}}^1\|^2}{n} + \ldots + \frac{\|\tilde{\boldsymbol{f}}^{T+t} - \hat{\boldsymbol{f}}^t\|^2}{n} \right)^{\frac{1}{2}}, \tag{B.92}$$

where the last inequality is obtained by using Cauchy-Schwarz inequality (twice).

We will inductively show that in the limit $T, n \to \infty$ (with the limit in $n$ taken first): i) the terms $\frac{\|\tilde{\boldsymbol{u}}^{T+1}-\hat{\boldsymbol{u}}^1\|^2}{n}, \ldots, \frac{\|\tilde{\boldsymbol{f}}^{T+1}-\hat{\boldsymbol{f}}^1\|^2}{n}, \ldots, \frac{\|\tilde{\boldsymbol{f}}^{T+t}-\hat{\boldsymbol{f}}^t\|^2}{n}$ all converge to 0 almost surely, and ii) each of the terms within the square brackets in (B.92) converges to a finite deterministic value.

Base case: $t = 1$. From Lemma B.7, we have

$$\lim_{T \to \infty} \lim_{n \to \infty} \frac{\|\tilde{\boldsymbol{u}}^{T+1} - \hat{\boldsymbol{u}}^1\|^2}{n} = 0. \tag{B.93}$$

From the definitions of $\tilde{\boldsymbol{f}}^{T+1}$ and $\hat{\boldsymbol{f}}^1$ in (5.1) and (B.57), we have

$$\|\tilde{\boldsymbol{f}}^{T+1} - \hat{\boldsymbol{f}}^1\|^2 = \left\| \boldsymbol{X}(\tilde{\boldsymbol{u}}^{T+1} - \hat{\boldsymbol{u}}_1) - \left( \sum_{i=1}^{T+1} \tilde{\mathsf{b}}_{T+1,i} \tilde{\boldsymbol{u}}^i - \bar{\mathsf{b}}_{1,1} \hat{\boldsymbol{u}}^1 \right) \right\|^2$$
$$\leq 2\|\boldsymbol{X}\|_{\mathrm{op}}^2 \|\tilde{\boldsymbol{u}}^{T+1} - \hat{\boldsymbol{u}}^1\|^2 + \left\| \sum_{i=1}^{T+1} \tilde{\mathsf{b}}_{T+1,i} \tilde{\boldsymbol{u}}^i - \bar{\mathsf{b}}_{1,1} \hat{\boldsymbol{u}}^1 \right\|^2. \tag{B.94}$$

From [10, Theorem 2.1], we know that the $\|\boldsymbol{X}\|_{\mathrm{op}} = |\lambda_1(\boldsymbol{X})| \overset{n \to \infty}{\to} |G^{-1}(1/\alpha)|$ almost surely. Therefore, from (B.93), we almost surely have

$$\lim_{T \to \infty} \lim_{n \to \infty} \|\boldsymbol{X}\|_{\mathrm{op}}^2 \frac{\|\tilde{\boldsymbol{u}}^{T+1} - \hat{\boldsymbol{u}}^1\|^2}{n} = 0. \tag{B.95}$$

For the second term in (B.94), recalling that $\tilde{\mathsf{b}}_{T+1,T+1-j} = \kappa_{j+1}\alpha^{-j}$ for $j \in [0,T]$ (see (5.1)), and $\bar{\mathsf{b}}_{1,1} = \sum_{j=0}^{\infty} \kappa_{j+1}^{\infty}\alpha^{-j}$ (see (B.57)), we write

$$
\sum_{i=1}^{T+1} \tilde{\mathsf{b}}_{T+1,i}\tilde{\boldsymbol{u}}^i - \bar{\mathsf{b}}_{1,1}\hat{\boldsymbol{u}}^1
$$

$$
= \sum_{j=0}^{T} (\kappa_{j+1} - \kappa_{j+1}^{\infty})\alpha^{-j}\tilde{\boldsymbol{u}}^{T+1-j} + \sum_{j=0}^{T} \kappa_{j+1}^{\infty}\alpha^{-j}(\tilde{\boldsymbol{u}}^{T+1-j} - \tilde{\boldsymbol{u}}^{T+1})
$$

$$
+ \sum_{j=0}^{T} \kappa_{j+1}^{\infty}\alpha^{-j}(\tilde{\boldsymbol{u}}^{T+1} - \hat{\boldsymbol{u}}^1) - \sum_{j=T+1}^{\infty} \kappa_{j+1}^{\infty}\alpha^{-j}\hat{\boldsymbol{u}}^1. \tag{B.96}
$$

Hence,

$$
\frac{1}{n}\Big\| \sum_{i=1}^{T+1} \tilde{\mathsf{b}}_{T+1,i}\tilde{\boldsymbol{u}}^i - \bar{\mathsf{b}}_{1,1}\hat{\boldsymbol{u}}^1 \Big\|^2 \leq \frac{4}{n}\Big\| \sum_{j=0}^{T} (\kappa_{j+1} - \kappa_{j+1}^{\infty})\alpha^{-j}\tilde{\boldsymbol{u}}^{T+1-j} \Big\|^2
$$

$$
+ \frac{4}{n}\Big\| \sum_{j=0}^{T} \kappa_{j+1}^{\infty}\alpha^{-j}(\tilde{\boldsymbol{u}}^{T+1-j} - \tilde{\boldsymbol{u}}^{T+1}) \Big\|^2 + \frac{4}{n}\Big\| \sum_{j=0}^{T} \kappa_{j+1}^{\infty}\alpha^{-j}(\tilde{\boldsymbol{u}}^{T+1} - \hat{\boldsymbol{u}}^1) \Big\|^2
$$

$$
+ \frac{4}{n}\Big\| \sum_{j=T+1}^{\infty} \kappa_{j+1}^{\infty}\alpha^{-j}\hat{\boldsymbol{u}}^1 \Big\|^2 := R_1 + R_2 + R_3 + R_4. \tag{B.97}
$$

First, by using passages analogous to (B.45)-(B.46), we almost surely have $\lim_{T\to\infty}\lim_{n\to\infty} R_1 = 0$. Considering $R_2$ next, Proposition B.1 implies that almost surely

$$
\lim_{n\to\infty} \frac{1}{n}\Big\| \sum_{j=0}^{T} \kappa_{j+1}^{\infty}\alpha^{-j}(\tilde{\boldsymbol{u}}^{T+1-j} - \tilde{\boldsymbol{u}}^{T+1}) \Big\|^2 = \Big( \sum_{j=0}^{T} \kappa_{j+1}^{\infty}\alpha^{-j}\mathbb{E}\{\tilde{U}_{T+1-j} - \tilde{U}_{T+1}\} \Big)^2
$$

$$
= \sum_{i=0}^{T}\sum_{j=0}^{T} \kappa_{i+1}^{\infty}\kappa_{j+1}^{\infty}\alpha^{-(i+j)}\frac{1}{n}\mathbb{E}\{(\tilde{U}_{T+1-i} - \tilde{U}_{T+1})(\tilde{U}_{T+1-j} - \tilde{U}_{T+1})\}
$$

$$
\overset{(a)}{=} \sum_{i=0}^{T}\sum_{j=0}^{T} \kappa_{i+1}^{\infty}\kappa_{j+1}^{\infty}\alpha^{-(i+j)}(\tilde{\sigma}_{T-j,T-i} + \tilde{\sigma}_{T,T} - \tilde{\sigma}_{T-i,T} - \tilde{\sigma}_{T-j,T}). \tag{B.98}
$$

Here, (a) is obtained from the definition $\tilde{U}_\ell = \tilde{F}_{\ell-1}/\alpha$ from (B.3), for $\ell \in [1, T+1]$. As $T \to \infty$, it was shown in (B.50)-(B.53) that the sum on the RHS of (B.98) converges to 0. Therefore

$$
\lim_{T\to\infty}\lim_{n\to\infty} \frac{1}{n}\Big\| \sum_{j=0}^{T} \kappa_{j+1}^{\infty}\alpha^{-j}(\tilde{\boldsymbol{u}}^{T+1-j} - \tilde{\boldsymbol{u}}^{T+1}) \Big\|^2 = 0 \text{ almost surely.} \tag{B.99}
$$

For the third term in (B.97), recalling that $\hat{\boldsymbol{u}}^1 = \sqrt{n}\boldsymbol{u}_{\mathrm{PCA}}$, we almost surely have

$$
\lim_{T\to\infty} \Big( \sum_{j=0}^{T} \kappa_{j+1}^{\infty}\alpha^{-j} \Big)^2 \lim_{T\to\infty}\lim_{n\to\infty} \frac{\|\tilde{\boldsymbol{u}}^{T+1} - \hat{\boldsymbol{u}}^1\|^2}{n} = 0, \tag{B.100}
$$

where we use Lemma B.7 and the fact that $\sum_{j=0}^{\infty} \kappa_{j+1}^{\infty}\alpha^{-j} = R(1/\alpha)$ is convergent (see (A.4)). The convergence of this series also implies that $\lim_{T\to\infty} \sum_{j=T+1}^{\infty} \kappa_{j+1}\alpha^{-j} = 0$, and hence the fourth term in (B.97) goes to 0. We have therefore shown that

$$
\lim_{T\to\infty}\lim_{n\to\infty} \frac{1}{n}\Big\| \sum_{i=1}^{T+1} \tilde{\mathsf{b}}_{T+1,i}\tilde{\boldsymbol{u}}^i - \bar{\mathsf{b}}_{1,1}\hat{\boldsymbol{u}}^1 \Big\|^2 = 0, \tag{B.101}
$$

almost surely. Using (B.95) and (B.101) in (B.94) shows that almost surely

$$
\lim_{T\to\infty}\lim_{n\to\infty} \frac{1}{n}\|\tilde{\boldsymbol{f}}^{T+1} - \hat{\boldsymbol{f}}^1\|^2 = 0. \tag{B.102}
$$

Recalling that $\tilde{\boldsymbol{u}}^{T+2} = u_2(\tilde{\boldsymbol{u}}^{T+1})$, $\hat{\boldsymbol{u}}^2 = u_2(\hat{\boldsymbol{f}}^1)$ and that $u_2$ is Lipschitz, we have $\|\tilde{\boldsymbol{u}}^{T+2} - \hat{\boldsymbol{u}}^2\| \le L_2\|\tilde{\boldsymbol{f}}^{T+1} - \hat{\boldsymbol{f}}^1\|$, where $L_2$ is the Lipschitz constant. Eq. (B.102) therefore implies

$$\lim_{T\to\infty}\lim_{n\to\infty}\frac{1}{n}\|\tilde{\boldsymbol{u}}^{T+2} - \hat{\boldsymbol{u}}^2\|^2 = 0 \text{ almost surely.} \tag{B.103}$$

By the triangle inequality, we have for $t \ge 1$:

$$\|\tilde{\boldsymbol{u}}^{T+t}\| - \|\tilde{\boldsymbol{u}}^{T+t} - \hat{\boldsymbol{u}}^t\| \le \|\hat{\boldsymbol{u}}^t\| \le \|\tilde{\boldsymbol{u}}^{T+t}\| + \|\tilde{\boldsymbol{u}}^{T+t} - \hat{\boldsymbol{u}}^t\|. \tag{B.104}$$

Therefore, from (B.93), Proposition B.1, (3.4) and (3.5), we almost surely have

$$\lim_{n\to\infty}\frac{\|\hat{\boldsymbol{u}}^1\|^2}{n} = \lim_{T\to\infty}\lim_{n\to\infty}\frac{\|\tilde{\boldsymbol{u}}^{T+1}\|^2}{n} = \lim_{T\to\infty}\frac{1}{\alpha^2}(\tilde{\mu}_{T+1}^2 + \tilde{\sigma}_{T+1,T+1}) \overset{(a)}{=} \frac{1}{\alpha^2}(\mu_1^2 + \sigma_{1,1}) = 1, \tag{B.105}$$

where (a) is due to (B.60). Similarly using (B.102), (B.103), Proposition B.1, and (3.5), we almost surely have

$$\lim_{n\to\infty}\frac{\|\hat{\boldsymbol{u}}^2\|^2}{n} = \lim_{T\to\infty}\lim_{n\to\infty}\frac{\|\tilde{\boldsymbol{u}}^{T+2}\|^2}{n} = \mathbb{E}\{u_2(\mu_2 U_* + Z_2)^2\},$$

$$\lim_{n\to\infty}\frac{\|\hat{\boldsymbol{f}}^1\|^2}{n} = \lim_{T\to\infty}\lim_{n\to\infty}\frac{\|\tilde{\boldsymbol{f}}^{T+1}\|^2}{n} = \mu_1^2 + \sigma_{1,1} = \alpha^2. \tag{B.106}$$

Using (B.93), (B.102), (B.103), (B.105), and (B.106) in (B.92), we conclude

$$\left|\frac{1}{n}\sum_{i=1}^n \psi(u_i^*, \tilde{u}_i^{T+1}, \tilde{u}_i^{T+2}, \tilde{f}_i^{T+1}) - \frac{1}{n}\sum_{i=1}^n \psi(u_i^*, \hat{u}_i^1, \hat{u}_i^2, \hat{f}_i^1)\right| = 0 \quad \text{almost surely.} \tag{B.107}$$

Induction step: For $t \ge 2$, assume towards induction that almost surely

$$\lim_{T\to\infty}\lim_{n\to\infty}\frac{1}{n}\|\tilde{\boldsymbol{f}}^{T+\ell-1} - \hat{\boldsymbol{f}}^{\ell-1}\|^2 = 0, \qquad \lim_{T\to\infty}\lim_{n\to\infty}\frac{1}{n}\|\tilde{\boldsymbol{u}}^{T+\ell} - \hat{\boldsymbol{u}}^\ell\|^2 = 0, \quad \text{for } 2 \le \ell \le t,$$

$$\lim_{T\to\infty}\lim_{n\to\infty}\left|\frac{1}{n}\sum_{i=1}^n \psi(u_i^*, \tilde{u}_i^{T+1}, \ldots, \tilde{u}_i^{T+\ell}, \tilde{f}_i^{T+1}, \ldots \tilde{f}_i^{T+\ell-1})\right.$$

$$\left. - \frac{1}{n}\sum_{i=1}^n \psi(u_i^*, \hat{u}_i^1, \ldots, \hat{u}_i^\ell, \hat{f}_i^1, \ldots \hat{f}_i^{\ell-1})\right| = 0, \qquad \text{for } 2 \le \ell \le t.$$

$$\tag{B.108}$$

Using the definitions of $\tilde{\boldsymbol{f}}^{T+t}$ and $\hat{\boldsymbol{f}}^t$ in (5.3) and (B.58) and applying the Cauchy-Schwarz inequality, we have

$$\frac{1}{n}\|\tilde{\boldsymbol{f}}^{T+t} - \hat{\boldsymbol{f}}^t\|^2 \le \frac{(t+1)}{n}\left(\|\boldsymbol{X}(\tilde{\boldsymbol{u}}^{T+t} - \hat{\boldsymbol{u}}^t)\|^2 + \sum_{\ell=2}^t \|\tilde{\mathsf{b}}_{T+t,T+\ell}\tilde{\boldsymbol{u}}^{T+\ell} - \bar{\mathsf{b}}_{t,\ell}\hat{\boldsymbol{u}}^\ell\|^2\right.$$

$$\left. + \left\|\sum_{i=1}^{T+1}\tilde{\mathsf{b}}_{T+t,i}\tilde{\boldsymbol{u}}^i - \bar{\mathsf{b}}_{t,1}\hat{\boldsymbol{u}}^1\right\|^2\right). \tag{B.109}$$

For the first term on the right, we have $\|\boldsymbol{X}(\tilde{\boldsymbol{u}}^{T+t} - \hat{\boldsymbol{u}}^t)\|^2 \le \|\boldsymbol{X}\|_{\mathrm{op}}^2\|\tilde{\boldsymbol{u}}^{T+t} - \hat{\boldsymbol{u}}^t\|^2$. Since $\|\boldsymbol{X}\|_{\mathrm{op}} \to |G^{-1}(1/\alpha)|$, using the induction hypothesis we obtain

$$\lim_{T\to\infty}\lim_{n\to\infty}\frac{1}{n}\|\boldsymbol{X}(\tilde{\boldsymbol{u}}^{T+t} - \hat{\boldsymbol{u}}^t)\|^2 = 0 \quad \text{almost surely.} \tag{B.110}$$

Next consider $\frac{1}{n}\|\tilde{\mathsf{b}}_{T+t,T+\ell}\tilde{\boldsymbol{u}}^{T+\ell} - \bar{\mathsf{b}}_{t,\ell}\hat{\boldsymbol{u}}^\ell\|^2$, which, for $\ell \in [2, t]$ can be bounded as

$$\frac{1}{n}\|\tilde{\mathsf{b}}_{T+t,T+\ell}\tilde{\boldsymbol{u}}^{T+\ell} - \bar{\mathsf{b}}_{t,\ell}\hat{\boldsymbol{u}}^\ell\|^2 \le 2\tilde{\mathsf{b}}_{T+t,T+\ell}\frac{\|\tilde{\boldsymbol{u}}^{T+\ell} - \hat{\boldsymbol{u}}^\ell\|^2}{n} + 2\frac{\|\hat{\boldsymbol{u}}^\ell\|^2}{n}(\tilde{\mathsf{b}}_{T+t,T+\ell} - \bar{\mathsf{b}}_{t,\ell})^2. \tag{B.111}$$

By the induction hypothesis, we almost surely have

$$\lim_{T\to\infty}\lim_{n\to\infty}\frac{\|\tilde{\boldsymbol{u}}^{T+\ell}-\hat{\boldsymbol{u}}^\ell\|^2}{n}=0,\quad\text{and}\tag{B.112}$$

$$\lim_{n\to\infty}\frac{\|\hat{\boldsymbol{u}}^\ell\|^2}{n}=\lim_{T\to\infty}\lim_{n\to\infty}\frac{\|\tilde{\boldsymbol{u}}^{T+\ell}\|^2}{n}=\lim_{T\to\infty}(\tilde{\mu}_{T+\ell}^2+\tilde{\sigma}_{T+\ell,T+\ell})=\mu_\ell^2+\sigma_{\ell,\ell},\tag{B.113}$$

where the last equality is due to (B.60). Furthermore, $\tilde{\mathsf{b}}_{T+t,T+t}=\kappa_1\to\kappa_1^\infty=\bar{\mathsf{b}}_{t,t}$ as $n\to\infty$. For $\ell\in[2,t-1]$, from (5.4) we have $\tilde{\mathsf{b}}_{T+t,T+\ell}=\kappa_{t-\ell+1}\prod_{i=\ell+1}^t\langle\mathsf{u}_i'(\tilde{\boldsymbol{f}}^{T+i-1})\rangle$. Proposition B.1 implies that the empirical distribution of $\tilde{\boldsymbol{f}}^{T+i-1}$ converges almost surely in Wasserstein-2 distance to the law of $\tilde{F}_{T+i-1}\equiv\tilde{\mu}_{T+i-1}U_*+\tilde{Z}_{T+i-1}$. Therefore, applying Lemma D.1, we almost surely have

$$\lim_{n\to\infty}\tilde{\mathsf{b}}_{T+t,T+\ell}=\kappa_{t-\ell+1}^\infty\prod_{i=\ell+1}^t\mathbb{E}\{\mathsf{u}_i'(\tilde{F}_{T+i-1})\}.\tag{B.114}$$

Since $\tilde{F}_{T+i-1}$ converges in distribution to $F_{i-1}\equiv\mu_{i-1}U_*+Z_{i-1}$ as $T\to\infty$, applying Lemma D.1 once again, we obtain

$$\lim_{T\to\infty}\lim_{n\to\infty}\tilde{\mathsf{b}}_{T+t,T+\ell}=\kappa_{t-\ell+1}^\infty\prod_{i=\ell+1}^t\mathbb{E}\{\mathsf{u}_i'(F_{i-1})\}.\tag{B.115}$$

Using (B.112), (B.113) and (B.115) in (B.111), we obtain

$$\lim_{T\to\infty}\lim_{n\to\infty}\frac{1}{n}\|\tilde{\mathsf{b}}_{T+t,T+\ell}\tilde{\boldsymbol{u}}^{T+\ell}-\bar{\mathsf{b}}_{t,\ell}\hat{\boldsymbol{u}}^\ell\|^2=0\quad\text{almost surely for }\ell\in[2,t].\tag{B.116}$$

To bound the last term in (B.109), we write it as

$$\Big\|\sum_{i=1}^{T+1}\tilde{\mathsf{b}}_{T+t,i}\tilde{\boldsymbol{u}}^i-\bar{\mathsf{b}}_{t,1}\boldsymbol{u}^1\Big\|^2=\Big\|\sum_{j=0}^T\tilde{\mathsf{b}}_{T+t,T+1-j}\tilde{\boldsymbol{u}}^{T+1-j}-\bar{\mathsf{b}}_{t,1}\hat{\boldsymbol{u}}^1\Big\|^2,\tag{B.117}$$

where from (5.4) we have

$$\tilde{\mathsf{b}}_{T+t,T+1-j}=\kappa_{t+j}\alpha^{-j}\prod_{i=2}^t\langle\mathsf{u}_i'(\tilde{\boldsymbol{f}}^{T+i-1})\rangle,\qquad 0\le j\le T.\tag{B.118}$$

Using this together with the formula for $\bar{\mathsf{b}}_{t,1}$ in (B.59), we have

$$\frac{1}{n}\Big\|\sum_{i=1}^{T+1}\tilde{\mathsf{b}}_{T+t,i}\tilde{\boldsymbol{u}}^i-\bar{\mathsf{b}}_{t,1}\hat{\boldsymbol{u}}^1\Big\|^2$$

$$=\frac{1}{n}\Big\|\prod_{\ell=2}^t\langle\mathsf{u}_\ell'(\tilde{\boldsymbol{f}}^{T+\ell-1})\rangle\sum_{j=0}^T\kappa_{t+j}\alpha^{-j}\,\tilde{\boldsymbol{u}}^{T+1-j}-\prod_{\ell=2}^t\mathbb{E}\{\mathsf{u}_\ell'(F_{\ell-1})\}\sum_{i=0}^\infty\kappa_{t+i}^\infty\alpha^{-i}\hat{\boldsymbol{u}}^1\Big\|^2$$

$$\le 3\left(\frac{1}{n}\Big\|\prod_{\ell=2}^t\langle\mathsf{u}_\ell'(\tilde{\boldsymbol{f}}^{T+\ell-1})\rangle\sum_{j=0}^T\kappa_{t+j}\alpha^{-j}\,\tilde{\boldsymbol{u}}^{T+1-j}-\prod_{\ell=2}^t\mathbb{E}\{\mathsf{u}_\ell'(F_{\ell-1})\}\sum_{j=0}^T\kappa_{t+j}^\infty\alpha^{-j}\,\tilde{\boldsymbol{u}}^{T+1-j}\Big\|^2\right.$$

$$+\frac{1}{n}\Big\|\prod_{\ell=2}^t\mathbb{E}\{\mathsf{u}_\ell'(F_{\ell-1})\}\sum_{j=0}^T\kappa_{t+j}^\infty\alpha^{-j}\,(\tilde{\boldsymbol{u}}^{T+1-j}-\hat{\boldsymbol{u}}^1)\Big\|^2+$$

$$\left.+\frac{1}{n}\Big\|\prod_{\ell=2}^t\mathbb{E}\{\mathsf{u}_\ell'(F_{\ell-1})\}\sum_{i=T+1}^\infty\kappa_{t+i}^\infty\alpha^{-i}\hat{\boldsymbol{u}}^1\Big\|^2\right):=3(S_1+S_2+S_3).\tag{B.119}$$

Considering the second term $S_2$ first, we have

$$\frac{1}{n}\Big\|\sum_{j=0}^T\kappa_{t+j}^\infty\alpha^{-j}\,(\tilde{\boldsymbol{u}}^{T+1-j}-\hat{\boldsymbol{u}}^1)\Big\|^2$$

$$\le 2\left(\frac{1}{n}\Big\|\sum_{j=0}^T\kappa_{t+j}^\infty\alpha^{-j}\,(\tilde{\boldsymbol{u}}^{T+1-j}-\tilde{\boldsymbol{u}}^{T+1})\Big\|^2+\Big(\sum_{j=0}^T\kappa_{t+j}^\infty\alpha^{-j}\Big)^2\frac{\|\tilde{\boldsymbol{u}}^{T+1}-\hat{\boldsymbol{u}}^1\|^2}{n}\right).\tag{B.120}$$

By an argument similar to (B.98)-(B.99), we have

$$\lim_{T\to\infty}\lim_{n\to\infty}\frac{1}{n}\Big\|\sum_{j=0}^{T}\kappa_{t+j}^{\infty}\alpha^{-j}\left(\tilde{\boldsymbol{u}}^{T+1-j}-\tilde{\boldsymbol{u}}^{T+1}\right)\Big\|^2=0\quad\text{almost surely.}\tag{B.121}$$

Moreover, since $R(1/\alpha)<\infty$, from (A.4) we have

$$\lim_{T\to\infty}\sum_{j=0}^{T}\kappa_{t+j}^{\infty}\alpha^{-j}=\alpha^{t-1}\Big(R(1/\alpha)-\sum_{i=0}^{t-2}\kappa_{i+1}^{\infty}\alpha^{-i}\Big).$$

Combining this with (B.93), we have that almost surely

$$\lim_{T\to\infty}\lim_{n\to\infty}S_2=0.\tag{B.122}$$

Next consider $S_3$. Since the series $\sum_{j=0}^{\infty}\kappa_{t+j}^{\infty}\alpha^{-j}$ converges, $\lim_{T\to 0}\sum_{i=T+1}^{\infty}\kappa_{t+i}^{\infty}\alpha^{-i}=0$. Furthermore, by (B.105), $\|\hat{\boldsymbol{u}}^1\|^2/n$ converges almost surely to a finite value. Therefore

$$\lim_{T\to\infty}\lim_{n\to\infty}S_3=0.\tag{B.123}$$

Finally, we consider the term $S_1$ in (B.119). We have

$$S_1\le 2\Big(\prod_{\ell=2}^{t}\langle\mathsf{u}_\ell'(\tilde{\boldsymbol{f}}^{T+\ell-1})\rangle\Big)^2\frac{1}{n}\Big\|\sum_{j=0}^{T}(\kappa_{t+j}-\kappa_{t+j}^{\infty})\alpha^{-j}\,\tilde{\boldsymbol{u}}^{T+1-j}\Big\|^2$$

$$+2\Big(\prod_{\ell=2}^{t}\langle\mathsf{u}_\ell'(\tilde{\boldsymbol{f}}^{T+\ell-1})\rangle-\prod_{\ell=2}^{t}\mathbb{E}\{\mathsf{u}_\ell'(F_{\ell-1})\}\Big)^2\frac{1}{n}\Big\|\sum_{j=0}^{T}\kappa_{t+j}^{\infty}\alpha^{-j}\,\tilde{\boldsymbol{u}}^{T+1-j}\Big\|^2.\tag{B.124}$$

Proposition B.1 implies that for $\ell\in[2,t]$, the empirical distribution of $\tilde{\boldsymbol{f}}^{T+\ell-1}$ converges almost surely in Wasserstein-2 distance to the law of $\tilde{F}_{T+\ell-1}$, which converges in distribution to $F_{\ell-1}$ (due to (B.60)). Therefore, applying Lemma D.1 twice (as in (B.114)-(B.115)) we almost surely have

$$\prod_{\ell=2}^{t}\langle\mathsf{u}_\ell'(\tilde{\boldsymbol{f}}_{T+\ell-1})\rangle=\prod_{\ell=2}^{t}\mathbb{E}\{\mathsf{u}_\ell'(F_{\ell-1})\}.\tag{B.125}$$

Next, we have already shown that $\lim_{T\to\infty}\lim_{n\to\infty}\frac{1}{n}\|\sum_{j=0}^{T}(\kappa_{t+j}-\kappa_{t+j}^{\infty})\alpha^{-j}\,\tilde{\boldsymbol{u}}^{T+1-j}\|^2=0$ almost surely. (See (B.45)-(B.46) and the subsequent argument.) This, together with (B.125) implies that that $\lim_{T\to\infty}\lim_{n\to\infty}S_1=0$ almost surely. Thus, using (B.122) and (B.123) in (B.119), we have

$$\lim_{T\to\infty}\lim_{n\to\infty}\frac{1}{n}\Big\|\sum_{i=1}^{T+1}\bar{\mathsf{b}}_{T+t,i}\tilde{\boldsymbol{u}}^i-\bar{\mathsf{b}}_{t,1}\hat{\boldsymbol{u}}^1\Big\|^2=0\text{ almost surely.}\tag{B.126}$$

Using (B.110), (B.116), and (B.119) in (B.109), we conclude

$$\lim_{T\to\infty}\lim_{n\to\infty}\frac{1}{n}\|\tilde{\boldsymbol{f}}^{T+t}-\hat{\boldsymbol{f}}^t\|^2=0\quad\text{almost surely .}\tag{B.127}$$

Since $\tilde{\boldsymbol{u}}^{T+t+1}=\mathsf{u}_{t+1}(\tilde{\boldsymbol{f}}^{T+t})$ and $\hat{\boldsymbol{u}}^{t+1}=\mathsf{u}_{t+1}(\hat{\boldsymbol{f}}^t)$, with $\mathsf{u}_{t+1}$ Lipschitz, (B.127) implies that

$$\lim_{T\to\infty}\lim_{n\to\infty}\frac{1}{n}\|\tilde{\boldsymbol{u}}^{T+t+1}-\hat{\boldsymbol{u}}^{t+1}\|^2=0\quad\text{almost surely .}\tag{B.128}$$

Using the arguments in (B.104)-(B.106), we also have almost surely:

$$\lim_{T\to\infty}\lim_{n\to\infty}\frac{\|\tilde{\boldsymbol{f}}^{t+T}\|^2}{n}=\lim_{n\to\infty}\frac{\|\hat{\boldsymbol{f}}^t\|^2}{n}=\mathbb{E}\{F_t^2\},$$

$$\lim_{T\to\infty}\lim_{n\to\infty}\frac{\|\tilde{\boldsymbol{u}}^{T+t+1}\|^2}{n}=\lim_{n\to\infty}\frac{\|\hat{\boldsymbol{u}}^{t+1}\|^2}{n}=\mathbb{E}\{\mathsf{u}_{t+1}(F_t)^2\}.\tag{B.129}$$

Using these together with the induction hypothesis (B.108) in (B.92) completes the proof that

$$\Big|\frac{1}{n}\sum_{i=1}^{n}\psi(u_i^*,\tilde{u}_i^{T+1},\ldots,\tilde{u}_i^{T+t+1},\tilde{f}_i^{T+1},\ldots\tilde{f}_i^{T+t})-\frac{1}{n}\sum_{i=1}^{n}\psi(u_i^*,\hat{u}_i^1,\ldots,\hat{u}_i^{t+1},\hat{f}_i^1,\ldots\hat{f}_i^t)\Big|$$

$$=0\quad\text{almost surely.}$$

$$\tag{B.130}$$

$\square$

## B.5 Proof of Theorem 1

We will first use Lemma B.8 to prove that the state evolution result holds for the iterates of the modified AMP, i.e., for $\psi \in \mathrm{PL}(2)$:

$$\lim_{n \to \infty} \frac{1}{n} \sum_{i=1}^{n} \psi(u_i^*, \hat{u}_i^1, \ldots, \hat{u}_i^{t+1}, \hat{f}_i^1, \ldots, \hat{f}_i^t) = \mathbb{E}\{\psi(U_*, U_1, \ldots, U_{t+1}, F_1, \ldots, F_t)\}. \quad \text{(B.131)}$$

Using the triangle inequality, for $T > 0$ we have the bound

$$\left| \frac{1}{n} \sum_{i=1}^{n} \psi(u_i^*, \hat{u}_i^1, \ldots, \hat{u}_i^{t+1}, \hat{f}_i^1, \ldots, \hat{f}_i^t) - \mathbb{E}\{\psi(U_*, U_1, \ldots, U_{t+1}, F_1, \ldots, F_t)\} \right|$$

$$\leq \left| \frac{1}{n} \sum_{i=1}^{n} \psi(u_i^*, \hat{u}_i^1, \ldots, \hat{u}_i^{t+1}, \hat{f}_i^1, \ldots, \hat{f}_i^t) - \frac{1}{n} \sum_{i=1}^{n} \psi(u_i^*, \tilde{u}_i^{T+1}, \ldots, \tilde{u}_i^{T+t+1}, \tilde{f}_i^{T+1}, \ldots, \tilde{f}_i^{T+t}) \right|$$

$$+ \left| \frac{1}{n} \sum_{i=1}^{n} \psi(u_i^*, \tilde{u}_i^{T+1}, \ldots, \tilde{u}_i^{T+t+1}, \tilde{f}_i^{T+1}, \ldots, \tilde{f}_i^{T+t}) \right.$$

$$\left. - \mathbb{E}\{\psi(U_*, \tilde{U}_{T+1}, \ldots, \tilde{U}_{T+t+1}, \tilde{F}_{T+1}, \ldots, \tilde{F}_{T+t})\} \right|$$

$$+ \left| \mathbb{E}\{\psi(U_*, \tilde{U}_{T+1}, \ldots, \tilde{U}_{T+t+1}, \tilde{F}_{T+1}, \ldots, \tilde{F}_{T+t})\} - \mathbb{E}\{\psi(U_*, U_1, \ldots, U_{t+1}, F_1, \ldots, F_t)\} \right|$$

$$:= S_1 + S_2 + S_3. \quad \text{(B.132)}$$

First consider $S_3$. From (B.60), $(U_*, \tilde{U}_{T+1}, \ldots, \tilde{U}_{T+t+1}, \tilde{F}_{T+1}, \ldots, \tilde{F}_{T+t}))$ converges in distribution to the the law of $(U_*, U_1, \ldots, u_{t+1}, F_1, \ldots, F_t)$ as $T \to \infty$. By Skorokhod's representation theorem [12], to compute the expectations in $S_3$, we can take the sequence of random vectors $(U_*, \tilde{U}_{T+1}, \ldots, \tilde{U}_{T+t+1}, \tilde{F}_{T+1}, \ldots, \tilde{F}_{T+t})$ to be such that they belong to the same probability space and converge almost surely to $(U_*, U_1, \ldots, U_{t+1}, F_1, \ldots, F_t)$ as $T \to \infty$. Then, using the pseudo-Lipschitz property of $\psi$ and using Cauchy-Schwarz inequality (twice, as in (B.92)), we obtain

$$S_3 \leq 2C(t+2) \left( 2 + \sum_{\ell=1}^{t+1} (\mathbb{E}\{\tilde{U}_{T+\ell}^2\} + \mathbb{E}\{U_\ell^2\}) + \sum_{\ell=1}^{t} (\mathbb{E}\{\tilde{F}_{T+\ell}^2\} + \mathbb{E}\{F_\ell^2\}) \right)^{1/2}$$

$$\cdot \left( \sum_{\ell=1}^{t+1} \mathbb{E}\{(\tilde{U}_{T+\ell} - U_\ell)^2\} + \sum_{\ell=1}^{t} \mathbb{E}\{(\tilde{F}_{T+\ell} - F_\ell)^2\} \right)^{1/2}. \quad \text{(B.133)}$$

From Lemma B.8, we have $\lim_{T \to \infty} \mathbb{E}\{\tilde{F}_{T+\ell}^2\} = \mathbb{E}\{F_\ell^2\}$ and $\lim_{T \to \infty} \mathbb{E}\{\tilde{U}_{T+\ell}^2\} = \mathbb{E}\{U_\ell^2\}$. Moreover, since for each $\ell$,

$$\mathbb{E}\{(\tilde{F}_{T+\ell} - F_\ell)^2\} \leq 2\mathbb{E}\{\tilde{F}_{T+\ell}^2\} + 2\mathbb{E}\{F_\ell^2\} < \infty \quad \forall T, \quad \text{(B.134)}$$

by dominated convergence we have $\lim_{T \to \infty} \mathbb{E}\{(\tilde{U}_{T+\ell} - U_\ell)^2\} = \lim_{T \to \infty} \mathbb{E}\{(\tilde{F}_{T+\ell} - F_\ell)^2\} = 0$. Therefore $\lim_{T \to \infty} S_3 = 0$. Furthermore, by Lemma B.8 and Proposition B.1, we also have $\lim_{T \infty} \lim_{n \to \infty} S_1 = \lim_{T \to \infty} \lim_{n \to \infty} S_2 = 0$ almost surely. This proves the state evolution result (B.131) for the modified AMP.

We now prove the result of Theorem 1 by showing that for $t \geq 1$, almost surely:

$$\lim_{n \to \infty} \left| \frac{1}{n} \sum_{i=1}^{n} \psi(u_i^*, u_i^1, \ldots, u_i^{t+1}, f_i^1, \ldots, f_i^t) - \frac{1}{n} \sum_{i=1}^{n} \psi(u_i^*, \hat{u}_i^1, \ldots, \hat{u}_i^{t+1}, \hat{f}_i^1, \ldots, \hat{f}_i^t) \right| = 0,$$

$$\text{(B.135)}$$

$$\lim_{n \to \infty} \frac{\|\boldsymbol{f}^t - \hat{\boldsymbol{f}}^t\|^2}{n} = 0, \quad \lim_{n \to \infty} \frac{\|\boldsymbol{u}^{t+1} - \hat{\boldsymbol{u}}^{t+1}\|^2}{n} = 0. \quad \text{(B.136)}$$

The proof of (B.135)-(B.136) is by induction and similar to that of (B.61). Noting that $\boldsymbol{u}^1 = \hat{\boldsymbol{u}}^1 = \sqrt{n}\boldsymbol{u}_{\text{PCA}}$, assume towards induction that (B.135)-(B.136) hold with $t$ replaced by $t-1$. Since $\psi \in \text{PL}(2)$, by the same arguments as in (B.92) we have

$$
\left| \frac{1}{n} \sum_{i=1}^n \psi(u_i^*, u_i^1, \ldots, u_i^{t+1}, f_i^1, \ldots f_i^t) - \frac{1}{n} \sum_{i=1}^n \psi(u_i^*, \hat{u}_i^1, \ldots, \hat{u}_i^{t+1}, \hat{f}_i^1, \ldots \hat{f}_i^t) \right|
$$

$$
\leq 2C(t+2) \left[ 1 + \frac{\|\boldsymbol{u}^*\|^2}{n} + \sum_{\ell=1}^{t+1} \left( \frac{\|\boldsymbol{u}^\ell\|^2}{n} + \frac{\|\hat{\boldsymbol{u}}^\ell\|^2}{n} \right) + \sum_{\ell=1}^t \left( \frac{\|\boldsymbol{f}^\ell\|^2}{n} + \frac{\|\hat{\boldsymbol{f}}^\ell\|^2}{n} \right) \right]^{\frac{1}{2}}
$$

$$
\cdot \left( \frac{\|\hat{\boldsymbol{u}}^1 - \boldsymbol{u}^1\|^2}{n} + \ldots + \frac{\|\boldsymbol{u}^{t+1} - \hat{\boldsymbol{u}}^{t+1}\|^2}{n} + \frac{\|\boldsymbol{f}^1 - \hat{\boldsymbol{f}}^1\|^2}{n} + \ldots + \frac{\|\boldsymbol{f}^t - \hat{\boldsymbol{f}}^t\|^2}{n} \right)^{\frac{1}{2}}. \quad \text{(B.137)}
$$

Using the definitions of $\boldsymbol{f}^t$ and $\hat{\boldsymbol{f}}^t$ in (3.2) and (B.58), and applying the Cauchy-Schwarz inequality, we have

$$
\frac{1}{n} \|\boldsymbol{f}^t - \hat{\boldsymbol{f}}^t\|^2 \leq \frac{(t+1)}{n} \left( \|\boldsymbol{X}(\boldsymbol{u}^t - \hat{\boldsymbol{u}}^t)\|^2 + \sum_{\ell=1}^t \|\mathsf{b}_{t,\ell}\boldsymbol{u}^\ell - \bar{\mathsf{b}}_{t,\ell}\hat{\boldsymbol{u}}^\ell\|^2 \right)
$$

$$
\leq (t+1) \left( \|\boldsymbol{X}\|_{\text{op}}^2 \frac{1}{n}\|\boldsymbol{u}^t - \hat{\boldsymbol{u}}^t\|^2 + \sum_{\ell=1}^t \frac{2}{n}\|\mathsf{b}_{t,\ell}\boldsymbol{u}^\ell - \bar{\mathsf{b}}_{t,\ell}\boldsymbol{u}^\ell\|^2 + \frac{2}{n}\|\bar{\mathsf{b}}_{t,\ell}\boldsymbol{u}^\ell - \bar{\mathsf{b}}_{t,\ell}\hat{\boldsymbol{u}}^\ell\|^2 \right). \quad \text{(B.138)}
$$

Recall that $\|\boldsymbol{X}\|_{\text{op}}$ converges almost surely to $|G^{-1}(1/\alpha)|$ and by the induction hypothesis, $\frac{1}{n}\|\boldsymbol{u}^\ell - \hat{\boldsymbol{u}}^\ell\|^2 \to 0$, for $\ell \in [1, t]$. Next, we note that $\mathsf{b}_{t,t} = \kappa_1 \to \kappa_1^\infty = \bar{\mathsf{b}}_{t,t}$ as $n \to \infty$. For $\ell \in [2, t-1]$, we have $\mathsf{b}_{t,\ell} = \kappa_{t-\ell+1} \prod_{i=\ell+1}^t \langle \mathsf{u}_i'(\boldsymbol{f}^{i-1}) \rangle$. The induction hypothesis (B.135) implies that the empirical distribution of $\boldsymbol{f}^{i-1}$ converges almost surely in Wasserstein-2 distance to the law of $F_{i-1}$ for $i \in [1, t]$. Therefore, applying Lemma D.1 we almost surely have

$$
\lim_{n \to \infty} \mathsf{b}_{t,\ell} = \kappa_{t-\ell+1}^\infty \prod_{i=\ell+1}^t \mathbb{E}\{\mathsf{u}_i'(F_{i-1})\}. \quad \text{(B.139)}
$$

This shows that $\lim_{n\to\infty} \frac{1}{n}\|\boldsymbol{f}^t - \hat{\boldsymbol{f}}^t\|^2 = 0$ almost surely. Since $\boldsymbol{u}^{t+1} = \mathsf{u}_{t+1}(\boldsymbol{f}^t)$ with $\mathsf{u}_{t+1}$ Lipschitz, we also have $\lim_{n\to\infty} \frac{1}{n}\|\boldsymbol{u}^{t+1} - \hat{\boldsymbol{u}}^{t+1}\|^2 = 0$ almost surely. Moreover using a triangle inequality argument similar to (B.104), for $\ell \in [1, t]$, we almost surely have

$$
\lim_{n\to\infty} \frac{\|\boldsymbol{f}^\ell\|^2}{n} = \lim_{n\to\infty} \frac{\|\hat{\boldsymbol{f}}^\ell\|^2}{n} = \mathbb{E}\{F_\ell^2\}, \qquad \lim_{n\to\infty} \frac{\|\boldsymbol{u}^{\ell+1}\|^2}{n} = \lim_{n\to\infty} \frac{\|\hat{\boldsymbol{u}}^{\ell+1}\|^2}{n} = \mathbb{E}\{\mathsf{u}_{\ell+1}(F_\ell)^2\}.
$$
$$\text{(B.140)}$$

Using this in (B.137), we conclude that

$$
\lim_{n\to\infty} \left| \frac{1}{n} \sum_{i=1}^n \psi(u_i^*, u_i^1, \ldots, u_i^{t+1}, f_i^1, \ldots f_i^t) - \frac{1}{n} \sum_{i=1}^n \psi(u_i^*, \hat{u}_i^1, \ldots, \hat{u}_i^{t+1}, \hat{f}_i^1, \ldots \hat{f}_i^t) \right| = 0,
$$
$$\text{(B.141)}$$

which combined with (B.131) completes the proof of the theorem. $\qquad \square$

## C   Proof of Theorem 2

This appendix is organized as follows. In Appendix C.1, we present the artificial AMP for the rectangular model (1.2), and provide a sketch of the proof. In Appendix C.2, we present the state evolution recursion associated with the artificial AMP iteration. In Appendix C.3, we prove that the first phase of this state evolution admits a unique fixed point. Using this fact, in Appendix C.4, we prove that the artificial AMP iterate at the end of the first phase approaches the left singular vector produced by PCA. Then, in Appendix C.5, we show that *(i)* the iterates in the second phase of the artificial AMP are close to the true AMP iterates, and *(ii)* the related state evolutions also remain close. Finally, in Appendix C.6, we give the proof of Theorem 2.

## C.1 Proof Sketch

**First phase.** We consider the following artificial AMP algorithm. We initialize with

$$\tilde{\boldsymbol{u}}^1 = \sqrt{\Delta_{\text{PCA}}}\boldsymbol{u}^* + \sqrt{1 - \Delta_{\text{PCA}}}\boldsymbol{n}, \quad \tilde{\boldsymbol{g}}^1 = \boldsymbol{X}^\mathsf{T}\tilde{\boldsymbol{u}}^1, \quad \tilde{\boldsymbol{v}}^1 = \frac{\gamma}{\alpha}\tilde{\boldsymbol{g}}^1, \quad \tilde{\boldsymbol{f}}^1 = \boldsymbol{X}\tilde{\boldsymbol{v}}^1 - \kappa_2\frac{\gamma}{\alpha}\tilde{\boldsymbol{u}}^1. \tag{C.1}$$

Here, $\boldsymbol{n}$ has i.i.d. standard Gaussian components and $\Delta_{\text{PCA}}$ is the (limiting) normalized squared correlation of the left PCA estimate, given in (2.3). As in the square case, the initialization of the artificial AMP is impractical. However, this is not a problem, as the artificial AMP is only used as a proof technique. Then, for $2 \le t \le T + 1$, the artificial AMP iterates are

$$\tilde{\boldsymbol{u}}^t = \frac{1}{\alpha}\tilde{\boldsymbol{f}}^{t-1}, \qquad \tilde{\boldsymbol{g}}^t = \boldsymbol{X}^\mathsf{T}\tilde{\boldsymbol{u}}^t - \sum_{i=1}^{t-1}\tilde{\mathsf{b}}_{t,i}\tilde{\boldsymbol{v}}^i,$$

$$\tilde{\boldsymbol{v}}^t = \frac{\gamma}{\alpha}\tilde{\boldsymbol{g}}^t, \qquad \tilde{\boldsymbol{f}}^t = \boldsymbol{X}\tilde{\boldsymbol{v}}^t - \sum_{i=1}^{t}\tilde{\mathsf{a}}_{t,i}\tilde{\boldsymbol{u}}^i, \tag{C.2}$$

where $\tilde{\mathsf{b}}_{t,t-j} = \kappa_{2j}\frac{\gamma}{\alpha}\left(\frac{\gamma}{\alpha^2}\right)^{j-1}$ for $j \in [1, t-1]$, and $\tilde{\mathsf{a}}_{t,t-j} = \kappa_{2(j+1)}\frac{\gamma}{\alpha}\left(\frac{\gamma}{\alpha^2}\right)^{j}$ for $j \in [0, t-1]$. We claim that, for sufficiently large $T$, $\tilde{\boldsymbol{u}}^{T+1}$ approaches the left PCA estimate $\boldsymbol{u}_{\text{PCA}}$, that is, $\lim_{T\to\infty}\lim_{n\to\infty}\frac{1}{\sqrt{m}}\|\tilde{\boldsymbol{u}}^{T+1} - \sqrt{m}\boldsymbol{u}_{\text{PCA}}\| = 0$. This result is proved in Lemma C.8 in Appendix C.4. Here we give a heuristic sanity check. Assume that the iterates $\tilde{\boldsymbol{u}}^{T+1}$ and $\tilde{\boldsymbol{v}}^{T+1}$ converge to the limits $\tilde{\boldsymbol{u}}^\infty$ and $\tilde{\boldsymbol{v}}^\infty$, respectively, in the sense that $\lim_{T\to\infty}\lim_{n\to\infty}\frac{1}{\sqrt{m}}\|\tilde{\boldsymbol{u}}^{T+1} - \tilde{\boldsymbol{u}}^\infty\| = 0$ and $\lim_{T\to\infty}\lim_{n\to\infty}\frac{1}{\sqrt{n}}\|\tilde{\boldsymbol{v}}^{T+1} - \tilde{\boldsymbol{v}}^\infty\| = 0$. Then, from (C.2), the limits $\tilde{\boldsymbol{u}}^\infty$ and $\tilde{\boldsymbol{v}}^\infty$ satisfy

$$\tilde{\boldsymbol{u}}^\infty = \frac{1}{\alpha}\boldsymbol{X}\tilde{\boldsymbol{v}}^\infty - \sum_{i=1}^{\infty}\kappa_{2i}\left(\frac{\gamma}{\alpha^2}\right)^i\tilde{\boldsymbol{u}}^\infty,$$

$$\tilde{\boldsymbol{v}}^\infty = \frac{\gamma}{\alpha}\boldsymbol{X}^\mathsf{T}\tilde{\boldsymbol{u}}^\infty - \gamma\sum_{i=1}^{\infty}\kappa_{2i}\left(\frac{\gamma}{\alpha^2}\right)^i\tilde{\boldsymbol{v}}^\infty. \tag{C.3}$$

By using (A.15), we can re-write (C.3) as

$$\left(1 + R\left(\frac{\gamma}{\alpha^2}\right)\right)\tilde{\boldsymbol{u}}^\infty = \frac{1}{\alpha}\boldsymbol{X}\tilde{\boldsymbol{v}}^\infty,$$

$$\left(1 + \gamma R\left(\frac{\gamma}{\alpha^2}\right)\right)\tilde{\boldsymbol{v}}^\infty = \frac{\gamma}{\alpha}\boldsymbol{X}^\mathsf{T}\tilde{\boldsymbol{u}}^\infty, \tag{C.4}$$

which leads to

$$\left(1 + \gamma R\left(\frac{\gamma}{\alpha^2}\right)\right)\left(1 + R\left(\frac{\gamma}{\alpha^2}\right)\right)\tilde{\boldsymbol{u}}^\infty = \frac{\gamma}{\alpha^2}\boldsymbol{X}\boldsymbol{X}^\mathsf{T}\tilde{\boldsymbol{u}}^\infty. \tag{C.5}$$

As a result, $\tilde{\boldsymbol{u}}^\infty$ is an eigenvector of $\boldsymbol{X}\boldsymbol{X}^\mathsf{T}$. Furthermore, by using (A.19), the eigenvalue $\frac{\alpha^2}{\gamma}\left(1 + \gamma R\left(\frac{\gamma}{\alpha^2}\right)\right)\left(1 + R\left(\frac{\gamma}{\alpha^2}\right)\right)$ can be re-written as $\left(D^{-1}\left(\frac{\gamma}{\alpha^2}\right)\right)^2$. Recall that, for $\tilde{\alpha} > \tilde{\alpha}_{\text{s}}$, $\boldsymbol{X}$ exhibits a spectral gap and its largest singular value converges to $D^{-1}\left(\frac{\gamma}{\alpha^2}\right)$. Thus, $\boldsymbol{u}^\infty$ must be aligned with the left principal singular vector of $\boldsymbol{X}$, as desired.

A key step in our analysis is to show that, as $T \to \infty$, the state evolution of the artificial AMP in the first phase has a unique fixed point. This is established in Lemma C.2, proved in Appendix C.3. As for the square case, we follow the approach of [20, Section 7]. The crucial difference with [20] is that we provide a result for all $\tilde{\alpha} > \tilde{\alpha}_{\text{s}}$, while the analysis of [20] requires that $\tilde{\alpha}$ is sufficiently large. To achieve this goal, we exploit the expression (2.3) of the limit correlation between $\boldsymbol{u}_{\text{PCA}}$ and $\boldsymbol{u}^*$, and show that, as soon as the left PCA estimate is correlated with the signal $\boldsymbol{u}^*$, state evolution is close to a limit map which is a contraction. For this approach to work, we need the rectangular free cumulants to be non-negative.

**Second phase.** The second phase is designed so that the iterates $(\tilde{\boldsymbol{g}}^{T+k}, \tilde{\boldsymbol{f}}^{T+k})$ are close to $(\boldsymbol{g}^k, \boldsymbol{f}^k)$, for $k \geq 2$. For $t \geq (T+2)$, the artificial AMP computes

$$\tilde{\boldsymbol{u}}^t = \mathsf{u}_{t-T}(\tilde{\boldsymbol{f}}^{t-1}), \qquad \tilde{\boldsymbol{g}}^t = \boldsymbol{X}^\mathsf{T} \tilde{\boldsymbol{u}}^t - \sum_{i=1}^{t-1} \tilde{\mathsf{b}}_{t,i} \tilde{\boldsymbol{v}}^i,$$

$$\tilde{\boldsymbol{v}}^t = \mathsf{v}_{t-T}(\tilde{\boldsymbol{g}}^t), \qquad \tilde{\boldsymbol{f}}^t = \boldsymbol{X} \tilde{\boldsymbol{v}}^t - \sum_{i=1}^{t} \tilde{\mathsf{a}}_{t,i} \tilde{\boldsymbol{u}}^t. \tag{C.6}$$

Here, the functions $\{v_k, u_k\}_{k \geq 2}$ are the ones used in the true AMP (3.10). Additionally, letting $u_1(x) = x/\alpha$ and $v_1(x) = \gamma x/\alpha$, the coefficients $\{\tilde{\mathsf{a}}_{t,i}\}$ and $\{\tilde{\mathsf{b}}_{t,i}\}$ are given by:

$$\tilde{\mathsf{a}}_{t,t-j} = \kappa_{2(j+1)} \langle \mathsf{v}'_{t-T}(\tilde{\boldsymbol{g}}^t) \rangle \left( \frac{\gamma}{\alpha^2} \right)^{(T+1-(t-j))_+} \prod_{i=\max\{t-j+1, T+2\}}^{t} \langle \mathsf{u}'_{i-T}(\tilde{\boldsymbol{f}}^{i-1}) \rangle \langle \mathsf{v}'_{i-1-T}(\tilde{\boldsymbol{g}}^{i-1}) \rangle,$$

$$(t-j) \in [1, t], \tag{C.7}$$

$$\tilde{\mathsf{b}}_{t,t-j} = \gamma \kappa_{2j} \langle \mathsf{u}'_{t-T}(\tilde{\boldsymbol{f}}^{t-1}) \rangle \left( \frac{\gamma}{\alpha^2} \right)^{(T-(t-j))_+} \prod_{i=\max\{t-j+1, T+1\}}^{t-1} \langle \mathsf{v}'_{i-T}(\tilde{\boldsymbol{g}}^{i}) \rangle \langle \mathsf{u}'_{i-T}(\tilde{\boldsymbol{f}}^{i-1}) \rangle,$$

$$(t-j) \in [1, t-1]. \tag{C.8}$$

Since the artificial AMP is initialized with $\tilde{\boldsymbol{u}}^1$ that is correlated with $\boldsymbol{u}^*$ and independent of the noise matrix $\boldsymbol{W}$, a state evolution result for it can be obtained directly from [20, Theorem 1.4]. We then show in Lemma C.9 in Appendix C.5 that the second phase iterates in (C.6) are close to the true AMP iterates in (3.10), and that their state evolution parameters are also close. This result yields Theorem 2, as shown in Appendix C.6.

## C.2 State Evolution for the Artificial AMP

Consider the artificial AMP iteration defined in (C.2) and (C.6), with initialization $\tilde{\boldsymbol{u}}^1 = \sqrt{\Delta_{\mathrm{PCA}}} \boldsymbol{u}^* + \sqrt{1 - \Delta_{\mathrm{PCA}}} \boldsymbol{n}$. Then, its associated state evolution recursion is expressed in terms of a sequence of mean vectors $\tilde{\boldsymbol{\mu}}_K = (\tilde{\mu}_t)_{t \in [0,K]}$, $\tilde{\boldsymbol{\nu}}_K = (\tilde{\nu}_t)_{t \in [1,K]}$ and covariance matrices $\tilde{\boldsymbol{\Sigma}}_K = (\tilde{\sigma}_{s,t})_{s,t \in [0,K]}$, $\tilde{\boldsymbol{\Omega}}_K = (\tilde{\Omega}_{s,t})_{s,t \in [1,K]}$ defined recursively as follows. We initialize with

$$\tilde{\mu}_0 = \alpha \sqrt{\Delta_{\mathrm{PCA}}}, \qquad \tilde{\sigma}_{0,0} = \alpha^2 (1 - \Delta_{\mathrm{PCA}}), \quad \tilde{\sigma}_{0,t} = \tilde{\sigma}_{t,0} = 0, \quad \text{for } t \geq 1. \tag{C.9}$$

Given $\tilde{\boldsymbol{\mu}}_K, \tilde{\boldsymbol{\Sigma}}_K, \tilde{\boldsymbol{\nu}}_K, \tilde{\boldsymbol{\Omega}}_K$, let

$$(\tilde{F}_0, \dots, \tilde{F}_K) = \tilde{\boldsymbol{\mu}}_K U_* + (\tilde{Y}_0, \dots, \tilde{Y}_K), \quad \text{where } (\tilde{Y}_0, \dots, \tilde{Y}_K) \sim \mathcal{N}(\boldsymbol{0}, \tilde{\boldsymbol{\Sigma}}_K), \tag{C.10}$$

$$\tilde{U}_t = \tilde{\mathsf{u}}_t(\tilde{F}_{t-1}) \quad \text{where } \tilde{\mathsf{u}}_t(x) = \begin{cases} x/\alpha, & 1 \leq t \leq (T+1), \\ \mathsf{u}_{t-T}(x), & t \geq T+2, \end{cases} \tag{C.11}$$

$$(\tilde{G}_1, \dots, \tilde{G}_K) = \tilde{\boldsymbol{\nu}}_K V_* + (\tilde{Z}_1, \dots, \tilde{Z}_K), \quad \text{where } (\tilde{Z}_1, \dots, \tilde{Z}_K) \sim \mathcal{N}(\boldsymbol{0}, \tilde{\boldsymbol{\Omega}}_K), \tag{C.12}$$

$$\tilde{V}_t = \tilde{\mathsf{v}}_t(\tilde{G}_t) \quad \text{where } \tilde{\mathsf{v}}_t(x) = \begin{cases} \gamma x/\alpha, & 1 \leq t \leq T+1, \\ \mathsf{v}_{t-T}(x), & t \geq T+2. \end{cases} \tag{C.13}$$

Given $\tilde{\boldsymbol{\mu}}_K$ and $\tilde{\boldsymbol{\Sigma}}_K$, the entries of $\tilde{\boldsymbol{\nu}}_{K+1}$ are given by $\tilde{\nu}_t = \alpha \mathbb{E}\{\tilde{U}_t U_*\}$ (for $t \in [1, K+1]$), and the entries of $\tilde{\boldsymbol{\Omega}}_{K+1}$ (for $s+1, t+1 \in [1, K+1]$) are given by

$$\tilde{\omega}_{s+1,t+1} = \gamma \sum_{j=0}^{s} \sum_{k=0}^{t} \left( \prod_{i=s-j+2}^{s+1} \mathbb{E}\{\tilde{\mathsf{u}}'_i(\tilde{F}_{i-1})\} \mathbb{E}\{\tilde{\mathsf{v}}'_{i-1}(\tilde{G}_{i-1})\} \right) \left( \prod_{i=t-k+2}^{t+1} \mathbb{E}\{\tilde{\mathsf{u}}'_i(\tilde{F}_{i-1})\} \mathbb{E}\{\tilde{\mathsf{v}}'_{i-1}(\tilde{G}_{i-1})\} \right)$$

$$\left[ \kappa^\infty_{2(j+k+1)} \mathbb{E}\{\tilde{U}_{s+1-j} \tilde{U}_{t+1-k}\} + \kappa^\infty_{2(j+k+2)} \mathbb{E}\{\tilde{\mathsf{u}}'_{s+1-j}(\tilde{F}_{s-j})\} \mathbb{E}\{\tilde{\mathsf{u}}'_{t+1-k}(\tilde{F}_{t-k})\} \mathbb{E}\{\tilde{V}_{s-j} \tilde{V}_{t-k}\} \right]. \tag{C.14}$$

(We use the convention that $\tilde{V}_0 = 0$.) Next, given $\tilde{\boldsymbol{\nu}}_{K+1}$ and $\tilde{\boldsymbol{\Omega}}_{K+1}$ for some $K \geq 1$, the entries of $\tilde{\boldsymbol{\mu}}_{K+1}$ are given by $\tilde{\mu}_t = \frac{\alpha}{\gamma} \mathbb{E}\{\tilde{V}_t V_*\}$ (for $t \in [0, K+1]$), and the entries of $\tilde{\boldsymbol{\Sigma}}_{K+1}$ (for $s, t \in$

$[0, K+1])$ are given by

$$
\begin{aligned}
\tilde{\sigma}_{s,t} = \sum_{j=0}^{s-1} \sum_{k=0}^{t-1} & \left( \prod_{i=s-j+1}^{s} \mathbb{E}\{\tilde{u}_i'(\tilde{F}_{i-1})\} \mathbb{E}\{\tilde{v}_i'(\tilde{G}_i)\} \right) \left( \prod_{i=t-k+1}^{t} \mathbb{E}\{\tilde{u}_i'(\tilde{F}_{i-1})\} \mathbb{E}\{\tilde{v}_i'(\tilde{G}_i)\} \right) \\
& \cdot \left[ \kappa_{2(j+k+1)}^{\infty} \mathbb{E}\{\tilde{V}_{s-j} \tilde{V}_{t-k}\} + \kappa_{2(j+k+2)}^{\infty} \mathbb{E}\{\tilde{v}_{s-j}'(\tilde{G}_{s-j})\} \mathbb{E}\{\tilde{v}_{t-k}'(\tilde{G}_{t-k})\} \mathbb{E}\{\tilde{U}_{s-j} \tilde{U}_{t-k}\} \right].
\end{aligned}
\tag{C.15}
$$

**Proposition C.1** (State evolution for artificial AMP). *Consider the setting of Theorem 2, the artificial AMP iteration described in* (C.2) *and* (C.6)*, with initialization given by* (C.1)*, and the corresponding state evolution parameters defined in* (C.9)-(C.15)*.*

*Then, for $t \geq 1$ and any PL(2) functions $\psi : \mathbb{R}^{2t+2} \to \mathbb{R}$ and $\varphi : \mathbb{R}^{2t+1} \to \mathbb{R}$, the following hold almost surely:*

$$
\lim_{m \to \infty} \frac{1}{m} \sum_{i=1}^{m} \psi(u_i^*, \tilde{u}_i^1, \dots, \tilde{u}_i^{t+1}, \tilde{f}_i^1, \dots \tilde{f}_i^t) = \mathbb{E}\left\{ \psi(U_*, \tilde{U}_1, \dots, \tilde{U}_{t+1}, \tilde{F}_1, \dots, \tilde{F}_t) \right\}, \tag{C.16}
$$

$$
\lim_{n \to \infty} \frac{1}{n} \sum_{i=1}^{n} \varphi(v_i^*, \tilde{v}_i^1, \dots, \tilde{v}_i^t, \tilde{g}_i^1, \dots \tilde{g}_i^t) = \mathbb{E}\left\{ \varphi(V_*, \tilde{V}_1, \dots, \tilde{V}_t, \tilde{G}_1, \dots, \tilde{G}_t) \right\}. \tag{C.17}
$$

The proposition follows directly from Theorem 1.4 in [20] since the initialization $\tilde{u}^1$ of the artificial AMP is independent of $W$.

### C.3 Fixed Point of State Evolution for the First Phase

From (C.9)-(C.15), we note that the state evolution recursion for the first phase ($t \in [1, T+1]$) has the following form:

$$
\tilde{\mu}_t = \tilde{\nu}_t = \alpha \sqrt{\Delta_{\text{PCA}}}, \quad \text{for } t \in [1, T+1],
$$

$$
\begin{aligned}
\tilde{\sigma}_{s,t} = \sum_{j=0}^{s-1} \sum_{k=0}^{t-1} \left( \frac{\gamma}{\alpha^2} \right)^{j+k} & \left( \kappa_{2(j+k+1)}^{\infty} \left( \frac{\gamma}{\alpha} \right)^2 (\alpha^2 \Delta_{\text{PCA}} + \tilde{\omega}_{s-j,t-k}) \right. \\
& \left. + \kappa_{2(j+k+2)}^{\infty} \left( \frac{\gamma}{\alpha^2} \right)^2 (\alpha^2 \Delta_{\text{PCA}} + \tilde{\sigma}_{s-j-1,t-k-1}) \right), \quad \text{for } s, t \in [1, T+1].
\end{aligned}
\tag{C.18}
$$

$$
\begin{aligned}
\tilde{\omega}_{s,t} = \gamma \sum_{j=0}^{s-1} \sum_{k=0}^{t-1} \left( \frac{\gamma}{\alpha^2} \right)^{j+k} & \left( \kappa_{2(j+k+1)}^{\infty} \frac{1}{\alpha^2} (\alpha^2 \Delta_{\text{PCA}} + \tilde{\sigma}_{s-j-1,t-k-1}) \right. \\
& \left. + \kappa_{2(j+k+2)}^{\infty} \left( \frac{\gamma}{\alpha^2} \right)^2 (\alpha^2 \Delta_{\text{PCA}} + \tilde{\omega}_{s-j-1,t-k-1}) \right), \quad \text{for } s, t \in [1, T+1].
\end{aligned}
$$

In this section, we prove the following result characterizing the fixed point of state evolution for the first phase in the rectangular setting.

**Lemma C.2** (Fixed point of state evolution for first phase – Rectangular matrices). *Consider the setting of Theorem 2, and the state evolution recursion for the first phase given by* (C.18)*. Assume that $\kappa_{2i}^{\infty} \geq 0$ for all $i \geq 2$, and that $\tilde{\alpha} > \tilde{\alpha}_s$. Pick any $\xi < 1$ such that $\tilde{\alpha}\sqrt{\xi} > \tilde{\alpha}_s$. Then,*

$$
\lim_{T \to \infty} \max_{s,t \in [0,T]} \xi^{\max(s,t)} |\tilde{\sigma}_{T+1-s,T+1-t} - a^*| = 0,
$$
$$
\lim_{T \to \infty} \max_{s,t \in [0,T]} \xi^{\max(s,t)} |\tilde{\omega}_{T+1-s,T+1-t} - b^*| = 0,
\tag{C.19}
$$

*where*

$$
a^* = \alpha^2 (1 - \Delta_{\text{PCA}}),
$$
$$
b^* = \frac{\Delta_{\text{PCA}} \gamma \alpha^2 (x R'(x) - R(x)) + \gamma R'(x)}{1 + \gamma R(x) - \gamma x R'(x)}, \quad \text{with } x = \frac{\gamma}{\alpha^2}.
\tag{C.20}
$$

As for the case of square matrices, we consider the space of infinite matrices $\boldsymbol{x} = (x_{s,t} : s, t \leq 0)$ equipped with the weighted $\ell_\infty$-norm defined in (B.8). Let $\mathcal{X} = \{\boldsymbol{x} : \|\boldsymbol{x}\|_\xi < \infty\}$ and, for any compact set $I \subset \mathbb{R}$, define $\mathcal{X}_I$ as in (B.9). Recall that both $\mathcal{X}$ and $\mathcal{X}_I$ are complete under $\|\cdot\|_\xi$. We embed the matrices $\tilde{\boldsymbol{\Sigma}}_{\bar{T}}, \tilde{\boldsymbol{\Omega}}_{\bar{T}}$ as elements $\boldsymbol{x}, \boldsymbol{y} \in \mathcal{X}$ with the following coordinate identification:

$$\tilde{\sigma}_{s,t} = x_{s-\bar{T},t-\bar{T}}, \quad \tilde{\omega}_{s,t} = y_{s-\bar{T},t-\bar{T}},$$
$$x_{s,t} = 0, \quad y_{s,t} = 0, \quad \text{if } s < -\bar{T} \text{ or } t < -\bar{T}$$

The idea is to approximate the maps $(\tilde{\boldsymbol{\Sigma}}_{\bar{T}-1}, \tilde{\boldsymbol{\Omega}}_{\bar{T}-1}) \mapsto \tilde{\boldsymbol{\Omega}}_{\bar{T}}$ and $(\tilde{\boldsymbol{\Sigma}}_{\bar{T}-1}, \tilde{\boldsymbol{\Omega}}_{\bar{T}}) \mapsto \tilde{\boldsymbol{\Sigma}}_{\bar{T}}$ with the *fixed limit* maps $h^\Sigma$ and $h^\Omega$, respectively, which are defined as

$$
\begin{aligned}
h^\Omega_{s,t}(\boldsymbol{x}, \boldsymbol{y}) = \gamma \sum_{j=0}^\infty \sum_{k=0}^\infty \left(\frac{\gamma}{\alpha^2}\right)^{j+k} &\left( \kappa^\infty_{2(j+k+1)} \frac{1}{\alpha^2} (\alpha^2 \Delta_{\mathrm{PCA}} + x_{s-j,t-k}) \right. \\
&\left. + \kappa^\infty_{2(j+k+2)} \left(\frac{\gamma}{\alpha^2}\right)^2 (\alpha^2 \Delta_{\mathrm{PCA}} + y_{s-j,t-k}) \right), \\
h^\Sigma_{s,t}(\boldsymbol{x}, \boldsymbol{y}) = \sum_{j=0}^\infty \sum_{k=0}^\infty \left(\frac{\gamma}{\alpha^2}\right)^{j+k} &\left( \kappa^\infty_{2(j+k+1)} \left(\frac{\gamma}{\alpha}\right)^2 (\alpha^2 \Delta_{\mathrm{PCA}} + y_{s-j,t-k}) \right. \\
&\left. + \kappa^\infty_{2(j+k+2)} \left(\frac{\gamma}{\alpha^2}\right)^2 (\alpha^2 \Delta_{\mathrm{PCA}} + x_{s-j,t-k}) \right).
\end{aligned}
\tag{C.21}
$$

First, we show that $(h^\Omega(\mathcal{X}_{I^*_\Sigma}, \mathcal{X}_{I^*_\Omega}), h^\Sigma(\mathcal{X}_{I^*_\Sigma}, \mathcal{X}_{I^*_\Omega})) \subseteq (\mathcal{X}_{I^*_\Omega}, \mathcal{X}_{I^*_\Sigma})$ for suitably defined compact sets $I^*_\Omega, I^*_\Sigma$.

**Lemma C.3** (Image of limit maps – Rectangular matrices). *Consider the maps $h^\Omega, h^\Sigma$ defined in (C.21). Assume that $\kappa^\infty_{2i} \geq 0$ for all $i \geq 1$, and that $\tilde{\alpha} > \tilde{\alpha}_s$. Then, there exist $I^*_\Omega = [-a_\Omega, a_\Omega]$ and $I^*_\Sigma = [-a_\Sigma, a_\Sigma]$ such that, if $(\boldsymbol{x}, \boldsymbol{y}) \in \mathcal{X}_{I^*_\Sigma} \times \mathcal{X}_{I^*_\Omega}$, then $(h^\Omega(\boldsymbol{x}, \boldsymbol{y}), h^\Sigma(\boldsymbol{x}, \boldsymbol{y})) \in \mathcal{X}_{I^*_\Omega} \times \mathcal{X}_{I^*_\Sigma}$.*

*Proof.* Let $(\boldsymbol{x}, \boldsymbol{y}) \in \mathcal{X}_{I^*_\Sigma} \times \mathcal{X}_{I^*_\Omega}$. Then, the following chain of inequalities holds:

$$
\begin{aligned}
|h^\Omega_{s,t}(\boldsymbol{x}, \boldsymbol{y})| &\overset{(a)}{\leq} \gamma \sum_{j=0}^\infty \sum_{k=0}^\infty \left(\frac{\gamma}{\alpha^2}\right)^{j+k} \left( \kappa^\infty_{2(j+k+1)} \frac{1}{\alpha^2} (\alpha^2 \Delta_{\mathrm{PCA}} + |x_{s-j,t-k}|) \right. \\
&\qquad\qquad\qquad \left. + \kappa^\infty_{2(j+k+2)} \left(\frac{\gamma}{\alpha^2}\right)^2 (\alpha^2 \Delta_{\mathrm{PCA}} + |y_{s-j,t-k}|) \right) \\
&\overset{(b)}{\leq} \gamma \sum_{j=0}^\infty \sum_{k=0}^\infty \left(\frac{\gamma}{\alpha^2}\right)^{j+k} \left( \kappa^\infty_{2(j+k+1)} \frac{1}{\alpha^2} (\alpha^2 \Delta_{\mathrm{PCA}} + a_\Sigma) \right. \\
&\qquad\qquad\qquad \left. + \kappa^\infty_{2(j+k+2)} \left(\frac{\gamma}{\alpha^2}\right)^2 (\alpha^2 \Delta_{\mathrm{PCA}} + a_\Omega) \right) \\
&\overset{(c)}{=} \gamma \left( \left(\Delta_{\mathrm{PCA}} + \frac{a_\Sigma}{\alpha^2}\right) R'\left(\frac{\gamma}{\alpha^2}\right) + (\alpha^2 \Delta_{\mathrm{PCA}} + a_\Omega) \left(\frac{\gamma}{\alpha^2} R'\left(\frac{\gamma}{\alpha^2}\right) - R\left(\frac{\gamma}{\alpha^2}\right)\right) \right).
\end{aligned}
\tag{C.22}
$$

Here, (a) follows from the hypothesis that $\kappa^\infty_i \geq 0$ for $i \geq 2$; (b) holds since $(\boldsymbol{x}, \boldsymbol{y}) \in \mathcal{X}_{I^*_\Sigma} \times \mathcal{X}_{I^*_\Omega}$; and (c) uses (A.16)-(A.17). With similar passages, we also obtain that

$$
|h^\Sigma_{s,t}(\boldsymbol{x}, \boldsymbol{y})| \leq \left( \gamma^2 \Delta_{\mathrm{PCA}} + \frac{\gamma^2 a_\Omega}{\alpha^2} \right) R'\left(\frac{\gamma}{\alpha^2}\right) + (\alpha^2 \Delta_{\mathrm{PCA}} + a_\Sigma) \left(\frac{\gamma}{\alpha^2} R'\left(\frac{\gamma}{\alpha^2}\right) - R\left(\frac{\gamma}{\alpha^2}\right)\right).
\tag{C.23}
$$

Set $x = \gamma/\alpha^2$. Then, by using (C.22) and (C.23), we obtain that the desired result holds if the following pair of inequalities is satisfied:

$$
\begin{aligned}
\Delta_{\mathrm{PCA}}(\gamma R'(x) + \gamma\alpha^2(xR'(x) - R(x))) + a_\Sigma x R'(x) + a_\Omega \gamma (xR'(x) - R(x)) &\leq a_\Omega, \\
\Delta_{\mathrm{PCA}}(\gamma^2 R'(x) + \alpha^2(xR'(x) - R(x))) + a_\Sigma(xR'(x) - R(x)) + a_\Omega \gamma x R'(x) &\leq a_\Sigma.
\end{aligned}
\tag{C.24}
$$

Set $\beta = a_\Sigma/a_\Omega$. Then, (C.24) can be rewritten as

$$
\begin{aligned}
\Delta_{\mathrm{PCA}}(\gamma R'(x) + \gamma\alpha^2(xR'(x) - R(x))) + a_\Omega (\beta x R'(x) + \gamma(xR'(x) - R(x))) &\leq a_\Omega, \\
\Delta_{\mathrm{PCA}}(\gamma^2 R'(x) + \alpha^2(xR'(x) - R(x))) + a_\Omega (\beta(xR'(x) - R(x)) + \gamma x R'(x)) &\leq \beta a_\Omega.
\end{aligned}
$$

This pair of inequalities holds for a sufficiently large $a_\Omega$ if

$$\beta x R'(x) + \gamma(x R'(x) - R(x)) < 1,$$
$$\beta(x R'(x) - R(x)) + \gamma x R'(x) < \beta. \tag{C.25}$$

Recall that, above the spectral threshold, namely, when $\tilde{\alpha} > \tilde{\alpha}_s$, the PCA estimator $\boldsymbol{u}_{\mathrm{PCA}}$ has strictly positive correlation with the signal $\boldsymbol{u}^*$:

$$\frac{\langle \boldsymbol{u}_{\mathrm{PCA}}, \boldsymbol{u}^* \rangle^2}{n} \xrightarrow{\text{a.s.}} \Delta_{\mathrm{PCA}} > 0.$$

Furthermore, from [20, Eq. (7.32)], we have that $\Delta_{\mathrm{PCA}}$ can be expressed as

$$\Delta_{\mathrm{PCA}} = \frac{T(R(x)) - x T'(R(x)) R'(x)}{1 + \gamma R(x)},$$

where $T(z) = (1 + z)(1 + \gamma z)$. We therefore obtain that

$$T(R(x)) - x T'(R(x)) R'(x) > 0. \tag{C.26}$$

By using (C.26), one can readily verify that $1 - x R'(x) + R(x) > 0$. Furthermore, we have that $x R'(x) > 0$, as $x > 0$ and the rectangular free cumulants are non-negative. Since $x R'(x) > 0$ and $1 - x R'(x) + R(x) > 0$, (C.25) can be rewritten as

$$\frac{\gamma x R'(x)}{1 - x R'(x) + R(x)} < \beta < \frac{1 - \gamma x R'(x) + \gamma R(x)}{x R'(x)}.$$

These above inequalities can be simultaneously satisfied for some value of $\beta$ if

$$\frac{\gamma x R'(x)}{1 - x R'(x) + R(x)} < \frac{1 - \gamma x R'(x) + \gamma R(x)}{x R'(x)}. \tag{C.27}$$

By using again that $x R'(x) > 0$ and $1 - x R'(x) + R(x) > 0$, (C.27) can be rewritten as

$$1 - (1 + \gamma)(x R'(x) - R(x)) + \gamma(x R'(x) - R(x))^2 > \gamma(x R'(x))^2. \tag{C.28}$$

The inequality (C.28) can be readily obtained from (C.26), and the proof is complete. $\qquad\square$

Next, we compute a fixed point of $(h^\Sigma, h^\Omega)$.

**Lemma C.4** (Fixed point of limit maps – Rectangular matrices)**.** *Consider the maps $h^\Omega, h^\Sigma$ defined in (C.21). Let $\boldsymbol{x}^* = (x^*_{s,t} : s, t \leq 0)$ and $\boldsymbol{y}^* = (y^*_{s,t} : s, t \leq 0)$ with $x^*_{s,t} = a^*$ and $y^*_{s,t} = b^*$, where $a^*$ and $b^*$ are defined in (C.20). Assume that $\tilde{\alpha} > \tilde{\alpha}_s$. Then, $(\boldsymbol{x}^*, \boldsymbol{y}^*)$ is a fixed point of $(h^\Sigma, h^\Omega)$.*

*Proof.* Note that, for $z = \gamma/\alpha^2$, the power series expansion (A.16) of $R'$ converges to a finite limit as $\tilde{\alpha} > \tilde{\alpha}_s$. Hence, by using the definition (C.21), we have that

$$h^\Omega_{s,t}(\boldsymbol{x}^*, \boldsymbol{y}^*) = \gamma \left( \left( \Delta_{\mathrm{PCA}} + \frac{a^*}{\alpha^2} \right) R'\left( \frac{\gamma}{\alpha^2} \right) + (\alpha^2 \Delta_{\mathrm{PCA}} + b^*) \left( \frac{\gamma}{\alpha^2} R'\left( \frac{\gamma}{\alpha^2} \right) - R\left( \frac{\gamma}{\alpha^2} \right) \right) \right),$$

$$h^\Sigma_{s,t}(\boldsymbol{x}^*, \boldsymbol{y}^*) = \left( \gamma^2 \Delta_{\mathrm{PCA}} + \frac{\gamma^2 b^*}{\alpha^2} \right) R'\left( \frac{\gamma}{\alpha^2} \right) + (\alpha^2 \Delta_{\mathrm{PCA}} + a^*) \left( \frac{\gamma}{\alpha^2} R'\left( \frac{\gamma}{\alpha^2} \right) - R\left( \frac{\gamma}{\alpha^2} \right) \right). \tag{C.29}$$

Since a fixed point should satisfy $h^\Omega_{s,t}(\boldsymbol{x}^*, \boldsymbol{y}^*) = b^*$ and $h^\Sigma_{s,t}(\boldsymbol{x}^*, \boldsymbol{y}^*) = a^*$, writing $x = \gamma/\alpha^2$, (C.29) becomes

$$\begin{cases} \gamma \Delta_{\mathrm{PCA}}(R'(x) + \alpha^2(x R'(x) - R(x))) + a^* x R'(x) + b^* \gamma(x R'(x) - R(x)) = b^*, \\ \Delta_{\mathrm{PCA}}(\gamma^2 R'(x) + \alpha^2(x R'(x) - R(x))) + a^*(x R'(x) - R(x)) + b^* \gamma x R'(x) = a^*. \end{cases} \tag{C.30}$$

Solving (C.30) for $a^*$ and $b^*$, and using the expression for $\Delta_{\mathrm{PCA}}$ given in [20, Eq. (7.32)], we obtain the formulas for $(a^*, b^*)$ given in (C.20). $\qquad\square$

The next step is to show Lipschitz bounds on the maps $h^\Sigma, h^\Omega$.

**Lemma C.5** (Lipschitz bounds on limit maps). *Consider the map $(h^\Omega(\boldsymbol{x}, \boldsymbol{y}), h^\Sigma(\boldsymbol{x}, \boldsymbol{y})) : \mathcal{X}_{I_\Omega^*} \times \mathcal{X}_{I_\Sigma^*} \to \mathcal{X}_{I_\Omega^*} \times \mathcal{X}_{I_\Sigma^*}$ defined in (C.21) and where $I_\Omega^*$, $I_\Sigma^*$ are given by Lemma C.3. Assume that $\kappa_{2i}^\infty \geq 0$ for all $i \geq 1$, and let $\xi < 1$ be such that $\tilde{\alpha}\sqrt{\xi} > \tilde{\alpha}_s$. Then, for any $(\boldsymbol{x}, \boldsymbol{y}) \in \mathcal{X}_{I_\Sigma^*} \times \mathcal{X}_{I_\Omega^*}$,*

$$\|h^\Omega(\boldsymbol{x}, \boldsymbol{y}) - h^\Omega(\boldsymbol{x}', \boldsymbol{y}')\|_\xi \leq \tilde{x}R'(\tilde{x})\|\boldsymbol{x} - \boldsymbol{x}'\|_\xi + \gamma\left(\tilde{x}R'(\tilde{x}) - R(\tilde{x})\right)\|\boldsymbol{y} - \boldsymbol{y}'\|_\xi, \qquad \text{(C.31)}$$

$$\|h^\Sigma(\boldsymbol{x}, \boldsymbol{y}) - h^\Sigma(\boldsymbol{x}', \boldsymbol{y}')\|_\xi \leq \gamma\tilde{x}R'(\tilde{x})\|\boldsymbol{y} - \boldsymbol{y}'\|_\xi + \left(\tilde{x}R'(\tilde{x}) - R(\tilde{x})\right)\|\boldsymbol{x} - \boldsymbol{x}'\|_\xi, \qquad \text{(C.32)}$$

*where we have set $\tilde{x} = \gamma/(\xi\alpha^2)$.*

*Proof.* Since $\kappa_{2i}^\infty \geq 0$ for $i \geq 1$, we have

$$|h_{s,t}^\Omega(\boldsymbol{x}, \boldsymbol{y}) - h_{s,t}^\Omega(\boldsymbol{x}', \boldsymbol{y}')| \leq \gamma \sum_{j=0}^\infty \sum_{k=0}^\infty \left(\frac{\gamma}{\alpha^2}\right)^{j+k} \left(\kappa_{2(j+k+1)}^\infty \frac{1}{\alpha^2}|x_{s-j,t-k} - x'_{s-j,t-k}| \right.$$
$$\left. + \kappa_{2(j+k+2)}^\infty \frac{\gamma^2}{\alpha^4}|y_{s-j,t-k} - y'_{s-j,t-k}|\right). \qquad \text{(C.33)}$$

Note that

$$|x_{s-j,t-k} - x'_{s-j,t-k}| \leq \|\boldsymbol{x} - \boldsymbol{x}'\|_\xi \xi^{-\max(|s-j|,|t-k|)} \leq \|\boldsymbol{x} - \boldsymbol{x}'\|_\xi \xi^{-\max(|s|,|t|)-j-k},$$
$$|y_{s-j,t-k} - y'_{s-j,t-k}| \leq \|\boldsymbol{y} - \boldsymbol{y}'\|_\xi \xi^{-\max(|s-j|,|t-k|)} \leq \|\boldsymbol{y} - \boldsymbol{y}'\|_\xi \xi^{-\max(|s|,|t|)-j-k}. \qquad \text{(C.34)}$$

Thus, by combining (C.33) and (C.34), we have

$$\|h^\Omega(\boldsymbol{x}, \boldsymbol{y}) - h^\Omega(\boldsymbol{x}', \boldsymbol{y}')\|_\xi \leq \frac{\gamma}{\alpha^2} \sum_{j=0}^\infty \sum_{k=0}^\infty \left(\frac{\gamma}{\xi\alpha^2}\right)^{j+k} \left(\kappa_{2(j+k+1)}^\infty \|\boldsymbol{x} - \boldsymbol{x}'\|_\xi + \kappa_{2(j+k+2)}^\infty \frac{\gamma^2}{\alpha^2}\|\boldsymbol{y} - \boldsymbol{y}'\|_\xi\right). \qquad \text{(C.35)}$$

By using (A.16) and (A.17) to compute the sums in (C.35), we deduce that

$$\|h^\Omega(\boldsymbol{x}, \boldsymbol{y}) - h^\Omega(\boldsymbol{x}', \boldsymbol{y}')\|_\xi \leq \frac{\gamma}{\alpha^2} R'\left(\frac{\gamma}{\xi\alpha^2}\right)\|\boldsymbol{x} - \boldsymbol{x}'\|_\xi$$
$$+ \xi^2 \gamma \left(\frac{\gamma}{\xi\alpha^2}R'\left(\frac{\gamma}{\xi\alpha^2}\right) - R\left(\frac{\gamma}{\xi\alpha^2}\right)\right)\|\boldsymbol{y} - \boldsymbol{y}'\|_\xi. \qquad \text{(C.36)}$$

Recall that $\xi < 1$ and note from (A.17) that $\tilde{x}R'(\tilde{x}) \geq R(\tilde{x}) \geq 0$ with $\tilde{x} = \gamma/(\xi\alpha^2)$. Thus, the claim (C.31) readily follows from (C.36).

The proof of (C.32) is analogous. First, we use that $\kappa_{2i}^\infty \geq 0$ for $i \geq 1$ and obtain

$$|h_{s,t}^\Sigma(\boldsymbol{x}, \boldsymbol{y}) - h_{s,t}^\Sigma(\boldsymbol{x}', \boldsymbol{y}')| \leq \sum_{j=0}^\infty \sum_{k=0}^\infty \left(\frac{\gamma}{\alpha^2}\right)^{j+k} \left(\kappa_{2(j+k+1)}^\infty \frac{\gamma^2}{\alpha^2}|y_{s-j,t-k} - y'_{s-j,t-k}| \right.$$
$$\left. + \kappa_{2(j+k+2)}^\infty \frac{\gamma^2}{\alpha^4}|x_{s-j,t-k} - x'_{s-j,t-k}|\right). \qquad \text{(C.37)}$$

Thus, by using (C.34), we have

$$\|h^\Sigma(\boldsymbol{x}, \boldsymbol{y}) - h^\Sigma(\boldsymbol{x}', \boldsymbol{y}')\|_\xi \leq \sum_{j=0}^\infty \sum_{k=0}^\infty \left(\frac{\gamma}{\xi\alpha^2}\right)^{j+k} \left(\kappa_{2(j+k+1)}^\infty \frac{\gamma^2}{\alpha^2}\|\boldsymbol{y} - \boldsymbol{y}'\|_\xi + \kappa_{2(j+k+2)}^\infty \frac{\gamma^2}{\alpha^4}\|\boldsymbol{x} - \boldsymbol{x}'\|_\xi\right). \qquad \text{(C.38)}$$

Finally, by using (A.16) and (A.17) to compute the sums in (C.38), we deduce that

$$\|h^\Sigma(\boldsymbol{x}, \boldsymbol{y}) - h^\Sigma(\boldsymbol{x}', \boldsymbol{y}')\|_\xi \leq \frac{\gamma^2}{\alpha^2} R'\left(\frac{\gamma}{\xi\alpha^2}\right)\|\boldsymbol{y} - \boldsymbol{y}'\|_\xi$$
$$+ \xi^2 \left(\frac{\gamma}{\xi\alpha^2}R'\left(\frac{\gamma}{\xi\alpha^2}\right) - R\left(\frac{\gamma}{\xi\alpha^2}\right)\right)\|\boldsymbol{x} - \boldsymbol{x}'\|_\xi, \qquad \text{(C.39)}$$

which readily leads to (C.32). $\qquad\square$

Let us consider the map $G^{\Omega,\Sigma}$ obtained by the successive composition of $(\boldsymbol{x},\boldsymbol{y}) \mapsto (\boldsymbol{x}, h^\Omega(\boldsymbol{x},\boldsymbol{y}))$ and $(\boldsymbol{x},\boldsymbol{y}) \mapsto (h^\Sigma(\boldsymbol{x},\boldsymbol{y}),\boldsymbol{y})$, i.e.,

$$G^{\Omega,\Sigma}(\boldsymbol{x},\boldsymbol{y}) = (G_x^{\Omega,\Sigma}(\boldsymbol{x},\boldsymbol{y}), G_y^{\Omega,\Sigma}(\boldsymbol{x},\boldsymbol{y})) = \left(h^\Sigma(\boldsymbol{x}, h^\Omega(\boldsymbol{x},\boldsymbol{y})), h^\Omega(\boldsymbol{x},\boldsymbol{y})\right). \tag{C.40}$$

Given $\beta > 0$, define the norm $\|\cdot\|_{\xi,\beta}$ as

$$\|(\boldsymbol{x},\boldsymbol{y})\|_{\xi,\beta} = \|\boldsymbol{x}\|_\xi + \beta\|\boldsymbol{y}\|_\xi. \tag{C.41}$$

We now use the Lipschitz bounds of Lemma C.5 to prove that $G^{\Omega,\Sigma}$ is a contraction for a certain value of $\beta$.

**Lemma C.6** (Composition of limit maps is a contraction). *Consider the map $G^{\Omega,\Sigma}$ defined in* (C.40), *and let $I_\Omega^*$, $I_\Sigma^*$ be the sets given by Lemma C.3. Assume that $\kappa_{2i}^\infty \geq 0$ for all $i \geq 1$, and let $\xi < 1$ be such that $\tilde{\alpha}\sqrt{\xi} > \tilde{\alpha}_{\rm s}$. Then, if $(\boldsymbol{x},\boldsymbol{y}) \in \mathcal{X}_{I_\Sigma^*} \times \mathcal{X}_{I_\Omega^*}$, we have that $G^{\Omega,\Sigma}(\boldsymbol{x},\boldsymbol{y}) \in \mathcal{X}_{I_\Omega^*} \times \mathcal{X}_{I_\Sigma^*}$. Furthermore, there exists $\beta^* > 0$ and $\tau < 1$ such that, for any $(\boldsymbol{x},\boldsymbol{y}) \in \mathcal{X}_{I_\Sigma^*} \times \mathcal{X}_{I_\Omega^*}$,*

$$\|G^{\Omega,\Sigma}(\boldsymbol{x},\boldsymbol{y}) - G^{\Omega,\Sigma}(\boldsymbol{x}',\boldsymbol{y}')\|_{\xi,\beta^*} \leq \tau\|(\boldsymbol{x},\boldsymbol{y}) - (\boldsymbol{x}',\boldsymbol{y}')\|_{\xi,\beta^*}. \tag{C.42}$$

*Proof.* The claim that $G^{\Omega,\Sigma} : \mathcal{X}_{I_\Omega^*} \times \mathcal{X}_{I_\Sigma^*} \to \mathcal{X}_{I_\Omega^*} \times \mathcal{X}_{I_\Sigma^*}$ follows directly from Lemma C.3. We now show that (C.42) holds. By using the definition (C.40) and the Lipschitz bounds (C.31)-(C.32) of Lemma C.5, we obtain that

$$\|G^{\Omega,\Sigma}(\boldsymbol{x},\boldsymbol{y}) - G^{\Omega,\Sigma}(\boldsymbol{x}',\boldsymbol{y}')\|_{\xi,\beta} \leq \|\boldsymbol{x} - \boldsymbol{x}'\|_\xi \left(\tilde{x}R'(\tilde{x}) - R(\tilde{x}) + \gamma(\tilde{x}R'(\tilde{x}))^2 + \beta\tilde{x}R'(\tilde{x})\right)$$
$$+ \|\boldsymbol{y} - \boldsymbol{y}'\|_\xi \left(\gamma^2(\tilde{x}R'(\tilde{x}))^2 - \gamma^2\tilde{x}R'(\tilde{x})R(\tilde{x}) + \beta\gamma(\tilde{x}R'(\tilde{x}) - R(\tilde{x}))\right), \tag{C.43}$$

where we have set $\tilde{x} = \gamma/(\xi\alpha^2)$. Hence, the claim of the lemma holds if there exists $\beta^* > 0$ and $\tau < 1$ such that

$$\beta^*\tilde{x}R'(\tilde{x}) + \gamma(\tilde{x}R'(\tilde{x}))^2 - R(\tilde{x}) + \tilde{x}R'(\tilde{x}) \leq \tau,$$
$$\beta^*\gamma(\tilde{x}R'(\tilde{x}) - R(\tilde{x})) + \gamma^2(\tilde{x}R'(\tilde{x}))^2 - \gamma^2\tilde{x}R(\tilde{x})R'(\tilde{x}) \leq \tau\beta^*. \tag{C.44}$$

We note that, as $\tilde{\alpha}\sqrt{\xi} > \tilde{\alpha}_{\rm s}$, (C.26) holds with $\tilde{x}$ in place of $x$. Hence, one readily verifies that $1 - \gamma\tilde{x}R'(\tilde{x}) + R(\tilde{x}) > 0$. Furthermore, we have that $\tilde{x}R'(\tilde{x}) > 0$, as $\tilde{x} > 0$ and the rectangular free cumulants are non-negative. Thus, the two inequalities in (C.44) can be satisfied simultaneously if there exists $\beta^* > 0$ such that

$$\frac{\gamma^2(\tilde{x}R'(\tilde{x}))^2 - \gamma^2\tilde{x}R(\tilde{x})R'(\tilde{x})}{1 - \gamma\tilde{x}R'(\tilde{x}) + \gamma R(\tilde{x})} < \beta^* < \frac{1 - \gamma(\tilde{x}R'(\tilde{x}))^2 - \tilde{x}R'(\tilde{x}) + R(x)}{\tilde{x}R'(\tilde{x})}.$$

These last two inequalities can be satisfied simultaneously if

$$\frac{\gamma^2(\tilde{x}R'(\tilde{x}))^2 - \gamma^2\tilde{x}R(\tilde{x})R'(\tilde{x})}{1 - \gamma\tilde{x}R'(\tilde{x}) + \gamma R(\tilde{x})} < \frac{1 - \gamma(\tilde{x}R'(\tilde{x}))^2 - \tilde{x}R'(\tilde{x}) + R(x)}{\tilde{x}R'(\tilde{x})}. \tag{C.45}$$

By using again that $1 - \gamma\tilde{x}R'(\tilde{x}) + R(\tilde{x}) > 0$ and $\tilde{x}R'(\tilde{x}) > 0$, (C.45) can be rewritten as

$$\left(1 - \gamma(\tilde{x}R'(\tilde{x}))^2 - \tilde{x}R'(\tilde{x}) + R(x)\right)\left(1 - \gamma\tilde{x}R'(\tilde{x}) + \gamma R(\tilde{x})\right)$$
$$> \tilde{x}R'(\tilde{x})\left(\gamma^2(\tilde{x}R'(\tilde{x}))^2 - \gamma^2\tilde{x}R(\tilde{x})R'(\tilde{x})\right),$$

which again follows from (C.26) with $\tilde{x}$ in place of $x$. Thus, there exists $\beta^* > 0$ and $\tau < 1$ such that (C.44) is satisfied, completing the proof. $\qquad\square$

At this point, we show that the state evolution of $\tilde{\boldsymbol{\Sigma}}_{\bar{T}}, \tilde{\boldsymbol{\Omega}}_{\bar{T}}$ can be approximated via the fixed maps $h^\Sigma, h^\Omega$.

**Lemma C.7** (Limit maps approximate SE maps – Rectangular matrices). *Consider the map $(h^\Omega(\boldsymbol{x},\boldsymbol{y}), h^\Sigma(\boldsymbol{x},\boldsymbol{y})) : \mathcal{X}_{I_\Omega^*} \times \mathcal{X}_{I_\Sigma^*} \to \mathcal{X}_{I_\Omega^*} \times \mathcal{X}_{I_\Sigma^*}$ defined in* (C.21), *where $I_\Omega^*$, $I_\Sigma^*$ are given by Lemma C.3. Assume that $\kappa_{2i}^\infty \geq 0$ for all $i \geq 1$, and let $\xi < 1$ be such that $\tilde{\alpha}\sqrt{\xi} > \tilde{\alpha}_{\rm s}$. Then, for any $(\boldsymbol{x},\boldsymbol{y}) \in \mathcal{X}_{I_\Sigma^*} \times \mathcal{X}_{I_\Omega^*}$,*

$$\|\tilde{\boldsymbol{\Omega}}_{\bar{T}} - h^\Omega(\boldsymbol{x},\boldsymbol{y})\|_\xi \leq \tilde{x}R'(\tilde{x})\|\tilde{\boldsymbol{\Sigma}}_{\bar{T}-1} - \boldsymbol{x}\|_\xi + \gamma(\tilde{x}R'(\tilde{x}) - R(\tilde{x}))\|\tilde{\boldsymbol{\Omega}}_{\bar{T}-1} - \boldsymbol{y}\|_\xi + F_1(\bar{T}), \tag{C.46}$$

$$\|\tilde{\boldsymbol{\Sigma}}_{\bar{T}} - h^{\Sigma}(\boldsymbol{x}, \boldsymbol{y})\|_{\xi} \leq \gamma \tilde{x} R'(\tilde{x}) \|\tilde{\boldsymbol{\Omega}}_{\bar{T}-1} - \boldsymbol{y}\|_{\xi} + (\tilde{x} R'(\tilde{x}) - R(\tilde{x})) \|\tilde{\boldsymbol{\Sigma}}_{\bar{T}-1} - \boldsymbol{x}\|_{\xi} + F_2(\bar{T}),$$
(C.47)

*where $\tilde{x} = \gamma/(\xi\alpha^2)$ and*

$$\lim_{\bar{T}\to\infty} F_1(\bar{T}) = 0, \quad \lim_{\bar{T}\to\infty} F_2(\bar{T}) = 0.$$
(C.48)

*Proof.* First, we write

$$\|\tilde{\boldsymbol{\Omega}}_{\bar{T}} - h^{\Omega}(\boldsymbol{x}, \boldsymbol{y})\|_{\xi} = \sup_{s,t\leq 0} \xi^{\max(|s|,|t|)} |(\tilde{\boldsymbol{\Omega}}_{\bar{T}})_{s,t} - h^{\Omega}_{s,t}(\boldsymbol{x}, \boldsymbol{y})|$$

$$= \max\left(\sup_{\substack{s,t\leq 0 \\ \max(|s|,|t|)<\bar{T}}} \xi^{\max(|s|,|t|)} |(\tilde{\boldsymbol{\Omega}}_{\bar{T}})_{s,t} - h^{\Omega}_{s,t}(\boldsymbol{x}, \boldsymbol{y})|,\right.$$

$$\left.\sup_{\substack{s,t\leq 0 \\ \max(|s|,|t|)\geq\bar{T}}} \xi^{\max(|s|,|t|)} |(\tilde{\boldsymbol{\Omega}}_{\bar{T}})_{s,t} - h^{\Omega}_{s,t}(\boldsymbol{x}, \boldsymbol{y})|\right),$$

where $(\tilde{\boldsymbol{\Omega}}_{\bar{T}})_{s,t} = \tilde{\omega}_{s+\bar{T},t+\bar{T}}$ if $s \geq -\bar{T}$ and $t \geq -\bar{T}$, and $(\tilde{\boldsymbol{\Omega}}_{\bar{T}})_{s,t} = 0$ otherwise.

Let us look at the case $\max(|s|,|t|) < \bar{T}$, and define $I_1 = \{(j,k) : j \geq s + \bar{T} \text{ or } k \geq t + \bar{T}\}$. Then,

$$|(\tilde{\boldsymbol{\Omega}}_{\bar{T}})_{s,t} - h^{\Omega}_{s,t}(\boldsymbol{x}, \boldsymbol{y})|$$

$$= \left| \gamma \sum_{j=0}^{s+\bar{T}-1} \sum_{k=0}^{t+\bar{T}-1} \left(\frac{\gamma}{\alpha^2}\right)^{j+k} \left(\kappa_{2(j+k+1)}^{\infty} \frac{1}{\alpha^2} \left(\alpha^2 \Delta_{\mathrm{PCA}} + \tilde{\sigma}_{s-j+\bar{T}-1,t-k+\bar{T}-1}\right)\right.\right.$$

$$\left.+ \kappa_{2(j+k+2)}^{\infty} \frac{\gamma^2}{\alpha^4} \left(\alpha^2 \Delta_{\mathrm{PCA}} + \tilde{\omega}_{s-j+\bar{T}-1,t-k+\bar{T}-1}\right)\right)$$

$$- \gamma \sum_{j=0}^{\infty} \sum_{k=0}^{\infty} \left(\frac{\gamma}{\alpha^2}\right)^{j+k} \left(\kappa_{2(j+k+1)}^{\infty} \frac{1}{\alpha^2} \left(\alpha^2 \Delta_{\mathrm{PCA}} + x_{s-j,t-k}\right)\right.$$

$$\left.\left.+ \kappa_{2(j+k+2)}^{\infty} \frac{\gamma^2}{\alpha^4} \left(\alpha^2 \Delta_{\mathrm{PCA}} + y_{s-j,t-k}\right)\right)\right|$$
(C.49)

$$\leq \left| \gamma \sum_{j=0}^{s+\bar{T}-1} \sum_{k=0}^{t+\bar{T}-1} \left(\frac{\gamma}{\alpha^2}\right)^{j+k} \left(\kappa_{2(j+k+1)}^{\infty} \frac{1}{\alpha^2} \left(x_{s-j,t-k} - \tilde{\sigma}_{s-j+\bar{T}-1,t-k+\bar{T}-1}\right)\right.\right.$$

$$\left.\left.+ \kappa_{2(j+k+2)}^{\infty} \frac{\gamma^2}{\alpha^4} \left(y_{s-j,t-k} - \tilde{\omega}_{s-j+\bar{T}-1,t-k+\bar{T}-1}\right)\right)\right|$$

$$+ \left| \gamma \sum_{j,k\in I_1} \left(\frac{\gamma}{\alpha^2}\right)^{j+k} \left(\kappa_{2(j+k+1)}^{\infty} \frac{1}{\alpha^2} \left(\alpha^2 \Delta_{\mathrm{PCA}} + x_{s-j,t-k}\right)\right.\right.$$

$$\left.\left.+ \kappa_{2(j+k+2)}^{\infty} \frac{\gamma^2}{\alpha^4} \left(\alpha^2 \Delta_{\mathrm{PCA}} + y_{s-j,t-k}\right)\right)\right| := T_1 + T_2.$$

The term $T_1$ can be upper bounded as follows:

$$T_1 \overset{(a)}{\leq} \gamma \sum_{j=0}^{s+\bar{T}-1} \sum_{k=0}^{t+\bar{T}-1} \left(\frac{\gamma}{\alpha^2}\right)^{j+k} \left(\kappa_{2(j+k+1)}^{\infty} \frac{1}{\alpha^2} \left|x_{s-j,t-k} - \tilde{\sigma}_{s-j+\bar{T}-1,t-k+\bar{T}-1}\right|\right.$$

$$\left.+ \kappa_{2(j+k+2)}^{\infty} \frac{\gamma^2}{\alpha^4} \left|y_{s-j,t-k} - \tilde{\omega}_{s-j+\bar{T}-1,t-k+\bar{T}-1}\right|\right)$$

$$\leq \|\tilde{\boldsymbol{\Sigma}}_{\bar{T}-1} - \boldsymbol{x}\|_{\xi} \xi^{-\max(|s|,|t|)} \gamma \sum_{j=0}^{s+\bar{T}-1} \sum_{k=0}^{t+\bar{T}-1} \left(\frac{\gamma}{\xi\alpha^2}\right)^{j+k} \kappa_{2(j+k+1)}^{\infty} \frac{1}{\alpha^2}$$

$$+ \|\tilde{\boldsymbol{\Omega}}_{\bar{T}-1} - \boldsymbol{y}\|_{\xi} \xi^{-\max(|s|,|t|)} \gamma \sum_{j=0}^{s+\bar{T}-1} \sum_{k=0}^{t+\bar{T}-1} \left(\frac{\gamma}{\xi\alpha^2}\right)^{j+k} \kappa_{2(j+k+2)}^{\infty} \frac{\gamma}{\alpha^4}$$

$$
\overset{(b)}{\leq} \|\tilde{\boldsymbol{\Sigma}}_{\bar{T}-1} - \boldsymbol{x}\|_\xi \xi^{-\max(|s|,|t|)} \gamma \sum_{j=0}^\infty \sum_{k=0}^\infty \left(\frac{\gamma}{\xi\alpha^2}\right)^{j+k} \kappa^\infty_{2(j+k+1)} \frac{1}{\alpha^2}
$$

$$
+ \|\tilde{\boldsymbol{\Omega}}_{\bar{T}-1} - \boldsymbol{y}\|_\xi \xi^{-\max(|s|,|t|)} \gamma \sum_{j=0}^\infty \sum_{k=0}^\infty \left(\frac{\gamma}{\xi\alpha^2}\right)^{j+k} \kappa^\infty_{2(j+k+2)} \frac{\gamma}{\alpha^4}
$$

$$
\overset{(c)}{\leq} \|\tilde{\boldsymbol{\Sigma}}_{\bar{T}-1} - \boldsymbol{x}\|_\xi \xi^{-\max(|s|,|t|)} \tilde{x} R'(\tilde{x}) + \|\tilde{\boldsymbol{\Omega}}_{\bar{T}-1} - \boldsymbol{y}\|_\xi \xi^{-\max(|s|,|t|)} \gamma(\tilde{x} R'(\tilde{x}) - R(\tilde{x})),
$$
$$(\text{C.50})$$

where $\tilde{x} = \gamma/(\xi\alpha^2)$. Here, (a) and (b) follow from the hypothesis that $\kappa^\infty_{2i} \geq 0$ for $i \geq 1$, (c) uses (A.16), (A.17) and that $\xi \leq 1$. By using that $(\boldsymbol{x}, \boldsymbol{y}) \in \mathcal{X}_{I^*_\Sigma} \times \mathcal{X}_{I^*_\Omega}$, the term $T_2$ can be upper bounded as follows:

$$
T_2 \leq C_1 \sum_{j,k \in I_1} \left(\frac{\gamma}{\alpha^2}\right)^{j+k} (\kappa^\infty_{2(j+k+1)} + \kappa^\infty_{2(j+k+2)}),
$$
$$(\text{C.51})$$

where $C_1$ is a constant independent of $s, t, \bar{T}$. Note that, if $(j, k) \in I_1$, then $j + k \geq -\max(|s|, |t|) + \bar{T}$. Consequently, the RHS of (C.51) can upper bounded by

$$
C_2 \sum_{i=\bar{T}-\max(|s|,|t|)}^\infty \left(\frac{\gamma}{\alpha^2}\right)^i (i+1) \kappa^\infty_{2(i+1)},
$$
$$(\text{C.52})$$

where $C_2$ is a constant independent of $s, t, \bar{T}$. By combining (C.49), (C.50), (C.51) and (C.52), we obtain that

$$
\sup_{\substack{s,t \leq 0 \\ \max(|s|,|t|) < \bar{T}}} \xi^{\max(|s|,|t|)} |(\tilde{\boldsymbol{\Omega}}_{\bar{T}})_{s,t} - h^\Omega_{s,t}(\boldsymbol{x}, \boldsymbol{y})| \leq \|\tilde{\boldsymbol{\Sigma}}_{\bar{T}-1} - \boldsymbol{x}\|_\xi \tilde{x} R'(\tilde{x})
$$

$$
+ \|\tilde{\boldsymbol{\Omega}}_{\bar{T}-1} - \boldsymbol{y}\|_\xi \gamma(\tilde{x} R'(\tilde{x}) - R(\tilde{x})) + C_2 \sup_{0 \leq t \leq \bar{T}} \xi^t \sum_{i=\bar{T}-t}^\infty \left(\frac{\gamma}{\alpha^2}\right)^i (i+1) \kappa^\infty_{2(i+1)}.
$$
$$(\text{C.53})$$

Let us now look at the case $\max(|s|, |t|) \geq \bar{T}$. Recall that $|h^\Omega_{s,t}(\boldsymbol{x}, \boldsymbol{y})| \leq a_\Omega$, $\tilde{\sigma}_{0,0} = (1 - \Delta_{\mathrm{PCA}})\alpha^2$ and $\tilde{\sigma}_{0,t} = 0$ for $t \in [1, \bar{T}]$. Thus,

$$
|(\tilde{\boldsymbol{\Omega}}_{\bar{T}})_{s,t} - h^\Omega_{s,t}(\boldsymbol{x}, \boldsymbol{y})| \leq C_3,
$$

where $C_3$ is a constant independent of $s, t, \bar{T}$. This immediately implies that

$$
\sup_{\substack{s,t \leq 0 \\ \max(|s|,|t|) \geq \bar{T}}} \xi^{\max(|s|,|t|)} |(\tilde{\boldsymbol{\Omega}}_{\bar{T}})_{s,t} - h^\Omega_{s,t}(\boldsymbol{x}, \boldsymbol{y})| \leq C_3 \xi^{\bar{T}},
$$

which combined with (C.53) allows us to conclude that

$$
\|\tilde{\boldsymbol{\Omega}}_{\bar{T}} - h^\Omega(\boldsymbol{x}, \boldsymbol{y})\|_\xi \leq \|\tilde{\boldsymbol{\Sigma}}_{\bar{T}-1} - \boldsymbol{x}\|_\xi \tilde{x} R'(\tilde{x})
$$

$$
+ \|\tilde{\boldsymbol{\Omega}}_{\bar{T}-1} - \boldsymbol{y}\|_\xi \gamma(\tilde{x} R'(\tilde{x}) - R(\tilde{x})) + C_2 \sup_{0 \leq t \leq \bar{T}} \xi^t \sum_{i=\bar{T}-t}^\infty \left(\frac{\gamma}{\alpha^2}\right)^i (i+1) \kappa^\infty_{2(i+1)} + C_3 \xi^{\bar{T}}.
$$
$$(\text{C.54})$$

As $\tilde{\alpha} > \tilde{\alpha}_{\mathrm{s}}$ and the series in (A.16) is convergent for $z < 1/(\tilde{\alpha}_{\mathrm{s}})^2$, one readily verifies that

$$
\lim_{\bar{T} \to \infty} \sup_{0 \leq t \leq \bar{T}} \xi^t \sum_{i=\bar{T}-t}^\infty \left(\frac{\gamma}{\alpha^2}\right)^i (i+1) \kappa^\infty_{2(i+1)} = 0,
$$
$$(\text{C.55})$$

which concludes the proof of (C.46).

The proof of (C.47) follows similar passages, and we outline them below. First, we write

$$\|\tilde{\boldsymbol{\Sigma}}_{\bar{T}} - h^{\Sigma}(\boldsymbol{x}, \boldsymbol{y})\|_{\xi} = \sup_{s,t \leq 0} \xi^{\max(|s|,|t|)} |(\tilde{\boldsymbol{\Sigma}}_{\bar{T}})_{s,t} - h^{\Sigma}_{s,t}(\boldsymbol{x}, \boldsymbol{y})|$$

$$= \max \left( \sup_{\substack{s,t \leq 0 \\ \max(|s|,|t|) < \bar{T}}} \xi^{\max(|s|,|t|)} |(\tilde{\boldsymbol{\Sigma}}_{\bar{T}})_{s,t} - h^{\Sigma}_{s,t}(\boldsymbol{x}, \boldsymbol{y})|, \right.$$

$$\left. \sup_{\substack{s,t \leq 0 \\ \max(|s|,|t|) \geq \bar{T}}} \xi^{\max(|s|,|t|)} |(\tilde{\boldsymbol{\Sigma}}_{\bar{T}})_{s,t} - h^{\Sigma}_{s,t}(\boldsymbol{x}, \boldsymbol{y})| \right),$$

where $(\tilde{\boldsymbol{\Sigma}}_{\bar{T}})_{s,t} = \tilde{\sigma}_{s+\bar{T},t+\bar{T}}$ if $s \geq -\bar{T}$ and $t \geq -\bar{T}$, and $(\tilde{\boldsymbol{\Sigma}}_{\bar{T}})_{s,t} = 0$ otherwise. For the case $\max(|s|,|t|) < \bar{T}$, we have

$$|(\tilde{\boldsymbol{\Sigma}}_{\bar{T}})_{s,t} - h^{\Sigma}_{s,t}(\boldsymbol{x}, \boldsymbol{y})|$$

$$\leq \left| \sum_{j=0}^{s+\bar{T}-1} \sum_{k=0}^{t+\bar{T}-1} \left(\frac{\gamma}{\alpha^2}\right)^{j+k} \left( \kappa^{\infty}_{2(j+k+1)} \frac{\gamma^2}{\alpha^2} \left(y_{s-j,t-k} - \tilde{\omega}_{s-j+\bar{T},t-k+\bar{T}}\right) \right.\right.$$

$$\left.\left. + \kappa^{\infty}_{2(j+k+2)} \frac{\gamma^2}{\alpha^4} \left(x_{s-j,t-k} - \tilde{\sigma}_{s-j+\bar{T}-1,t-k+\bar{T}-1}\right) \right) \right| \qquad \text{(C.56)}$$

$$+ \left| \sum_{j,k \in I_1} \left(\frac{\gamma}{\alpha^2}\right)^{j+k} \left( \kappa^{\infty}_{2(j+k+1)} \frac{\gamma^2}{\alpha^2} \left(\alpha^2 \Delta_{\text{PCA}} + y_{s-j,t-k}\right) \right.\right.$$

$$\left.\left. + \kappa^{\infty}_{2(j+k+2)} \frac{\gamma^2}{\alpha^4} \left(\alpha^2 \Delta_{\text{PCA}} + x_{s-j,t-k}\right) \right) \right| := T_3 + T_4.$$

By using (A.16), (A.17) and the non-negativity of the rectangular free cumulants, the term $T_3$ can be upper bounded as follows:

$$T_3 \leq \|\tilde{\boldsymbol{\Omega}}_{\bar{T}} - \boldsymbol{y}\|_{\xi} \xi^{-\max(|s|,|t|)} \gamma \tilde{x} R'(\tilde{x}) + \|\tilde{\boldsymbol{\Sigma}}_{\bar{T}-1} - \boldsymbol{x}\|_{\xi} \xi^{-\max(|s|,|t|)} (\tilde{x} R'(\tilde{x}) - R(\tilde{x})). \quad \text{(C.57)}$$

Furthermore, the term $T_4$ can be upper bounded as

$$T_4 \leq C_4 \sum_{i=\bar{T}-\max(|s|,|t|)}^{\infty} \left(\frac{\gamma}{\alpha^2}\right)^i (i+1) \kappa^{\infty}_{2(i+1)}, \qquad \text{(C.58)}$$

where $C_4$ is a constant independent of $s, t, \bar{T}$. For the case $\max(|s|,|t|) \geq \bar{T}$, we have

$$\sup_{\substack{s,t \leq 0 \\ \max(|s|,|t|) \geq \bar{T}}} \xi^{\max(|s|,|t|)} |(\tilde{\boldsymbol{\Sigma}}_{\bar{T}})_{s,t} - h^{\Sigma}_{s,t}(\boldsymbol{x}, \boldsymbol{y})| \leq C_5 \xi^{\bar{T}}, \qquad \text{(C.59)}$$

where $C_5$ is a constant independent of $s, t, \bar{T}$. By combining (C.56), (C.57), (C.58) and (C.59), we conclude that

$$\|\tilde{\boldsymbol{\Sigma}}_{\bar{T}} - h^{\Sigma}(\boldsymbol{x}, \boldsymbol{y})\|_{\xi} \leq \|\tilde{\boldsymbol{\Omega}}_{\bar{T}} - \boldsymbol{y}\|_{\xi} \gamma \tilde{x} R'(\tilde{x})$$

$$+ \|\tilde{\boldsymbol{\Sigma}}_{\bar{T}-1} - \boldsymbol{x}\|_{\xi} (\tilde{x} R'(\tilde{x}) - R(\tilde{x})) + C_4 \sup_{0 \leq t \leq \bar{T}} \xi^t \sum_{i=\bar{T}-t}^{\infty} \left(\frac{\gamma}{\alpha^2}\right)^i (i+1) \kappa^{\infty}_{2(i+1)} + C_5 \xi^{\bar{T}},$$

which, together with (C.55), concludes the proof of (C.47). $\qquad \square$

Finally, we can put everything together and prove Lemma C.2.

*Proof of Lemma C.2.* Fix $\epsilon > 0$ and denote by $(G^{\Omega,\Sigma})^{T_0}$ the $T_0$-fold composition of the map $G^{\Omega,\Sigma}$ defined in (C.40). Note that Lemma C.4 implies that $(\boldsymbol{x}^*, \boldsymbol{y}^*)$ is a fixed point of $G^{\Omega,\Sigma}$, and Lemma C.6 implies that this fixed point is unique. Then, for any $(\boldsymbol{x}, \boldsymbol{y}) \in \mathcal{X}_{I^*_{\Sigma}} \times \mathcal{X}_{I^*_{\Omega}}$,

$$\| (G^{\Omega,\Sigma})^{T_0} (\boldsymbol{x}, \boldsymbol{y}) - (\boldsymbol{x}^*, \boldsymbol{y}^*)\|_{\xi,\beta^*} = \| (G^{\Omega,\Sigma})^{T_0} (\boldsymbol{x}, \boldsymbol{y}) - (G^{\Omega,\Sigma})^{T_0} (\boldsymbol{x}^*, \boldsymbol{y}^*)\|_{\xi,\beta^*}$$

$$\leq \tau^{T_0} \|(\boldsymbol{x}, \boldsymbol{y}) - (\boldsymbol{x}^*, \boldsymbol{y}^*)\|_{\xi,\beta^*}, \qquad \text{(C.60)}$$

where the inequality follows from Lemma C.6. Note that $\tau < 1$ and $\mathcal{X}_{I_\Omega^*} \times \mathcal{X}_{I_\Sigma^*}$ is bounded under $\|\cdot\|_{\xi,\beta^*}$. Hence, we can make the RHS of (C.60) smaller than $\epsilon/2$ by choosing a sufficiently large $T_0$. Furthermore, an application of Lemma C.7 gives that

$$
\begin{aligned}
\|(\tilde{\boldsymbol{\Sigma}}_{\bar{T}}, \tilde{\boldsymbol{\Omega}}_{\bar{T}}) - G^{\Omega,\Sigma}(\boldsymbol{x},\boldsymbol{y})\|_{\xi,\beta^*} &\leq \|\tilde{\boldsymbol{\Sigma}}_{\bar{T}-1} - \boldsymbol{x}\|_\xi \left( \tilde{x}R'(\tilde{x}) - R(\tilde{x}) + \gamma(\tilde{x}R'(\tilde{x}))^2 + \beta^*\tilde{x}R'(\tilde{x}) \right) \\
&\quad + \|\tilde{\boldsymbol{\Omega}}_{\bar{T}-1} - \boldsymbol{y}\|_\xi \left( \gamma^2(\tilde{x}R'(\tilde{x}))^2 - \gamma^2\tilde{x}R'(\tilde{x})R(\tilde{x}) + \beta^*\gamma(\tilde{x}R'(\tilde{x}) - R(\tilde{x})) \right) + H(\bar{T}) \\
&\leq \tau\|(\tilde{\boldsymbol{\Sigma}}_{\bar{T}-1}, \tilde{\boldsymbol{\Omega}}_{\bar{T}-1}) - (\boldsymbol{x},\boldsymbol{y})\|_{\xi,\beta^*} + H(\bar{T}),
\end{aligned}
\tag{C.61}
$$

where $\lim_{\bar{T}\to\infty} H(\bar{T}) = 0$ and the inequality follows from (C.44). Therefore, for all sufficiently large $\bar{T}$,

$$
\|(\tilde{\boldsymbol{\Sigma}}_{\bar{T}+T_0}, \tilde{\boldsymbol{\Omega}}_{\bar{T}+T_0}) - \left(G^{\Omega,\Sigma}\right)^{T_0}(\boldsymbol{x},\boldsymbol{y})\|_{\xi,\beta^*} \leq \tau^{T_0}\|(\tilde{\boldsymbol{\Sigma}}_{\bar{T}}, \tilde{\boldsymbol{\Omega}}_{\bar{T}}) - (\boldsymbol{x},\boldsymbol{y})\|_{\xi,\beta^*} + \frac{\epsilon}{4}. \tag{C.62}
$$

Note that $(\boldsymbol{x},\boldsymbol{y}) \in \mathcal{X}_{I_\Sigma^*} \times \mathcal{X}_{I_\Omega^*}$ implies that $\|\boldsymbol{x}\|_\xi \leq a_\Sigma$ and $\|\boldsymbol{y}\|_\xi \leq a_\Omega$. In addition, by following the same argument as in Lemma C.3, one can show that $|\tilde{\omega}_{s,t}| \leq a_\Omega$ and $|\tilde{\sigma}_{s,t}| \leq a_\Sigma$, which in turn implies that $\|\tilde{\boldsymbol{\Omega}}_{\bar{T}}\|_\xi \leq a_\Omega$ and $\|\tilde{\boldsymbol{\Sigma}}_{\bar{T}}\|_\xi \leq a_\Sigma$. As a result, we can make the RHS of (C.62) smaller than $\epsilon/2$ by choosing a sufficiently large $T_0$. As the RHS of both (C.60) and (C.62) can be made smaller than $\epsilon/2$, an application of the triangle inequality gives that

$$
\limsup_{\bar{T}\to\infty} \|(\tilde{\boldsymbol{\Sigma}}_{\bar{T}}, \tilde{\boldsymbol{\Omega}}_{\bar{T}}) - (\boldsymbol{x}^*, \boldsymbol{y}^*)\|_{\xi,\beta^*} \leq \epsilon, \tag{C.63}
$$

which, after setting $\bar{T} = T + 1$, implies the desired result. $\qquad\square$

### C.4  Convergence to PCA Estimator for the First Phase

In this section, we prove that the artificial AMP iterate at the end of the first phase converges in normalized $\ell_2$-norm to the left singular vector produced by PCA.

**Lemma C.8** (Convergence to PCA estimator – Rectangular matrices)**.** *Consider the setting of Theorem 2, and the first phase of the artificial AMP iteration described in* (C.2)*, with initialization given by* (C.1)*. Assume that $\kappa_{2i}^\infty \geq 0$ for all $i \geq 1$, and that $\tilde{\alpha} > \tilde{\alpha}_s$. Then,*

$$
\lim_{T\to\infty} \lim_{n\to\infty} \frac{1}{\sqrt{m}} \|\tilde{\boldsymbol{u}}^{T+1} - \sqrt{m}\boldsymbol{u}_{\mathrm{PCA}}\| = 0 \quad a.s. \tag{C.64}
$$

*Proof.* Consider the following decomposition of $\tilde{\boldsymbol{u}}^{T+1}$:

$$
\tilde{\boldsymbol{u}}^{T+1} = \zeta_{T+1}\boldsymbol{u}_{\mathrm{PCA}} + \boldsymbol{r}^{T+1}, \tag{C.65}
$$

where $\zeta_{T+1} = \langle \tilde{\boldsymbol{u}}^{T+1}, \boldsymbol{u}_{\mathrm{PCA}} \rangle$ and $\langle \boldsymbol{r}^{T+1}, \boldsymbol{u}_{\mathrm{PCA}} \rangle = 0$. Define

$$
\boldsymbol{e}^{T+1} = \left( \boldsymbol{X}\boldsymbol{X}^\mathsf{T} - \left(D^{-1}\left(1/\tilde{\alpha}^2\right)\right)^2 \boldsymbol{I}_m \right) \tilde{\boldsymbol{u}}^{T+1}, \tag{C.66}
$$

where $D^{-1}$ is the inverse of the $D$-transform of $\Lambda$. Then, by using (C.65), (C.66) can be rewritten as

$$
\left( \boldsymbol{X}\boldsymbol{X}^\mathsf{T} - \left(D^{-1}\left(1/\tilde{\alpha}^2\right)\right)^2 \boldsymbol{I}_m \right) \boldsymbol{r}^{T+1} = \boldsymbol{e}^{T+1} - \left( \boldsymbol{X}\boldsymbol{X}^\mathsf{T} - \left(D^{-1}\left(1/\tilde{\alpha}^2\right)\right)^2 \boldsymbol{I}_m \right) \zeta_{T+1}\boldsymbol{u}_{\mathrm{PCA}}. \tag{C.67}
$$

Note that $\boldsymbol{X}$ (and consequently $\boldsymbol{X}\boldsymbol{X}^\mathsf{T}$) has a spectral gap, in the sense that, almost surely, $\sigma_1(\boldsymbol{X}) \to D^{-1}(1/\tilde{\alpha}^2)$ and $\sigma_2(\boldsymbol{X}) \to b < D^{-1}(1/\tilde{\alpha}^2)$. Furthermore, $\boldsymbol{r}^{T+1}$ is orthogonal to the left singular vector associated to the singular value $\sigma_1(\boldsymbol{X})$. Thus, by following passages analogous to (B.36), (B.37) and (B.38), we obtain that

$$
\left\| \left( \boldsymbol{X}\boldsymbol{X}^\mathsf{T} - \left(D^{-1}\left(1/\tilde{\alpha}^2\right)\right)^2 \boldsymbol{I}_m \right) \boldsymbol{r}^{T+1} \right\| \geq c\|\boldsymbol{r}^{T+1}\|, \tag{C.68}
$$

where $c > 0$ is a constant (independent of $n, m, T$).

Next, we prove that almost surely

$$
\lim_{T\to\infty} \lim_{n\to\infty} \frac{1}{\sqrt{m}} \left\| \boldsymbol{e}^{T+1} - \left( \boldsymbol{X}\boldsymbol{X}^\mathsf{T} - \left(D^{-1}\left(1/\tilde{\alpha}^2\right)\right)^2 \boldsymbol{I}_m \right) \zeta_{T+1}\boldsymbol{u}_{\mathrm{PCA}} \right\| = 0. \tag{C.69}
$$

An application of the triangle inequality gives that

$$
\left\| \boldsymbol{e}^{T+1} - \left( \boldsymbol{X}\boldsymbol{X}^\mathsf{T} - \left( D^{-1}\left(1/\tilde{\alpha}^2\right)\right)^2 \boldsymbol{I}_m \right) \zeta_{T+1}\boldsymbol{u}_{\mathrm{PCA}} \right\|
$$
$$
\leq \left\| \boldsymbol{e}^{T+1} \right\| + \left\| \left( \boldsymbol{X}\boldsymbol{X}^\mathsf{T} - \left( D^{-1}\left(1/\tilde{\alpha}^2\right)\right)^2 \boldsymbol{I}_m \right) \zeta_{T+1}\boldsymbol{u}_{\mathrm{PCA}} \right\|. \tag{C.70}
$$

The second term on the RHS of (C.70) is equal to

$$
|\zeta_{T+1}| \left| \lambda_1(\boldsymbol{X}\boldsymbol{X}^\mathsf{T}) - \left( D^{-1}\left(1/\tilde{\alpha}^2\right)\right)^2 \right|. \tag{C.71}
$$

By using Theorem 2.8 of [11], we have that, for $\tilde{\alpha} > \tilde{\alpha}_{\mathrm{s}}$, almost surely,

$$
\lim_{m\to\infty} \left| \lambda_1(\boldsymbol{X}\boldsymbol{X}^\mathsf{T}) - \left( D^{-1}\left(1/\tilde{\alpha}^2\right)\right)^2 \right| = 0. \tag{C.72}
$$

Furthermore,

$$
\frac{1}{\sqrt{m}}|\zeta_{T+1}| \leq \frac{1}{\sqrt{m}}\|\tilde{\boldsymbol{u}}^{T+1}\| = \frac{1}{\alpha\sqrt{m}}\|\tilde{\boldsymbol{f}}^T\|.
$$

By Proposition C.1, we have that

$$
\lim_{m\to\infty} \frac{1}{\alpha\sqrt{m}}\|\tilde{\boldsymbol{f}}^T\| = \frac{1}{\alpha}\sqrt{\tilde{\mu}_T^2 + \tilde{\sigma}_{T,T}},
$$

which, for sufficiently large $T$, is upper bounded by a constant independent of $n, m, T$, as $\tilde{\mu}_T = \alpha\sqrt{\Delta_{\mathrm{PCA}}}$ and $\tilde{\sigma}_{T,T}$ converges to $\alpha^2(1 - \Delta_{\mathrm{PCA}})$ as $T \to \infty$ by Lemma C.2. By combining this result with (C.72), we deduce that

$$
\lim_{T\to\infty}\lim_{m\to\infty} \frac{1}{\sqrt{m}} \left\| \left( \boldsymbol{X}\boldsymbol{X}^\mathsf{T} - \left( D^{-1}\left(1/\tilde{\alpha}^2\right)\right)^2 \boldsymbol{I}_m \right) \zeta_{T+1}\boldsymbol{u}_{\mathrm{PCA}} \right\| = 0. \tag{C.73}
$$

In order to bound the first term on the RHS of (C.70), we proceed as follows:

$$
\lim_{m\to\infty}\frac{1}{m}\|\boldsymbol{e}^{T+1}\|^2 = \lim_{m\to\infty}\frac{1}{m}\left\| \left( \boldsymbol{X}\boldsymbol{X}^\mathsf{T} - \left( D^{-1}\left(1/\tilde{\alpha}^2\right)\right)^2 \boldsymbol{I}_m \right) \tilde{\boldsymbol{u}}^{T+1} \right\|^2
$$

$$
\overset{(a)}{=} \lim_{m\to\infty}\frac{1}{m}\left\| \tilde{\alpha}^2 \left( \frac{1}{\alpha}\tilde{\boldsymbol{f}}^{T+1} + \frac{1}{\tilde{\alpha}^2}\sum_{i=1}^{T+1}\kappa_{2(T-i+2)}\left(\frac{1}{\tilde{\alpha}^2}\right)^{T-i+1}\tilde{\boldsymbol{u}}^i + \frac{\gamma}{\tilde{\alpha}^2}\sum_{i=1}^{T}\kappa_{2(T-i+1)}\left(\frac{1}{\tilde{\alpha}^2}\right)^{T-i} \right. \right.
$$
$$
\left. \left. \cdot \left( \tilde{\boldsymbol{u}}^{i+1} + \frac{1}{\tilde{\alpha}^2}\sum_{j=1}^{i}\kappa_{2(i-j+1)}\left(\frac{1}{\tilde{\alpha}^2}\right)^{i-j}\tilde{\boldsymbol{u}}^j \right) \right) - \left(D^{-1}\left(1/\tilde{\alpha}^2\right)\right)^2\tilde{\boldsymbol{u}}^{T+1} \right\|^2
$$

$$
\overset{(b)}{=} \lim_{m\to\infty}\frac{1}{m}\left\| \tilde{\alpha}^2 \left( \frac{1}{\alpha}\tilde{\boldsymbol{f}}^{T+1} + \frac{1}{\tilde{\alpha}^2}\sum_{i=1}^{T+1}\kappa_{2(T-i+2)}^\infty\left(\frac{1}{\tilde{\alpha}^2}\right)^{T-i+1}\tilde{\boldsymbol{u}}^i + \frac{\gamma}{\tilde{\alpha}^2}\sum_{i=1}^{T}\kappa_{2(T-i+1)}^\infty\left(\frac{1}{\tilde{\alpha}^2}\right)^{T-i} \right. \right.
$$
$$
\left. \left. \cdot \left( \tilde{\boldsymbol{u}}^{i+1} + \frac{1}{\tilde{\alpha}^2}\sum_{j=1}^{i}\kappa_{2(i-j+1)}^\infty\left(\frac{1}{\tilde{\alpha}^2}\right)^{i-j}\tilde{\boldsymbol{u}}^j \right) \right) - \left(D^{-1}\left(1/\tilde{\alpha}^2\right)\right)^2\tilde{\boldsymbol{u}}^{T+1} \right\|^2
$$

$$
\overset{(c)}{=} \mathbb{E}\left\{ \left| \tilde{\alpha}^2 \left( \frac{1}{\alpha}\tilde{F}_{T+1} + \frac{1}{\tilde{\alpha}^2}\sum_{i=1}^{T+1}\kappa_{2(T-i+2)}^\infty\left(\frac{1}{\tilde{\alpha}^2}\right)^{T-i+1}\tilde{U}_i + \frac{\gamma}{\tilde{\alpha}^2}\sum_{i=1}^{T}\kappa_{2(T-i+1)}^\infty\left(\frac{1}{\tilde{\alpha}^2}\right)^{T-i} \right. \right. \right.
$$
$$
\left. \left. \left. \cdot \left( \tilde{U}_{i+1} + \frac{1}{\tilde{\alpha}^2}\sum_{j=1}^{i}\kappa_{2(i-j+1)}^\infty\left(\frac{1}{\tilde{\alpha}^2}\right)^{i-j}\tilde{U}_j \right) \right) - \left(D^{-1}\left(1/\tilde{\alpha}^2\right)\right)^2\tilde{U}_{T+1} \right|^2 \right\}. 
$$
$$
\tag{C.74}
$$

Here, (a) uses the iteration (C.2) of the first phase of the artificial AMP; (b) uses that, for all $i$, $\kappa_{2i} \to \kappa_{2i}^\infty$ as $n \to \infty$, as well as an argument similar to (B.45)-(B.46); and (c) follows from Proposition C.1, where $\tilde{U}_t$ for $t \in [1, T+1]$ and $\tilde{F}_{T+1}$ are defined in (C.10) and (C.11). After some

manipulations we can upper bound the RHS of (C.74) by triangle inequality as

$$
5 \cdot \mathbb{E}\left\{ \left( \tilde{\alpha}^2 + \sum_{i=0}^{T} \kappa_{2(i+1)}^{\infty} \left( \frac{1}{\tilde{\alpha}^2} \right)^i + \gamma \sum_{i=0}^{T-1} \kappa_{2(i+1)}^{\infty} \left( \frac{1}{\tilde{\alpha}^2} \right)^i + \gamma \sum_{i=1}^{T} \sum_{j=0}^{T-i} \kappa_{2i}^{\infty} \kappa_{2(j+1)}^{\infty} \left( \frac{1}{\tilde{\alpha}^2} \right)^{i+j} \right. \right.
$$
$$
\left. \left. - \left( D^{-1} \left( 1/\tilde{\alpha}^2 \right) \right)^2 \right)^2 \tilde{U}_{T+1}^2 \right\}
$$

$$
+ 5 \cdot \mathbb{E}\left\{ \left( \gamma \sum_{i=1}^{T} \sum_{j=0}^{T-i} \kappa_{2i}^{\infty} \kappa_{2(j+1)}^{\infty} \left( \frac{1}{\tilde{\alpha}^2} \right)^{i+j} (\tilde{U}_{T-i-j+1} - \tilde{U}_{T+1}) \right)^2 \right\}
$$

$$
+ 5 \cdot \mathbb{E}\left\{ \left( \gamma \sum_{i=0}^{T-1} \kappa_{2(i+1)}^{\infty} \left( \frac{1}{\tilde{\alpha}^2} \right)^i (\tilde{U}_{T-i+1} - \tilde{U}_{T+1}) \right)^2 \right\}
$$

$$
+ 5 \cdot \mathbb{E}\left\{ \left( \sum_{i=0}^{T} \kappa_{2(i+1)}^{\infty} \left( \frac{1}{\tilde{\alpha}^2} \right)^i (\tilde{U}_{T-i+1} - \tilde{U}_{T+1}) \right)^2 \right\}
$$

$$
+ 5 \cdot \mathbb{E}\left\{ \tilde{\alpha}^4 \left( \frac{1}{\alpha} \tilde{F}_{T+1} - \tilde{U}_{T+1} \right)^2 \right\} := S_1 + S_2 + S_3 + S_4 + S_5.
$$

(C.75)

The term $S_5$ can be expressed as

$$
S_5 = 5 \frac{\alpha^2}{\gamma^2} (\tilde{\sigma}_{T+1,T+1} - 2\tilde{\sigma}_{T+1,T} + \tilde{\sigma}_{T,T}).
$$

Thus, by Lemma C.2, we have that

$$
\lim_{T \to \infty} S_5 = 0. \tag{C.76}
$$

The term $S_1$ can be expressed as

$$
S_1 = 5 \frac{\tilde{\mu}_T^2 + \tilde{\sigma}_{T,T}}{\alpha^2} \cdot \left( \tilde{\alpha}^2 + \sum_{i=0}^{T} \kappa_{2(i+1)}^{\infty} \left( \frac{1}{\tilde{\alpha}^2} \right)^i + \gamma \sum_{i=0}^{T-1} \kappa_{2(i+1)}^{\infty} \left( \frac{1}{\tilde{\alpha}^2} \right)^i \right.
$$
$$
\left. + \gamma \sum_{i=1}^{T} \sum_{j=0}^{T-i} \kappa_{2i}^{\infty} \kappa_{2(j+1)}^{\infty} \left( \frac{1}{\tilde{\alpha}^2} \right)^{i+j} - \left( D^{-1} \left( 1/\tilde{\alpha}^2 \right) \right)^2 \right)^2.
$$

Thus, by Lemma C.2, we have that

$$
\lim_{T \to \infty} S_1 = 5 \cdot \left( \tilde{\alpha}^2 + \sum_{i=0}^{\infty} \kappa_{2(i+1)}^{\infty} \left( \frac{1}{\tilde{\alpha}^2} \right)^i + \gamma \sum_{i=0}^{\infty} \kappa_{2(i+1)}^{\infty} \left( \frac{1}{\tilde{\alpha}^2} \right)^i \right.
$$
$$
\left. + \gamma \sum_{i=1}^{\infty} \sum_{j=0}^{\infty} \kappa_{2i}^{\infty} \kappa_{2(j+1)}^{\infty} \left( \frac{1}{\tilde{\alpha}^2} \right)^{i+j} - \left( D^{-1} \left( 1/\tilde{\alpha}^2 \right) \right)^2 \right)^2 = 0, \tag{C.77}
$$

where the last equality follows from (A.15) and (A.19). The term $S_4$ can be expressed as

$$
S_4 = \frac{5}{\alpha^2} \sum_{i,j=0}^{T} \kappa_{2(i+1)}^{\infty} \kappa_{2(j+1)}^{\infty} \left( \frac{1}{\tilde{\alpha}^2} \right)^{i+j} (\tilde{\sigma}_{T-j,T-i} + \tilde{\sigma}_{T,T} - \tilde{\sigma}_{T,T-i} - \tilde{\sigma}_{T,T-j}),
$$

which can upper bounded by

$$
\frac{5}{\alpha^2} \sum_{i,j=0}^{T} \kappa_{2(i+1)}^{\infty} \kappa_{2(j+1)}^{\infty} \left( \frac{1}{\tilde{\alpha}^2} \right)^{i+j}
$$
$$
\left( |\tilde{\sigma}_{T-j,T-i} - \alpha^2 (1 - \Delta_{\text{PCA}})| + |\tilde{\sigma}_{T,T} - \alpha^2 (1 - \Delta_{\text{PCA}})| \right.
$$
$$
\left. + |\tilde{\sigma}_{T,T-i} - \alpha^2 (1 - \Delta_{\text{PCA}})| + |\tilde{\sigma}_{T,T-j} - \alpha^2 (1 - \Delta_{\text{PCA}})| \right). \tag{C.78}
$$

By Lemma C.2, for any $\epsilon > 0$, there exists $T^*(\epsilon)$ such that for all $T > T^*(\epsilon)$, the quantity in (C.78) is upper bounded by

$$\epsilon \cdot \frac{5}{\alpha^2} \sum_{i,j=0}^{T} \kappa_{2(i+1)}^{\infty} \kappa_{2(j+1)}^{\infty} \left( \frac{1}{\tilde{\alpha}^2} \right)^{i+j} \cdot \left( \xi^{-\max(i,j)} + 1 + \xi^{-i} + \xi^{-j} \right)$$

$$\overset{(a)}{\leq} \epsilon \cdot \frac{20}{\alpha^2} \sum_{i,j=0}^{T} \kappa_{2(i+1)}^{\infty} \kappa_{2(j+1)}^{\infty} \left( \frac{1}{\xi \tilde{\alpha}^2} \right)^{i+j}$$

$$\overset{(b)}{\leq} \epsilon \cdot \frac{20}{\alpha^2} \sum_{i,j=0}^{\infty} \kappa_{2(i+1)}^{\infty} \kappa_{2(j+1)}^{\infty} \left( \frac{1}{\xi \tilde{\alpha}^2} \right)^{i+j}$$

$$\overset{(c)}{\leq} \epsilon \cdot \frac{20}{\alpha^2} \left( R \left( \frac{1}{\xi \tilde{\alpha}^2} \right) \right)^2.$$

Here, (a) uses that $\xi < 1$, (b) uses that $\kappa_{2i} \geq 0$ for $i \geq 1$, and (c) uses the power series expansion (A.15) of $R$, which converges to a finite limit as $\sqrt{\xi}\tilde{\alpha} > \tilde{\alpha}_{\mathrm{s}}$. Since $\epsilon$ can be taken arbitrarily small, we deduce that

$$\lim_{T \to \infty} S_4 = 0. \tag{C.79}$$

By using the same argument, we also have that

$$\lim_{T \to \infty} S_3 = 0. \tag{C.80}$$

Finally, the term $S_2$ is upper bounded by

$$\frac{5\gamma^2}{\alpha^2} \sum_{i=1}^{T} \sum_{j=0}^{T-i} \sum_{k=1}^{T} \sum_{\ell=0}^{T-k} \kappa_{2i}^{\infty} \kappa_{2k}^{\infty} \kappa_{2(j+1)}^{\infty} \kappa_{2(\ell+1)}^{\infty} \left( \frac{1}{\tilde{\alpha}^2} \right)^{i+j+k+\ell}$$
$$\cdot \Big( |\sigma_{T,T} - \alpha^2(1 - \Delta_{\mathrm{PCA}})| + |\sigma_{T,T-i-j} - \alpha^2(1 - \Delta_{\mathrm{PCA}})|$$
$$+ |\sigma_{T,T-k-\ell} - \alpha^2(1 - \Delta_{\mathrm{PCA}})| + |\sigma_{T-i-j,T-k-\ell} - \alpha^2(1 - \Delta_{\mathrm{PCA}})| \Big). \tag{C.81}$$

By Lemma C.2, for any $\epsilon > 0$, there exists $T^*(\epsilon)$ such that for all $T > T^*(\epsilon)$, the quantity in (C.81) is upper bounded by

$$\epsilon \cdot \frac{20\gamma^2}{\alpha^2} \sum_{i=1}^{T} \sum_{j=0}^{T-i} \sum_{k=1}^{T} \sum_{\ell=0}^{T-k} \kappa_{2i}^{\infty} \kappa_{2k}^{\infty} \kappa_{2(j+1)}^{\infty} \kappa_{2(\ell+1)}^{\infty} \left( \frac{1}{\xi \tilde{\alpha}^2} \right)^{i+j+k+\ell}$$

$$\leq \epsilon \cdot \frac{20\gamma^2}{\alpha^2} \sum_{i=1}^{\infty} \sum_{j=0}^{\infty} \sum_{k=1}^{\infty} \sum_{\ell=0}^{\infty} \kappa_{2i}^{\infty} \kappa_{2k}^{\infty} \kappa_{2(j+1)}^{\infty} \kappa_{2(\ell+1)}^{\infty} \left( \frac{1}{\xi \tilde{\alpha}^2} \right)^{i+j+k+\ell}$$

$$\leq \epsilon \cdot \frac{20\gamma^2}{\alpha^2} \left( R \left( \frac{1}{\xi \tilde{\alpha}^2} \right) \right)^4,$$

where we use again that $\kappa_{2i} \geq 0$ for $i \geq 1$ and the power series expansion (A.15) of $R$. Since $\epsilon$ can be taken arbitrarily small, we deduce that

$$\lim_{T \to \infty} S_2 = 0. \tag{C.82}$$

By combining (C.74), (C.75), (C.76), (C.77), (C.79), (C.80) and (C.82), we conclude that

$$\lim_{T \to \infty} \lim_{m \to \infty} \frac{1}{\sqrt{m}} \left\| e^{T+1} \right\| = 0, \tag{C.83}$$

which, combined with (C.73), gives (C.69). Finally, by using (C.68) and (C.69), we have that

$$\lim_{T \to \infty} \lim_{m \to \infty} \frac{1}{\sqrt{m}} \left\| r^{T+1} \right\| = 0. \tag{C.84}$$

Thus, from the decomposition (C.65), we conclude that, as $m \to \infty$ and $T \to \infty$, $\tilde{u}^{T+1}$ is aligned with $u_{\mathrm{PCA}}$. Furthermore, from another application of Proposition C.1, we obtain

$$\lim_{T \to \infty} \lim_{m \to \infty} \frac{1}{\sqrt{m}} \| \tilde{u}^{T+1} \| = \lim_{T \to \infty} \frac{1}{\alpha} \sqrt{\tilde{\mu}_T^2 + \tilde{\sigma}_{T,T}} = 1, \tag{C.85}$$

which implies that $\lim_{T \to \infty} \lim_{m \to \infty} \zeta_{T+1} = 1$ and concludes the proof. $\qquad \square$

## C.5 Analysis for the Second Phase

As in the proof of the square case, we define a modified version of the true AMP algorithm, in which the memory coefficients $\{a_{t,i}, b_{t+1,i}\}_{i \in [1,t]}$ are replaced by deterministic values obtained from state evolution. This modified AMP is initialized with

$$\hat{\boldsymbol{u}}^1 = \sqrt{m}\, \boldsymbol{u}_{\text{PCA}}, \quad \hat{\boldsymbol{g}}^1 = \left(1 + \gamma \sum_{i=1}^{\infty} \kappa_{2i}^{\infty} \left(\frac{\gamma}{\alpha^2}\right)^i\right)^{-1} \boldsymbol{X}^{\mathsf{T}} \hat{\boldsymbol{u}}^1, \quad \hat{\boldsymbol{v}}^1 = v_1(\hat{\boldsymbol{g}}^1) = \frac{\gamma}{\alpha} \hat{\boldsymbol{g}}^1. \quad \text{(C.86)}$$

Then, for $t \geq 1$, we iteratively compute:

$$\hat{\boldsymbol{f}}^t = \boldsymbol{X}\hat{\boldsymbol{v}}^t - \sum_{i=1}^{t} \bar{\mathsf{a}}_{t,i}\hat{\boldsymbol{u}}^i, \quad \hat{\boldsymbol{u}}^{t+1} = \mathsf{u}_{t+1}(\hat{\boldsymbol{f}}^t), \quad \hat{\boldsymbol{g}}^{t+1} = \boldsymbol{X}\hat{\boldsymbol{u}}^{t+1} - \sum_{i=1}^{t} \bar{\mathsf{b}}_{t+1,i}\hat{\boldsymbol{v}}^i, \quad \hat{\boldsymbol{v}}^{t+1} = \mathsf{v}_{t+1}(\hat{\boldsymbol{g}}^{t+1}). \tag{C.87}$$

The deterministic memory coefficients are: $\bar{\mathsf{a}}_{1,1} = \alpha \sum_{i=1}^{\infty} \kappa_{2i}^{\infty} \left(\frac{\gamma}{\alpha^2}\right)^i$, and for $t \geq 2$:

$$\bar{\mathsf{a}}_{t,1} = \mathbb{E}\{\mathsf{v}'_t(G_t)\} \prod_{i=2}^{t} \mathbb{E}\{\mathsf{u}'_i(F_{i-1})\}\mathbb{E}\{\mathsf{v}'_{i-1}(G_{i-1})\} \left(\sum_{i=0}^{\infty} \kappa_{2(i+t)}^{\infty} \left(\frac{\gamma}{\alpha^2}\right)^i\right), \tag{C.88}$$

$$\bar{\mathsf{a}}_{t,t-j} = \mathbb{E}\{\mathsf{v}'_t(G_t)\} \prod_{i=t-j+1}^{t} \mathbb{E}\{\mathsf{u}'_i(F_{i-1})\}\mathbb{E}\{\mathsf{v}'_{i-1}(G_{i-1})\}\kappa_{2(j+1)}^{\infty}, \qquad \text{for } (t-j) \in [2, t]. \tag{C.89}$$

Furthermore, for $t \geq 1$,

$$\bar{\mathsf{b}}_{t+1,1} = \gamma \mathbb{E}\{\mathsf{u}'_{t+1}(F_t)\} \prod_{i=2}^{t} \mathbb{E}\{\mathsf{v}'_i(G_i)\}\mathbb{E}\{\mathsf{u}'_i(F_{i-1})\} \left(\kappa_{2t}^{\infty} + \sum_{i=1}^{\infty} \kappa_{2(i+t)}^{\infty} \left(\frac{\gamma}{\alpha^2}\right)^i\right), \tag{C.90}$$

$$\bar{\mathsf{b}}_{t+1,t+1-j} = \gamma \mathbb{E}\{\mathsf{u}'_{t+1}(F_t)\} \prod_{i=t+2-j}^{t} \mathbb{E}\{\mathsf{v}'_i(G_i)\}\mathbb{E}\{\mathsf{u}'_i(F_{i-1})\} \kappa_{2j}^{\infty}, \qquad \text{for } (t+1-j) \in [2, t]. \tag{C.91}$$

We recall that $\{\kappa_{2i}^{\infty}\}$ are the rectangular free cumulants of the limiting singular value distribution $\Lambda$, and the random variables $\{F_i, G_i\}$ are given by (3.15)-(3.17). The following lemma shows that, as $T$ grows, the iterates of the second phase of the artificial AMP (described in Section C.1) approach those of the modified AMP algorithm above, as do the corresponding state evolution parameters.

**Lemma C.9.** *Consider the setting of Theorem 2. Assume that $\kappa_{2i}^{\infty} \geq 0$ for all $i \geq 1$, and that $\tilde{\alpha} > \tilde{\alpha}_{\text{s}}$. Consider the modified version of the true AMP in (C.86)-(C.87), and the artificial AMP in (C.1), (C.2), and (C.6) along with its state evolution recursion given by (C.9)-(C.15). Then, the following results hold for $s, t \geq 1$:*

1.

$$\lim_{T \to \infty} \tilde{\mu}_{T+t} = \mu_t, \qquad \lim_{T \to \infty} \tilde{\sigma}_{T+s,T+t} = \sigma_{s,t}, \tag{C.92}$$

$$\lim_{T \to \infty} \tilde{\nu}_{T+t} = \nu_t, \qquad \lim_{T \to \infty} \tilde{\omega}_{T+s,T+t} = \omega_{s,t}, \tag{C.93}$$

2. *For any $\mathrm{PL}(2)$ functions $\psi : \mathbb{R}^{2t+2} \to \mathbb{R}$ and $\varphi : \mathbb{R}^{2t+1} \to \mathbb{R}$, we almost surely have:*

$$\lim_{T \to \infty} \lim_{n \to \infty} \left| \frac{1}{m} \sum_{i=1}^{m} \psi(u_i^*, \tilde{u}_i^{T+1}, \dots, \tilde{u}_i^{T+t+1}, \tilde{f}_i^{T+1}, \dots \tilde{f}_i^{T+t}) \right.$$
$$\left. - \frac{1}{m} \sum_{i=1}^{m} \psi(u_i^*, \hat{u}_i^1, \dots, \hat{u}_i^{t+1}, \hat{f}_i^1, \dots \hat{f}_i^t) \right| = 0, \tag{C.94}$$

$$\lim_{T \to \infty} \lim_{n \to \infty} \left| \frac{1}{n} \sum_{i=1}^{n} \varphi(v_i^*, \tilde{v}_i^{T+1}, \dots, \tilde{v}_i^{T+t}, \tilde{g}_i^{T+1}, \dots \tilde{g}_i^{T+t}) \right.$$
$$\left. - \frac{1}{n} \sum_{i=1}^{n} \varphi(v_i^*, \hat{v}_i^1, \dots, \hat{v}_i^t, \hat{g}_i^1, \dots \hat{g}_i^t) \right| = 0. \tag{C.95}$$

*Proof.* **Proof of** (C.92)- (C.93). For $t \in [1, T+1]$, from (C.18) we have $\tilde{\mu}_t = \tilde{\nu}_t = \alpha\sqrt{\Delta_{\mathrm{PCA}}} = \mu_1 = \nu_1$. Next, Lemma C.4 shows that $\lim_{T\to\infty} \tilde{\sigma}_{T+1,T+1} = a^*$ and $\lim_{T\to\infty} \tilde{\omega}_{T+1,T+1} = b^*$, where $a^*, b^*$ are defined in (C.20). We now verify that $\sigma_{11} = a^*$ and $\omega_{11} = b^*$. Setting $s = t = 0$ in (3.19) and solving for $\omega_{11}$, we obtain:

$$\omega_{1,1} = b^* = \frac{\Delta_{\mathrm{PCA}}\gamma\alpha^2(xR'(x) - R(x)) + \gamma R'(x)}{1 + \gamma R(x) - \gamma x R'(x)}, \quad \text{where } x = \frac{\gamma}{\alpha^2}. \tag{C.96}$$

Here, we have used (A.16) and (A.17) to express the double sums in terms of $R(x)$ and $R'(x)$. Similarly, from (3.20), we obtain

$$\sigma_{1,1} = \gamma x R'(x)(\alpha^2 \Delta_{\mathrm{PCA}} + \omega_{1,1}) + \gamma R'(x) - \alpha^2 R(x), \quad \text{where } x = \gamma/\alpha^2. \tag{C.97}$$

Using the formula for $\Delta_{\mathrm{PCA}}$ in [20, Eq. (7.32)], it can be verified that the above expression for $\sigma_{1,1}$ reduces to $a^* = \alpha^2(1 - \Delta_{\mathrm{PCA}})$, as required.

Assume towards induction that the following holds for $1 \le k, \ell \le t$:

$$\lim_{T\to\infty} \tilde{\mu}_{T+\ell} = \mu_\ell, \quad \lim_{T\to\infty} \tilde{\sigma}_{T+k,T+\ell} = \sigma_{k,\ell}, \quad \lim_{T\to\infty} \tilde{\nu}_{T+\ell} = \nu_\ell, \quad \lim_{T\to\infty} \tilde{\omega}_{T+k,T+\ell} = \omega_{k,\ell}. \tag{C.98}$$

Consider $\tilde{\nu}_{T+t+1} = \alpha\mathbb{E}\{\tilde{U}_{T+t+1}U_*\} = \alpha\mathbb{E}\{\mathsf{u}_{t+1}(\tilde{F}_{T+t})U_*\}$. By the induction hypothesis $\tilde{F}_{T+t} = \tilde{\mu}_{T+t}U_* + \tilde{Y}_{T+t}$ converges in distribution to $F_t = \mu_t U_* + Y_t$, and by arguments similar to (B.64), the sequence of random variables $\{\mathsf{u}_{t+1}(\tilde{F}_{T+t})U_*\}$ is uniformly integrable. Hence,

$$\lim_{T\to\infty} \tilde{\nu}_{T+t+1} = \alpha\mathbb{E}\{\mathsf{u}_{t+1}(F_t)U_*\} = \nu_{t+1}. \tag{C.99}$$

Next, for $s \le t$, consider $\tilde{\omega}_{T+s+1,T+t+1}$ which is defined via (C.14). We write $\tilde{\omega}_{T+s+1,T+t+1} = O_1 + O_2 + O_3 + O_4$, where

$$O_1 = \gamma \sum_{j=0}^{s-1}\sum_{k=0}^{t-1} \Big( \prod_{i=s-j+2}^{s+1} \mathbb{E}\{\mathsf{u}'_i(\tilde{F}_{T+i-1})\}\mathbb{E}\{\mathsf{v}'_{i-1}(\tilde{G}_{T+i-1})\} \Big)$$

$$\Big( \prod_{i=t-k+2}^{t+1} \mathbb{E}\{\mathsf{u}'_i(\tilde{F}_{T+i-1})\}\mathbb{E}\{\mathsf{v}'_{i-1}(\tilde{G}_{T+i-1})\} \Big) \cdot \Big[ \kappa_{2(j+k+1)}^\infty \mathbb{E}\{\tilde{U}_{T+s+1-j}\tilde{U}_{T+t+1-k}\}$$

$$+ \kappa_{2(j+k+2)}^\infty \mathbb{E}\{\mathsf{u}'_{s+1-j}(\tilde{F}_{T+s-j})\}\mathbb{E}\{\mathsf{u}'_{t+1-k}(\tilde{F}_{T+t-k})\}\mathbb{E}\{\tilde{V}_{T+s-j}\tilde{V}_{T+t-k}\} \Big], \tag{C.100}$$

$$O_2 = \gamma \sum_{j=0}^{s-1}\sum_{k=t}^{T+t} \Big(\frac{\gamma}{\alpha^2}\Big)^{k-t} \Big( \prod_{i=s-j+2}^{s+1} \mathbb{E}\{\mathsf{u}'_i(\tilde{F}_{T+i-1})\}\mathbb{E}\{\mathsf{v}'_{i-1}(\tilde{G}_{T+i-1})\} \Big)$$

$$\Big( \prod_{i=2}^{t+1} \mathbb{E}\{\mathsf{u}'_i(\tilde{F}_{T+i-1})\}\mathbb{E}\{\mathsf{v}'_{i-1}(\tilde{G}_{T+i-1})\} \Big) \cdot \Big[ \kappa_{2(j+k+1)}^\infty \mathbb{E}\{\tilde{U}_{T+s+1-j}\tilde{U}_{T+t+1-k}\}$$

$$+ \kappa_{2(j+k+2)}^\infty \frac{1}{\alpha} \mathbb{E}\{\mathsf{u}'_{s+1-j}(\tilde{F}_{T+s-j})\}\mathbb{E}\{\tilde{V}_{T+s-j}\tilde{V}_{T+t-k}\} \Big], \tag{C.101}$$

$$O_3 = \gamma \sum_{j=s}^{T+s}\sum_{k=0}^{t-1} \Big(\frac{\gamma}{\alpha^2}\Big)^{j-s} \Big( \prod_{i=2}^{s+1} \mathbb{E}\{\mathsf{u}'_i(\tilde{F}_{T+i-1})\}\mathbb{E}\{\mathsf{v}'_{i-1}(\tilde{G}_{T+i-1})\} \Big)$$

$$\Big( \prod_{i=t-k+2}^{t+1} \mathbb{E}\{\mathsf{u}'_i(\tilde{F}_{T+i-1})\}\mathbb{E}\{\mathsf{v}'_{i-1}(\tilde{G}_{T+i-1})\} \Big) \cdot \Big[ \kappa_{2(j+k+1)}^\infty \mathbb{E}\{\tilde{U}_{T+s+1-j}\tilde{U}_{T+t+1-k}\}$$

$$+ \kappa_{2(j+k+2)}^\infty \frac{1}{\alpha} \mathbb{E}\{\mathsf{u}'_{t+1-k}(\tilde{F}_{T+t-k})\}\mathbb{E}\{\tilde{V}_{T+s-j}\tilde{V}_{T+t-k}\} \Big], \tag{C.102}$$

$$O_4 = \gamma \sum_{j=s}^{T+s}\sum_{k=t}^{T+t} \Big(\frac{\gamma}{\alpha^2}\Big)^{j+k-s-t} \Big( \prod_{i=2}^{s+1} \mathbb{E}\{\mathsf{u}'_i(\tilde{F}_{T+i-1})\}\mathbb{E}\{\mathsf{v}'_{i-1}(\tilde{G}_{T+i-1})\} \Big)$$

$$\Big( \prod_{i=2}^{t+1} \mathbb{E}\{\mathsf{u}'_i(\tilde{F}_{T+i-1})\}\mathbb{E}\{\mathsf{v}'_{i-1}(\tilde{G}_{T+i-1})\} \Big) \cdot \Big[ \kappa_{2(j+k+1)}^\infty \mathbb{E}\{\tilde{U}_{T+s+1-j}\tilde{U}_{T+t+1-k}\}$$

$$+ \kappa_{2(j+k+2)}^\infty \frac{1}{\alpha^2} \mathbb{E}\{\tilde{V}_{T+s-j}\tilde{V}_{T+t-k}\} \Big]. \tag{C.103}$$

By the induction hypothesis, for $i \in [2, t+1]$, we have $\tilde{F}_{T+i-1} \overset{d}{\to} F_{i-1}$ and $\tilde{G}_{T+i-1} \overset{d}{\to} G_{i-1}$. Since $u_i$ and $v_{i-1}$ are Lipschitz and continuously differentiable, Lemma D.1 implies

$$\lim_{T \to \infty} \mathbb{E}\{u_i'(\tilde{F}_{T+i-1})\} = \mathbb{E}\{u_i'(F_{i-1})\}, \quad \lim_{T \to \infty} \mathbb{E}\{v_{i-1}'(\tilde{G}_{T+i-1})\} = \mathbb{E}\{v_{i-1}'(G_{i-1})\},$$
$$\text{for } i \in [2, t+1]. \tag{C.104}$$

Next, note that

$$(\tilde{U}_{T+s+1-j}, \tilde{V}_{T+s-j}) = \begin{cases} (u_{s+1-j}(\tilde{F}_{T+s-j}), v_{s-j}(\tilde{G}_{T+s-j})), & 0 \le j \le s-1, \\ (\tilde{F}_{T+s-j}/\alpha, \tilde{G}_{T+s-j}\gamma/\alpha), & s \le j \le T+s-1, \\ (\tilde{F}_0/\alpha, 0), & j = T+s. \end{cases} \tag{C.105}$$

An analogous set of expressions holds for the pair $(\tilde{U}_{T+t+1-k}, \tilde{V}_{T+t-k})$. For $j \in [0, s-1]$ and $k \in [0, t-1]$, using an argument similar to that used to obtain (B.73), we deduce that the sequences $\{u_{s+1-j}(\tilde{F}_{T+s-j}) u_{t+1-k}(\tilde{F}_{T+t-k})\}$ and $\{v_{s-j}(\tilde{G}_{T+s-j}) v_{t-k}(\tilde{G}_{T+t-k})\}$ are each uniformly integrable. This, together with the induction hypothesis, implies that

$$\lim_{T \to \infty} O_1 = \gamma \sum_{j=0}^{s-1} \sum_{k=0}^{t-1} \Big( \prod_{i=s-j+2}^{s+1} \mathbb{E}\{u_i'(F_{i-1})\} \mathbb{E}\{v_{i-1}'(G_{i-1})\} \Big)$$

$$\Big( \prod_{i=t-k+2}^{t+1} \mathbb{E}\{u_i'(F_{i-1})\} \mathbb{E}\{v_{i-1}'(G_{i-1})\} \Big) \cdot \Big[ \kappa_{2(j+k+1)}^\infty \mathbb{E}\{U_{s+1-j} U_{t+1-k}\} \tag{C.106}$$

$$+ \kappa_{2(j+k+2)}^\infty \mathbb{E}\{u_{s+1-j}'(F_{s-j})\} \mathbb{E}\{u_{t+1-k}'(F_{t-k})\} \mathbb{E}\{V_{s-j} V_{t-k}\} \Big].$$

Next consider the term $O_4$. In this case, for $j \in [s, T+s-1]$ and $k \in [t, T+t-1]$:

$$\mathbb{E}\{\tilde{U}_{T+s+1-j} \tilde{U}_{T+t+1-k}\} = \frac{1}{\alpha^2} \mathbb{E}\{\tilde{F}_{T+s-j} \tilde{F}_{T+t-k}\} = \Delta_{\text{PCA}} + \frac{1}{\alpha^2} \tilde{\sigma}_{T-(j-s), T-(k-t)},$$
$$\mathbb{E}\{\tilde{V}_{T+s-j} \tilde{V}_{T+t-k}\} = \frac{\gamma^2}{\alpha^2} \mathbb{E}\{\tilde{G}_{T+s-j} \tilde{G}_{T+t-k}\} = \frac{\gamma^2}{\alpha^2}(\alpha^2 \Delta_{\text{PCA}} + \tilde{\omega}_{T-(j-s), T-(k-t)}). \tag{C.107}$$

When $j = T+s$ or $k = T+t$, the formula above for $\mathbb{E}\{\tilde{U}_{T+s+1-j} \tilde{U}_{T+t+1-k}\}$ still holds, while the one for $\mathbb{E}\{\tilde{V}_{T+s-j} \tilde{V}_{T+t-k}\}$ becomes 0 as $\tilde{V}_0 = 0$. From Lemma C.2, for any $\delta > 0$, for sufficiently large $T$ we have

$$|\tilde{\sigma}_{T+s-j, T+t-k} - a^*| < \delta \xi^{-\max\{j+1-s, k+1-t\}},$$
$$|\tilde{\omega}_{T+s-j, T+t-k} - b^*| < \delta \xi^{-\max\{j+1-s, k+1-t\}}, \quad j \in [s, T+s], \ k \in [t, T+t], \tag{C.108}$$

for some $\xi > 0$ such that $\tilde{\alpha}\sqrt{\xi} > \tilde{\alpha}_s$. From (3.15)-(3.18), we note that $\mathbb{E}\{U_{s-j} U_{t-k}\} = \frac{1}{\alpha^2} \mathbb{E}\{F_0^2\} = 1$ and $\mathbb{E}\{V_{s-j} V_{t-k}\} = \frac{\gamma^2}{\alpha^2} \mathbb{E}\{G_1^2\} = \frac{\gamma^2}{\alpha^2}(\alpha^2 \Delta_{\text{PCA}} + b^*)$. Combining this with (C.107) and (C.108), we have for sufficiently large $T$:

$$|\mathbb{E}\{\tilde{U}_{T+1+s-j} \tilde{U}_{T+1+t-k}\} - \mathbb{E}\{U_{s-j} U_{t-k}\}| < \frac{\delta}{\alpha^2} \xi^{-\max\{j+1-s, k+1-t\}},$$

$$|\mathbb{E}\{\tilde{V}_{T+s-j} \tilde{V}_{T+t-k}\} - \mathbb{E}\{V_{s-j} V_{t-k}\}| < \frac{\gamma^2 \delta}{\alpha^2} \xi^{-\max\{j+1-s, k+1-t\}}, \text{ for } j \ge s, k \ge t. \tag{C.109}$$

We now write $O_4$ in (C.103) as

$$O_4 = \gamma \Big( \prod_{i=2}^{s+1} \mathbb{E}\{u_i'(\tilde{F}_{T+i-1})\} \mathbb{E}\{v_{i-1}'(\tilde{G}_{T+i-1})\} \Big) \Big( \prod_{i=2}^{t+1} \mathbb{E}\{u_i'(\tilde{F}_{T+i-1})\} \mathbb{E}\{v_{i-1}'(\tilde{G}_{T+i-1})\} \Big)$$

$$\Big[ \sum_{j=s}^{T+s} \sum_{k=t}^{T+t} \Big( \frac{\gamma}{\alpha^2} \Big)^{j+k-s-t} \Big[ \kappa_{2(j+k+1)}^\infty \mathbb{E}\{U_{s+1-j} U_{t+1-k}\} + \kappa_{2(j+k+2)}^\infty \frac{1}{\alpha^2} \mathbb{E}\{V_{s-j} V_{t-k}\} \Big]$$

$$+ \Delta_{4U} + \Delta_{4V} \Big], \tag{C.110}$$

where

$$\Delta_{4U} = \sum_{j=s}^{T+s} \sum_{k=t}^{T+t} \left(\frac{\gamma}{\alpha^2}\right)^{j+k-s-t} \kappa_{2(j+k+1)}^{\infty} [\mathbb{E}\{\tilde{U}_{T+1+s-j}\tilde{U}_{T+1+t-k}\} - \mathbb{E}\{U_{s+1-j}U_{t+1-k}\}],$$

$$\Delta_{4V} = \frac{1}{\alpha^2} \sum_{j=s}^{T+s} \sum_{k=t}^{T+t} \left(\frac{\gamma}{\alpha^2}\right)^{j+k-s-t} \kappa_{2(j+k+2)}^{\infty} [\mathbb{E}\{\tilde{V}_{T+s-j}\tilde{V}_{T+t-k}\} - \mathbb{E}\{V_{s-j}V_{t-k}\}].$$

(C.111)

Using (C.109), for sufficiently large $T$ we have

$$|\Delta_{4U}| < \frac{\delta}{\alpha^2} \sum_{j=0}^{T} \sum_{k=0}^{T} \left(\frac{\gamma}{\xi\alpha^2}\right)^{j+k} \kappa_{2(j+k+s+t+1)}^{\infty} < \delta C_{s,t},$$

$$|\Delta_{4V}| < \frac{\gamma^2\delta}{\alpha^2} \sum_{j=0}^{T} \sum_{k=0}^{T} \left(\frac{\gamma}{\xi\alpha^2}\right)^{j+k} \kappa_{2(j+k+s+t+2)}^{\infty} < \delta C_{s,t},$$

(C.112)

for a positive constant $C_{s,t}$, since each of the double sums in (C.112) is bounded as $T \to \infty$, for $\xi\tilde{\alpha}^2 := \xi\alpha^2/\gamma > \tilde{\alpha}_s^2$. Therefore, $\Delta_{4U}, \Delta_{4V}$ both tend to 0 as $T \to \infty$. Using this in (C.110) along with (C.104), we obtain

$$\lim_{T\to\infty} O_4 = \gamma \prod_{i=2}^{s+1} \mathbb{E}\{u_i'(F_{i-1})\}\mathbb{E}\{v_{i-1}'(G_{i-1})\} \prod_{i=2}^{t+1} \mathbb{E}\{u_i'(F_{i-1})\}\mathbb{E}\{v_{i-1}'(G_{i-1})\}$$

$$\sum_{j=s}^{\infty} \sum_{k=t}^{\infty} \left(\frac{\gamma}{\alpha^2}\right)^{j+k-s-t} \left[\kappa_{2(j+k+1)}^{\infty}\mathbb{E}\{U_{s+1-j}U_{t+1-k}\} + \kappa_{2(j+k+2)}^{\infty}\frac{1}{\alpha^2}\mathbb{E}\{V_{s-j}V_{t-k}\}\right].$$

(C.113)

Next, consider $O_2$ in (C.101), which we write as

$$O_2 = \gamma \left(\prod_{i=2}^{t+1} \mathbb{E}\{u_i'(\tilde{F}_{T+i-1})\}\mathbb{E}\{v_{i-1}'(\tilde{G}_{T+i-1})\}\right) \sum_{j=0}^{s-1} \prod_{i=s-j+2}^{s+1} \mathbb{E}\{u_i'(\tilde{F}_{T+i-1})\}\mathbb{E}\{v_{i-1}'(\tilde{G}_{T+i-1})\}$$

$$\left[\sum_{k=t}^{T+t} \left(\frac{\gamma}{\alpha^2}\right)^{k-t} \left[\kappa_{2(j+k+1)}^{\infty}\mathbb{E}\{U_{s+1-j}U_{t+1-k}\} + \frac{\kappa_{2(j+k+2)}^{\infty}}{\alpha}\mathbb{E}\{u_{s+1-j}'(\tilde{F}_{T+s-j})\}\mathbb{E}\{V_{s-j}V_{t-k}\}\right]\right.$$

$$\left.\Delta_{3U,j} + \Delta_{3V,j}\right],$$

(C.114)

where

$$\Delta_{3U,j} = \sum_{k=t}^{T+t} \left(\frac{\gamma}{\alpha^2}\right)^{k-t} \kappa_{2(j+k+1)}^{\infty} [\mathbb{E}\{\tilde{U}_{T+s+1-j}\tilde{U}_{T+t+1-k}\} - \mathbb{E}\{U_{s+1-j}U_{t+1-k}\}],$$

$$\Delta_{3V,j} = \frac{1}{\alpha}\mathbb{E}\{u_{s+1-j}'(\tilde{F}_{T+s-j})\} \sum_{k=t}^{T+t} \left(\frac{\gamma}{\alpha^2}\right)^{k-t} \kappa_{2(j+k+2)}^{\infty} [\mathbb{E}\{\tilde{V}_{T+s-j}\tilde{V}_{T+t-k}\} - \mathbb{E}\{V_{s-j}V_{t-k}\}].$$

(C.115)

From (C.105), we recall that for $j \in [0, s-1]$, $k \in [t, T+t]$:

$$\mathbb{E}\{\tilde{U}_{T+s+1-j}\tilde{U}_{T+t+1-k}\} = \frac{1}{\alpha}\mathbb{E}\{u_{s+1-j}(\tilde{F}_{T+s-j})\tilde{F}_{T-(k-t)}\},$$

$$\mathbb{E}\{\tilde{V}_{T+s-j}\tilde{V}_{T+t-k}\} = \frac{\gamma}{\alpha}\mathbb{E}\{v_{s-j}(\tilde{G}_{T+s-j})\tilde{G}_{T-(k-t)}\}.$$

(C.116)

Using the induction hypothesis and arguments similar to (B.82)-(B.86), for any $\delta > 0$ and sufficiently large $T$ we have

$$|\mathbb{E}\{\tilde{U}_{T+s+1-j}\tilde{U}_{T+t+1-k}\} - \mathbb{E}\{U_{s+1-j}U_{t+1-k}\}| < \frac{\delta}{\alpha}\xi^{-(k-t)},$$

$$|E\{\tilde{V}_{T+s-j}\tilde{V}_{T+t-k}\} - E\{V_{s-j}V_{t-k}\}| < \frac{\gamma\delta}{\alpha}\xi^{-(k-t)}, \quad j \in [0, s-1], \ k \in [t, T+t].$$

(C.117)

Using this in (C.115), following steps similar to (B.88) and (B.89), and noting the convergence of the power series defining $R(\gamma/\xi\alpha^2)$, we have $\lim_{T\to\infty}\Delta_{3U,j} = \lim_{T\to\infty}\Delta_{3V,j} = 0$ for $j \in [0, s-1]$. Using this in (C.114) along with (C.104), we have

$$\lim_{T\to\infty} O_2 = \gamma\Big(\prod_{i=2}^{t+1}\mathbb{E}\{u_i'(F_{i-1})\}\mathbb{E}\{v_{i-1}'(G_{i-1})\}\Big)\sum_{j=0}^{s-1}\prod_{i=s-j+2}^{s+1}\mathbb{E}\{u_i'(F_{i-1})\}\mathbb{E}\{v_{i-1}'(G_{i-1})\}$$

$$\sum_{k=t}^{\infty}\Big(\frac{\gamma}{\alpha^2}\Big)^{k-t}\Big[\kappa_{2(j+k+1)}^{\infty}\mathbb{E}\{U_{s+1-j}U_{t+1-k}\} + \kappa_{2(j+k+2)}^{\infty}\frac{1}{\alpha}\mathbb{E}\{u_{s+1-j}'(F_{s-j})\}\mathbb{E}\{V_{s-j}V_{t-k}\}\Big].$$

(C.118)

Using a similar sequence of steps, we also have

$$\lim_{T\to\infty} O_3 = \gamma\Big(\prod_{i=2}^{s+1}\mathbb{E}\{u_i'(F_{i-1})\}\mathbb{E}\{v_{i-1}'(G_{i-1})\}\Big)\sum_{k=0}^{t-1}\prod_{i=t-k+2}^{t+1}\mathbb{E}\{u_i'(F_{i-1})\}\mathbb{E}\{v_{i-1}'(G_{i-1})\}$$

$$\sum_{j=s}^{\infty}\Big(\frac{\gamma}{\alpha^2}\Big)^{j-s}\Big[\kappa_{2(j+k+1)}^{\infty}\mathbb{E}\{U_{s+1-j}U_{t+1-k}\} + \kappa_{2(j+k+2)}^{\infty}\frac{1}{\alpha}\mathbb{E}\{u_{t+1-k}'(F_{t-k})\}\mathbb{E}\{V_{s-j}V_{t-k}\}\Big].$$

(C.119)

Noting that the sums of the limits in (C.106), (C.113), (C.118) and (C.119) equals $\omega_{s+1,t+1}$ (defined in (3.19)), we have shown that $\lim_{T\to\infty}\tilde{\omega}_{T+s+1,T+t+1} = \omega_{s+1,t+1}$. The sequence of steps to show that $\lim_{T\to\infty}\tilde{\sigma}_{T+s+1,T+t+1} = \sigma_{s+1,t+1}$ is very similar, and is omitted to avoid repetition.

**Proof of** (C.94)-(C.95). Since $\psi, \varphi \in \mathrm{PL}(2)$, using the Cauchy-Schwarz inequality (as in (B.92)), for a universal constant $C > 0$ we have

$$\left|\frac{1}{m}\sum_{i=1}^{m}\psi(u_i^*, \tilde{u}_i^{T+1}, \dots, \tilde{u}_i^{T+t+1}, \tilde{f}_i^{T+1}, \dots \tilde{f}_i^{T+t}) - \frac{1}{m}\sum_{i=1}^{m}\psi(u_i^*, \hat{u}_i^1, \dots, \hat{u}_i^{t+1}, \hat{f}_i^1, \dots \hat{f}_i^t)\right|$$

$$\leq 2C(t+2)\left[1 + \frac{\|\boldsymbol{u}^*\|^2}{m} + \sum_{\ell=1}^{t+1}\Big(\frac{\|\tilde{\boldsymbol{u}}^{T+\ell}\|^2}{m} + \frac{\|\hat{\boldsymbol{u}}^\ell\|^2}{m}\Big) + \sum_{\ell=1}^{t}\Big(\frac{\|\tilde{\boldsymbol{f}}^{T+\ell}\|^2}{m} + \frac{\|\hat{\boldsymbol{f}}^\ell\|^2}{m}\Big)\right]^{\frac{1}{2}}$$

$$\cdot\left(\frac{\|\tilde{\boldsymbol{u}}^{T+1} - \hat{\boldsymbol{u}}^1\|^2}{m} + \dots + \frac{\|\tilde{\boldsymbol{u}}^{T+t+1} - \hat{\boldsymbol{u}}^{t+1}\|^2}{m} + \frac{\|\tilde{\boldsymbol{f}}^{T+1} - \hat{\boldsymbol{f}}^1\|^2}{m} + \dots + \frac{\|\tilde{\boldsymbol{f}}^{T+t} - \hat{\boldsymbol{f}}^t\|^2}{m}\right)^{\frac{1}{2}},$$

(C.120)

$$\left|\frac{1}{n}\sum_{i=1}^{n}\varphi(v_i^*, \tilde{v}_i^{T+1}, \dots, \tilde{v}_i^{T+t}, \tilde{g}_i^{T+1}, \dots \tilde{g}_i^{T+t}) - \frac{1}{n}\sum_{i=1}^{n}\varphi(v_i^*, \hat{v}_i^1, \dots, \hat{v}_i^t, \hat{g}_i^1, \dots \hat{g}_i^t)\right|$$

$$\leq 2C(t+2)\left[1 + \frac{\|\boldsymbol{v}^*\|^2}{n} + \sum_{\ell=1}^{t}\Big(\frac{\|\tilde{\boldsymbol{v}}^{T+\ell}\|^2}{n} + \frac{\|\hat{\boldsymbol{v}}^\ell\|^2}{n}\Big) + \sum_{\ell=1}^{t}\Big(\frac{\|\tilde{\boldsymbol{g}}^{T+\ell}\|^2}{n} + \frac{\|\hat{\boldsymbol{g}}^\ell\|^2}{n}\Big)\right]^{\frac{1}{2}}$$

$$\cdot\left(\frac{\|\tilde{\boldsymbol{v}}^{T+1} - \hat{\boldsymbol{v}}^1\|^2}{n} + \dots + \frac{\|\tilde{\boldsymbol{v}}^{T+t} - \hat{\boldsymbol{v}}^t\|^2}{n} + \frac{\|\tilde{\boldsymbol{g}}^{T+1} - \hat{\boldsymbol{g}}^1\|^2}{n} + \dots + \frac{\|\tilde{\boldsymbol{g}}^{T+t} - \hat{\boldsymbol{g}}^t\|^2}{n}\right)^{\frac{1}{2}}.$$

(C.121)

The proof strategy is similar to the square case. We inductively show that in the limit $T, n \to \infty$ (with the limit in $n$ taken first): i) the terms in the last line of (C.120) and (C.121) all converge to 0 almost surely, and ii) each of the terms within the square brackets in (C.120) and (C.121) converges to a finite deterministic value.

Base case $t = 1$: Recalling that $\hat{\boldsymbol{u}}^1 = \sqrt{m}\boldsymbol{u}_{\mathrm{PCA}}$, from Lemma C.8, we have

$$\lim_{T\to\infty}\lim_{m\to\infty}\frac{\|\tilde{\boldsymbol{u}}^{T+1} - \hat{\boldsymbol{u}}^1\|^2}{m} = 0.$$

(C.122)

Writing $x = \gamma/\alpha^2$ for brevity, recall that $\sum_{i=1}^{\infty} \kappa_{2i}^{\infty} x^i = R(x)$. From the definitions of $\tilde{g}^{T+1}$ and $\hat{g}^1$ in (C.2) and (C.86) and we have

$$\tilde{g}^{T+1} - \hat{g}^1 = \frac{1}{1 + \gamma R(x)} X^{\mathsf{T}}(\tilde{u}^{T+1} - \hat{u}^1) + \left[ \frac{\gamma R(x)}{1 + \gamma R(x)} X^{\mathsf{T}} \tilde{u}^{T+1} - \alpha \sum_{j=1}^{T} \kappa_{2j} x^j \tilde{v}^{T+1-j} \right],$$

where we have used $\tilde{\mathsf{b}}_{T+1,T+1-j} = \alpha \kappa_{2j} x^j$ for $j \in [1, T]$. Therefore

$$
\begin{aligned}
\frac{\|\tilde{g}^{T+1} - \hat{g}^1\|^2}{n} &\leq \frac{2}{(1 + \gamma R(x))^2} \|X\|_{\mathrm{op}}^2 \frac{\|\tilde{u}^{T+1} - \hat{u}^1\|^2}{n} \\
&+ \frac{2}{n} \left\| \frac{\gamma R(x)}{1 + \gamma R(x)} X^{\mathsf{T}} \tilde{u}^{T+1} - \alpha \sum_{j=1}^{T} \kappa_{2j} x^j \tilde{v}^{T+1-j} \right\|^2 =: 2(S_1 + S_2).
\end{aligned}
\tag{C.123}
$$

Since $\|X\|_{\mathrm{op}} \overset{n \to \infty}{\longrightarrow} D^{-1}(x)$, from (C.122) we have $\lim_{T,n \to \infty} S_1 = 0$. (Here and in the remainder of the proof, $\lim_{T,n \to \infty}$ denotes the limit $n \to \infty$ taken first and then $T \to \infty$.) Next, using the definition of $\tilde{g}^{T+1}$ in (C.2), we write the second term $S_2$ as

$$
\begin{aligned}
S_2 &= \frac{1}{n} \left\| \frac{\gamma R(x)}{1 + \gamma R(x)} \tilde{g}^{T+1} - \frac{\alpha}{1 + \gamma R(x)} \sum_{j=1}^{T} \kappa_{2j} x^j \tilde{v}^{T+1-j} \right\|^2 \\
&\leq \frac{2}{n} \left\| \frac{\gamma R(x)}{1 + \gamma R(x)} \tilde{g}^{T+1} - \frac{\alpha}{1 + \gamma R(x)} \sum_{j=1}^{T} \kappa_{2j}^{\infty} x^j \tilde{v}^{T+1-j} \right\|^2 + \frac{2\alpha^2}{(1 + \gamma R(x))^2} \Delta_{S_2},
\end{aligned}
\tag{C.124}
$$

where

$$\Delta_{S_2} := \frac{1}{n} \left\| \sum_{j=1}^{T} (\kappa_{2j}^{\infty} - \kappa_{2j}) x^j \tilde{v}^{T+1-j} \right\|^2 = \frac{1}{n} \sum_{i,j=1}^{T} (\kappa_{2i}^{\infty} - \kappa_{2i})(\kappa_{2j}^{\infty} - \kappa_{2j}) x^{i+j} \frac{\langle \tilde{v}^{T+1-i}, \tilde{v}^{T+1-j} \rangle}{n}.$$
$$\tag{C.125}$$

Using the state evolution result of Proposition C.1, we almost surely have

$$
\begin{aligned}
\lim_{n \to \infty} \frac{\langle \tilde{v}^{T+1-i}, \tilde{v}^{T+1-j} \rangle}{n} &= \mathbb{E}\{\tilde{V}_{T+1-i} \tilde{V}_{T+1-j}\} \\
&= \frac{\gamma^2}{\alpha^2} (\alpha^2 \Delta_{\mathrm{PCA}} + \tilde{\omega}_{T+1-i,T+1-j}) < C,
\end{aligned}
\tag{C.126}
$$

for some universal constant $C > 0$. Here, $\tilde{\omega}_{T+1-i,T+1-j}$ is defined in (C.14), and we recall from (C.12)-(C.13) that

$$\tilde{V}_{T+1-j} = \frac{\gamma}{\alpha} \tilde{G}_{T+1-j} \quad \text{with} \quad \tilde{G}_{T+1-j} = \alpha \sqrt{\Delta_{\mathrm{PCA}}} V_* + \tilde{Z}_{T+1-j}, \quad \text{for } j \in [0, T]. \tag{C.127}$$

Since $\kappa_{2i} \to \kappa_{2i}^{\infty}$ as $n \to \infty$, for $i \in [1, T]$ (by the model assumptions), using (C.126) in (C.125),

$$\lim_{T \to \infty} \lim_{n \to \infty} \Delta_{S_2} = 0 \quad \text{almost surely.} \tag{C.128}$$

Next, using Proposition C.1, for any $T > 0$, the first term in (C.124) has the following almost sure limit as $n \to \infty$:

$$
\begin{aligned}
&\lim_{n \to \infty} \frac{1}{n} \left\| \frac{\gamma R(x)}{1 + \gamma R(x)} \tilde{g}^{T+1} - \frac{\alpha}{1 + \gamma R(x)} \sum_{j=1}^{T} \kappa_{2j}^{\infty} x^j \tilde{v}^{T+1-j} \right\|^2 \\
&= \mathbb{E}\left\{ \left( \frac{\gamma R(x)}{1 + \gamma R(x)} \tilde{G}_{T+1} - \frac{\alpha}{1 + \gamma R(x)} \sum_{j=1}^{T} \kappa_{2j}^{\infty} x^j \tilde{V}_{T+1-j} \right)^2 \right\} \\
&\overset{(a)}{=} \frac{\gamma^2}{(1 + \gamma R(x))^2} \mathbb{E}\left\{ \left( \left( R(x) - \sum_{j=1}^{T} \kappa_{2j}^{\infty} x^j \right) \tilde{G}_{T+1} + \sum_{j=1}^{T} \kappa_{2j}^{\infty} x^j (\tilde{G}_{T+1} - \tilde{G}_{T+1-j}) \right)^2 \right\},
\end{aligned}
\tag{C.129}
$$

where (a) is obtained using (C.127). From (A.15), we have $\lim_{T\to\infty}\sum_{j=1}^{T}\kappa_{2j}^{\infty}x^j = R(x)$. Furthermore, using (C.127) we have

$$\mathbb{E}\left\{\left(\sum_{j=1}^{T}\kappa_{2j}^{\infty}x^j(\tilde{G}_{T+1} - \tilde{G}_{T+1-j})\right)^2\right\}$$

$$= \sum_{i,j=1}^{T}\kappa_{2i}^{\infty}\kappa_{2j}^{\infty}\,x^{i+j}\left(\tilde{\omega}_{T+1,T+1} - \tilde{\omega}_{T+1,T+1-i} - \tilde{\omega}_{T+1,T+1-j} + \tilde{\omega}_{T+1-i,T+1-j}\right)$$

$$\longrightarrow 0 \ \text{ as } \ T \to \infty, \tag{C.130}$$

where the $T \to \infty$ limit is obtained using Lemma C.2 and steps similar to (B.50)-(B.53). Using (C.128)-(C.130) in (C.124), we have

$$\lim_{T\to\infty}\lim_{n\to\infty} S_2 = 0 \quad \text{almost surely.} \tag{C.131}$$

Hence using (C.123), we have shown that $\lim_{T,n\to\infty}\frac{1}{n}\|\tilde{\boldsymbol{g}}^{T+1} - \hat{\boldsymbol{g}}^{1}\|^2 = 0$ almost surely.

The proof that $\lim_{T,n\to\infty}\frac{1}{n}\|\tilde{\boldsymbol{f}}^{T+1} - \hat{\boldsymbol{f}}^{1}\|^2 = 0$ uses similar steps: from the definitions of $\tilde{\boldsymbol{f}}^{T+1}$ and $\hat{\boldsymbol{f}}^{1}$ in (C.2) and (C.87), we have

$$\tilde{\boldsymbol{f}}^{T+1} - \hat{\boldsymbol{f}}^{1} = \frac{\gamma}{\alpha}\boldsymbol{X}(\tilde{\boldsymbol{g}}^{T+1} - \hat{\boldsymbol{g}}^{1}) + \bar{\mathsf{a}}_{1,1}\hat{\boldsymbol{u}}^{1} - \sum_{j=0}^{T}\tilde{\mathsf{a}}_{T+1,T+1-j}\tilde{\boldsymbol{u}}^{T+1-j}, \tag{C.132}$$

where $\bar{\mathsf{a}}_{1,1} = \alpha\sum_{j=0}^{\infty}\kappa_{2(j+1)}^{\infty}x^{j+1}$ and $\tilde{\mathsf{a}}_{T+1,T+1-j} = \alpha\kappa_{2(j+1)}x^{j+1}$ for $j \in [0,T]$. Therefore,

$$\frac{\|\tilde{\boldsymbol{f}}^{T+1} - \hat{\boldsymbol{f}}^{1}\|^2}{n} \leq \frac{5\gamma^2}{\alpha^2}\|\boldsymbol{X}\|_{\text{op}}^2 \frac{\|\tilde{\boldsymbol{g}}^{T+1} - \hat{\boldsymbol{g}}^{1}\|^2}{n} + 5\bar{\mathsf{a}}_{1,1}^2\frac{\|\hat{\boldsymbol{u}}^{1} - \tilde{\boldsymbol{u}}^{T+1}\|^2}{n}$$

$$+ \frac{5}{n}\left\|\alpha\sum_{j=0}^{T}\kappa_{2(j+1)}^{\infty}x^{j+1}(\tilde{\boldsymbol{u}}^{T+1} - \tilde{\boldsymbol{u}}^{T+1-j})\right\|^2 + 5\alpha^2\left(\sum_{j=T+1}^{\infty}\kappa_{2(j+1)}^{\infty}x^{j+1}\right)^2\frac{\|\hat{\boldsymbol{u}}^{1}\|^2}{n}$$

$$+ \frac{5}{n}\left\|\alpha\sum_{j=0}^{T}(\kappa_{2(j+1)}^{\infty} - \kappa_{2(j+1)})x^{j+1}\tilde{\boldsymbol{u}}^{T+1-j}\right\|^2. \tag{C.133}$$

We have shown $\lim_{T,n\to\infty}\frac{1}{n}\|\tilde{\boldsymbol{g}}^{T+1} - \hat{\boldsymbol{g}}^{1}\|^2 = 0$ and $\lim_{T,n\to\infty}\frac{1}{n}\|\tilde{\boldsymbol{u}}^{T+1} - \hat{\boldsymbol{u}}^{1}\|^2$, hence the first two terms in (C.133) converge to 0. For the third term in (C.133), we first apply Proposition C.1 to express the $n \to \infty$ limit in terms of state evolution parameters of the artificial AMP, which can then be shown to converge to 0 as $T \to \infty$ using Lemma C.2 and steps similar to (B.50)-(B.53). Since the power series $\sum_{j=0}^{\infty}\kappa_{2(j+1)}^{\infty}x^{j+1} = R(x)$ converges, and $\|\hat{\boldsymbol{u}}^{1}\|^2/n = m/n = \gamma$, the fourth term converges to 0 as $T, n \to \infty$. As $\kappa_{2(j+1)} \to \kappa_{2(j+1)}^{\infty}$ as $n \to \infty$, by arguments similar to (B.45)-(B.46), the final term in (C.133) also converges to 0.

Recalling that $\tilde{\boldsymbol{v}}^{T+1} - \hat{\boldsymbol{v}}^{1} = \frac{\gamma}{\alpha}(\tilde{\boldsymbol{g}}^{T+1} - \hat{\boldsymbol{g}}^{1})$, it follows that $\lim_{T,n\to\infty}\frac{1}{n}\|\tilde{\boldsymbol{v}}^{T+1} - \hat{\boldsymbol{v}}^{1}\|^2 = 0$ almost surely. Finally, a triangle inequality sandwiching argument like the one used in (B.104)-(B.105) yields

$$\lim_{T\to\infty}\lim_{n\to\infty}\frac{\|\tilde{\boldsymbol{v}}^{T+1}\|^2}{n} = \lim_{T\to\infty}\lim_{n\to\infty}\frac{\|\hat{\boldsymbol{v}}^{1}\|^2}{n} = \frac{\gamma^2}{\alpha^2}(\alpha^2\Delta_{\text{PCA}} + \omega_{1,1}),$$

$$\lim_{T\to\infty}\lim_{n\to\infty}\frac{\|\tilde{\boldsymbol{u}}^{T+1}\|^2}{m} = \lim_{T\to\infty}\lim_{n\to\infty}\frac{\|\hat{\boldsymbol{u}}^{1}\|^2}{m} = 1. \tag{C.134}$$

This completes the proof of (C.94)-(C.95) for $t = 1$.

Induction step: For $t \geq 1$, assume that the following hold almost surely for $\ell \in [1,t]$:

$$\lim_{T\to\infty}\lim_{n\to\infty}\frac{\|\hat{\boldsymbol{u}}^{\ell} - \tilde{\boldsymbol{u}}^{T+\ell}\|^2}{m} = \lim_{T\to\infty}\lim_{n\to\infty}\frac{\|\hat{\boldsymbol{g}}^{\ell} - \tilde{\boldsymbol{g}}^{T+\ell}\|^2}{n} = \lim_{T\to\infty}\lim_{n\to\infty}\frac{\|\hat{\boldsymbol{v}}^{\ell} - \tilde{\boldsymbol{v}}^{T+\ell}\|^2}{n} = 0. \tag{C.135}$$

We now show that $\lim_{T,n\to\infty} \frac{1}{n}\|\tilde{\boldsymbol{f}}^{T+t} - \hat{\boldsymbol{f}}^{t}\|^2 = 0$. We have already shown this for $t = 1$ above. For $t \geq 2$, using the definitions $\tilde{\boldsymbol{f}}^{T+t}$ and $\hat{\boldsymbol{f}}^{t}$ in (C.2) and (C.87), and applying the Cauchy-Schwarz inequality, we have

$$
\frac{1}{n}\|\tilde{\boldsymbol{f}}^{T+t} - \hat{\boldsymbol{f}}^{t}\|^2 \leq \frac{(t+1)}{n}\left( \|\boldsymbol{X}(\tilde{\boldsymbol{v}}^{T+t} - \hat{\boldsymbol{v}}^t)\|^2 + \sum_{\ell=2}^{t} \|\tilde{\mathsf{a}}_{T+t,T+\ell}\tilde{\boldsymbol{u}}^{T+\ell} - \bar{\mathsf{a}}_{t,\ell}\hat{\boldsymbol{u}}^\ell\|^2 \right.
$$
$$
\left. + \left\| \sum_{i=1}^{T+1} \tilde{\mathsf{a}}_{T+t,i}\tilde{\boldsymbol{u}}^i - \bar{\mathsf{a}}_{t,1}\hat{\boldsymbol{u}}^1 \right\|^2 \right). \tag{C.136}
$$

The decomposition and the analysis of the three terms in (C.136) is similar to that in (B.109) for the square case. Using arguments similar to (B.110)-(B.127), we obtain $\lim_{T,n\to\infty} \frac{1}{n}\|\tilde{\boldsymbol{f}}^{T+t} - \hat{\boldsymbol{f}}^{t}\|^2 = 0$. Recalling that $\hat{\boldsymbol{u}}^{t+1} = \mathsf{u}_{t+1}(\hat{\boldsymbol{f}}^t)$ and $\tilde{\boldsymbol{u}}^{T+t+1} = \mathsf{u}_{t+1}(\tilde{\boldsymbol{f}}^{T+t})$ with $\mathsf{u}_{t+1}$ Lipschitz, we also have $\lim_{T,n\to\infty} \frac{1}{n}\|\tilde{\boldsymbol{u}}^{T+t+1} - \hat{\boldsymbol{u}}^{t+1}\|^2 = 0$ almost surely. The proof that $\lim_{T,n\to\infty} \frac{1}{n}\|\tilde{\boldsymbol{g}}^{T+t+1} - \hat{\boldsymbol{g}}^{t+1}\|^2 = 0$ uses a decomposition similar to (C.136) and is along the same lines. Since $\hat{\boldsymbol{v}}^{t+1} = \mathsf{v}_{t+1}(\hat{\boldsymbol{g}}^{t+1})$ and $\tilde{\boldsymbol{v}}^{T+t+1} = \mathsf{v}_{t+1}(\tilde{\boldsymbol{g}}^{T+t+1})$ with $\mathsf{v}_{t+1}$ Lipschitz, it follows that $\lim_{T,n\to\infty} \frac{1}{n}\|\tilde{\boldsymbol{v}}^{T+t+1} - \hat{\boldsymbol{v}}^{t+1}\|^2 = 0$ almost surely.

Using these results together with a triangle inequality sandwich argument similar to (B.104)-(B.105), we have $\lim_{n\to\infty} \frac{1}{n}\|\hat{\boldsymbol{u}}^{t+1}\|^2 = \lim_{T,n\to\infty} \|\tilde{\boldsymbol{u}}^{T+t+1}\|^2 = \mathbb{E}\{\mathsf{u}_{t+1}(F_t)^2\}$. Similarly, $\lim_{n\to\infty} \frac{1}{n}\|\hat{\boldsymbol{v}}^{t+1}\|^2 = \lim_{T,n\to\infty} \frac{1}{n}\|\tilde{\boldsymbol{v}}^{T+t+1}\|^2 = \mathbb{E}\{\mathsf{v}_{t+1}(G_{t+1})^2\}$. Using these results in (C.120) and (C.121) completes the inductive proof of (C.94)-(C.95). $\qquad\square$

## C.6 Proof of Theorem 2

The proof is along the same lines as that for the square case in Section B.5; to avoid repetition, we only sketch the main steps. The first step is to show using Lemma C.9 that the state evolution result holds for the the modified AMP. That is, the following almost sure limits hold for $t \geq 1$:

$$
\lim_{m\to\infty} \frac{1}{m}\sum_{i=1}^{m} \psi(u_i^*, \hat{u}_i^1, \ldots, \hat{u}_i^{t+1}, \hat{f}_i^1, \ldots \hat{f}_i^t) = \mathbb{E}\left\{\psi(U_*, U_1, \ldots, U_{t+1}, F_1, \ldots, F_t)\right\}, \tag{C.137}
$$

$$
\lim_{n\to\infty} \frac{1}{n}\sum_{i=1}^{n} \varphi(\hat{v}_i^*, \hat{v}_i^1, \ldots, \hat{v}_i^t, \hat{g}_i^1, \ldots g_i^t) = \mathbb{E}\left\{\varphi(V_*, V_1, \ldots, V_t, G_1, \ldots, G_t)\right\}. \tag{C.138}
$$

For each of (C.137) and (C.138), we use a three-term decomposition as in (B.132). Using arguments similar to those used to analyze (B.132), we can show that each of the terms goes to 0 as $T, n \to \infty$.

The second part of the proof is to inductively show that the following statements hold almost surely for $t \geq 1$:

$$
\lim_{m\to\infty} \left| \frac{1}{m}\sum_{i=1}^{m} \psi(u_i^*, u_i^1, \ldots, u_i^{t+1}, f_i^1, \ldots, f_i^t) - \frac{1}{m}\sum_{i=1}^{m} \psi(u_i^*, \hat{u}_i^1, \ldots, \hat{u}_i^{t+1}, \hat{f}_i^1, \ldots, \hat{f}_i^t) \right| = 0,
$$
$$
\tag{C.139}
$$

$$
\lim_{m\to\infty} \frac{\|\boldsymbol{f}^t - \hat{\boldsymbol{f}}^t\|^2}{n} = 0, \quad \lim_{m\to\infty} \frac{\|\boldsymbol{u}^{t+1} - \hat{\boldsymbol{u}}^{t+1}\|^2}{m} = 0, \tag{C.140}
$$

$$
\lim_{n\to\infty} \left| \frac{1}{n}\sum_{i=1}^{n} \varphi(v_i^*, v_i^1, \ldots, v_i^t, g_i^1, \ldots, g_i^t) - \frac{1}{n}\sum_{i=1}^{n} \varphi(v_i^*, \hat{v}_i^1, \ldots, \hat{v}_i^t, \hat{g}_i^1, \ldots, \hat{g}_i^t) \right| = 0, \tag{C.141}
$$

$$
\lim_{n\to\infty} \frac{\|\boldsymbol{g}^t - \hat{\boldsymbol{g}}^t\|^2}{n} = 0, \quad \lim_{n\to\infty} \frac{\|\boldsymbol{v}^t - \hat{\boldsymbol{v}}^t\|^2}{n} = 0. \tag{C.142}
$$

Since $\psi \in \text{PL}(2)$, by the same arguments as in (B.137), we have

$$\left| \frac{1}{m} \sum_{i=1}^{m} \psi(u_i^*, u_i^1, \ldots, u_i^{t+1}, f_i^1, \ldots f_i^t) - \frac{1}{m} \sum_{i=1}^{m} \psi(u_i^*, \hat{u}_i^1, \ldots, \hat{u}_i^{t+1}, \hat{f}_i^1, \ldots \hat{f}_i^t) \right|$$

$$\leq 2C(t+2) \left[ 1 + \frac{\|\boldsymbol{u}^*\|^2}{m} + \sum_{\ell=1}^{t+1} \left( \frac{\|\boldsymbol{u}^\ell\|^2}{m} + \frac{\|\hat{\boldsymbol{u}}^\ell\|^2}{m} \right) + \sum_{\ell=1}^{t} \left( \frac{\|\boldsymbol{f}^\ell\|^2}{m} + \frac{\|\hat{\boldsymbol{f}}^\ell\|^2}{m} \right) \right]^{\frac{1}{2}}$$

$$\cdot \left( \frac{\|\hat{\boldsymbol{u}}^1 - \boldsymbol{u}^1\|^2}{m} + \ldots + \frac{\|\boldsymbol{u}^{t+1} - \hat{\boldsymbol{u}}^{t+1}\|^2}{m} + \frac{\|\boldsymbol{f}^1 - \hat{\boldsymbol{f}}^1\|^2}{m} + \ldots + \frac{\|\boldsymbol{f}^t - \hat{\boldsymbol{f}}^t\|^2}{m} \right)^{\frac{1}{2}}. \quad \text{(C.143)}$$

Using $\varphi \in \text{PL}(2)$, an analogous bound holds for the term in (C.141).

We then argue that $\lim_{n\to\infty} \frac{1}{m} \|\boldsymbol{f}^t - \hat{\boldsymbol{f}}^t\|^2 = 0$; this follows from a bound similar to (B.138) and the induction hypothesis. (In the argument, $\hat{\boldsymbol{u}}^t, \boldsymbol{u}^t, \{\mathsf{b}_{t,\ell}, \bar{\mathsf{b}}_{t,\ell}\}_{\ell \in [1,t]}$ in (B.138) are replaced by $\hat{\boldsymbol{v}}^t, \boldsymbol{v}^t$, $\{\mathsf{a}_{t,\ell}, \bar{\mathsf{a}}_{t,\ell}\}_{\ell \in [1,t]}$, respectively.) Then, recalling $\hat{\boldsymbol{u}}^{t+1} = \mathsf{u}_{t+1}(\hat{\boldsymbol{f}}^t)$ and $\boldsymbol{u}^{t+1} = \mathsf{u}_{t+1}(\boldsymbol{f}^t)$, since $\mathsf{u}_{t+1}$ Lipschitz, it follows that $\lim_{n\to\infty} \frac{1}{m} \|\boldsymbol{u}^{t+1} - \hat{\boldsymbol{u}}^{t+1}\|^2 = 0$. Using the triangle inequality sandwiching argument in (B.104), the terms $\frac{1}{m}\|\boldsymbol{f}^t\|^2$, $\frac{1}{m}\|\hat{\boldsymbol{f}}^t\|^2$, $\frac{1}{m}\|\boldsymbol{u}^t\|^2$, and $\frac{1}{m}\|\hat{\boldsymbol{u}}^t\|^2$ converge to deterministic limits (analogous to (B.140)). This leads to (C.139) via (C.143). The results (C.141)-(C.142) are obtained using a similar sequence of steps.

Combining (C.139) with (C.137) and (C.141) with (C.138) yields the result of Theorem 2. $\qquad \square$

# D   An auxiliary lemma

The following result is proved in [7, Lemma 6].

**Lemma D.1.** *Let $F \colon \mathbb{R} \to \mathbb{R}$ be a Lipschitz function, with derivative $F'$ that is continuous almost everywhere in the first argument. Let $U_m$ be a sequence of random variables in $\mathbb{R}$ converging in distribution to the random variable $U$ as $m \to \infty$. Furthermore, assume that the distribution of $U$ is absolutely continuous with respect to the Lebesgue measure. Then,*

$$\lim_{m\to\infty} \mathbb{E}\{F'(U_m)\} = \mathbb{E}\{F'(U)\}.$$