# OpenReview forum: "PCA Initialization for Approximate Message Passing in Rotationally Invariant Models"
_NeurIPS.cc/2021/Conference — NeurIPS 2021 Poster_

### Official Review · Reviewer_MuPE · 2021-07-01

**Rating:** 10
**Confidence:** 4

**Summary:**

This paper considers the rank-1 signal recovery for rotationally invariant noise matrices. It uses PCA to generate an initialization of AMP that is correlated with the signal but independent of the noise. Different from the IID Gaussian noise matrix case, rotationally invariant noise matrix makes the construction and the state evolution of AMP be more difficult. For example, the Onsager term in AMP not only involves the memory in last iteration, but also contains all the memories in preceding iterations. Therefore, the PCA initialization will change the expression of AMP (and its SE) in all the iterations. In a word, this work focuses on a hard rank-1 signal recovery problem with general rotationally invariant noise matrices. The authors propose a modified AMP to solve this problem. Furthermore, the state evolution of the proposed AMP is also rigorously proved.

**Limitations And Societal Impact:**

yes

**Main Review:**

The following are the concerns of the reviewer.

1.	The symmetric square matrix case can be treated as an instance of the rectangular matrix (let m=n). Right? If so, it is unnecessary to discuss the symmetric case. The authors can directly focus on the rectangular case and then briefly introduce symmetric case as a special instance.

2.	The authors claim that the existing AMP requires an initialization that is correlated with the signal but independent of the noise, which is not practical. Thus, they consider the PCA initialization to solve this issue. However, the reader may be curious that how the existing works solve this issue without PCA. Do you mean that the existing AMP does not work? If yes, how they do the simulation?

3.	The reviewer guesses that the AMP structure in (3.1) and (3.2) is not firstly proposed by this paper. The related references should be cited when they introduce them.

4.	The main concerns are the two assumptions of Theorem 1. When the assumption “alpha > alpha_S” holds? Is it hard to satisfy this assumption? Are there any examples to show that this assumption holds?

5.	Some numerical results are provided to support the second (non-negative) assumption of Theorem 2. This assumption rigorously holds for IID Gaussian matrix A. But it is still unclear if this assumption holds for the general rotationally invariant matrix, which is the scenario that this paper focuses. This may limit the contribution of this work.

6.	In the simulation results, the proposed AMP always outperforms the existing PCA for IID Gaussian matrix A (see Fig. 1(a) and Fig. 2(a)). However, for other general matrices focused on this paper, the proposed AMP may be worse than the existing PCA in some cases (see Fig. 1(b), Fig. 1(c), Fig. 2(b) and Fig. 2(c)). Sometimes, the proposed AMP is always worse than the existing PCA (see Fig. 1(d) and Fig. 2(d)). This also limits the contribution of this work.

7.	The reviewer is looking forward to seeing that the authors can construct a new VAMP/OAMP or memory AMP that always outperforms the existing PCA.

8.  Symbol u_t is confusing. It is used for estimate function and also for function output.

**Time Spent Reviewing:**

20

---

> ### Author Response · Authors · 2021-08-09
> **Response to Reviewer MuPE**
>
> Thanks very much for the detailed comments. We reply below to the concerns mentioned in the review, and will edit the revision accordingly.
>
> --- **(1)** ---
>
> Setting $m=n$ in the rectangular case would give a square matrix, but it would not be necessarily symmetric. Thus, the symmetric square case cannot be treated as an instance of the rectangular case. Actually, it is possible to do a reduction in the opposite direction in the Gaussian noise setting, i.e., deduce the AMP and state evolution for the rectangular case from the square symmetric case of dimension $(m+n) \times (m+n)$, see [25]. A similar reduction may be possible in the rotationally invariant setting, but we avoid this route given the complexity of the state evolution equations.
>
> --- **(2)** ---
>
> The existing AMP simulations (such as the one in [20]) assume that a correlated initialization independent of the noise is available. We note that when the signal has non-zero mean, the all-ones vector provides such an initialization. However, when the signal has zero mean, prior works required an oracle initialization, which is not feasible in practice.
>
> --- **(3)** ---
>
> The reviewer is correct. We will cite [20] and the related works [14, 47] that first proposed an AMP with the structure in (3.1) and (3.2).
>
> --- **(4)** ---
>
> Let us recall that the parameter $\alpha_{\rm s}$ denotes the spectral threshold: for $\alpha>\alpha_{\rm s}$, the PCA estimator has non-zero asymptotic correlation with the signal (the expression of the limit correlation is given in (2.2)); for $\alpha < \alpha_{\rm s}$, the correlation of the PCA estimator with the signal vanishes. Thus, the assumption $\alpha>\alpha_{\rm s}$ simply means that our initialization method is effective. The value of $\alpha_{\rm s}$ depends on the distribution of the noise, and it is equal to the inverse of the G-transform computed at the right edge of the spectrum, see lines 118-119. More concretely, for the Marcenko-Pastur noise with aspect ratio $c=2$ considered in Figs. 1(a) and 2(a), we have that $\alpha_{\rm s}=1+\sqrt{c}=1+\sqrt{2}$. Furthermore, for the uniform noise considered in Figs. 1(b)-(d) and 2(b)-(d), we have that $\alpha_{\rm s}=0$, which means that for any $\alpha>0$, the PCA estimator will be effective.
>
> --- **(5)** ---
>
> The assumption in Theorems 1 and 2 that the free cumulants are non-negative appears to be an artifact of the proof technique. We expect that it should be possible to relax it, and give experimental evidence that our results hold even in settings in which some of the free cumulants are negative (uniformly distributed noise spectrum, see Fig. 1(b) for the square symmetric case and Fig. 1(c)-1(d) for the rectangular case). We regard the generalization of our approach to any compactly supported noise distribution (without assumptions on the sign of the free cumulants) as an interesting open problem.
>
> --- **(6)** ---
>
> As correctly pointed out by the reviewer, in some cases, the proposed AMP with PCA initialization is outperformed by PCA alone. This is due to the suboptimal choice of the denoisers. More specifically, we have chosen denoisers that are Bayes-optimal for Gaussian noise, but these are likely far from optimal in other rotationally invariant settings. As discussed in lines 199-217, determining the optimal denoiser for each iteration is a challenging problem (due to the complicated nature of the state evolution equations) and it is left for future work. We also note that the performance of PCA can always be matched by AMP by simply using a linear denoiser in each iteration.
>
> --- **(7)** ---
>
> We agree with the reviewer that this is an exciting direction for future work.
>
> --- **(8)** ---
>
> Thank you for this comment, we will change the notation to make it clearer.

---

> > ### Comment · Reviewer_MuPE · 2021-08-19
> > **The reviewer suggests to accept this paper**
> >
> > The authors have addressed all my concerns. From my point of view, it is a nice paper. The results are interesting to the researchers in the AMP field.

---

### Official Review · Reviewer_NNbz · 2021-07-13

**Rating:** 6
**Confidence:** 3

**Summary:**

The authors consider the problem of estimating a structured rank-1 signal in the presence of rotationally invariant noise, which is a class of perturbations that is more general than Gaussian noise. For this problem, they propose an AMP algorithm that is initialized by PCA, and they derivate a state-evolution characterization that holds in the infinite-dimensional limit. The rationale is that PCA is a near-optimal estimator in the absence of structure, whereas AMP facilitates the exploitation of structure yet allows a rigorous asymptotic analysis. Their work builds upon previous work, such as PCA-initialized AMP with Gaussian noise, but is different in that, to handle rotationally invariant noise, their algorithm uses a PCA correction at every iteration, not just the first one. Their state-evolution analysis uses a two-phase artificial AMP formulation that has previously been used in the context of generalized linear models.  Although their analysis assumes a noise distribution with non-negative free cumulants, they provide empirical simulation results that suggest that derived state evolution is correct with any noise distribution with compact support.

**Ethical Concerns:**

The reviewer does not perceive any ethical concerns.

**Limitations And Societal Impact:**

The reviewer does not see any possibility of negative societal impact.

**Main Review:**

The contribution is a significant extension of existing results in that it extends PCA-initialized rank-1 AMP from the case of Gaussian noise to more generic noise distributions.  The results are intuitive yet rigorous and mathematically sophisticated. Furthermore, the analysis agrees with simulations conducted outside the stated assumptions (on non-negative free cumulants). The writing is very clear and well organized.

As pointed out by other reviewers, the lack of experimental results on real data constitutes a major weakness of this paper. Thus, I believe that it would be difficult for this paper to attract a wide audience.

Overall, this seems like a good contribution to NeurIPS, but not a great one.

**Time Spent Reviewing:**

3 hours

---

> ### Author Response · Authors · 2021-08-09
> **Response to Reviewer NNbz**
>
> Thank you for the encouraging comments and positive evaluation of our work.

---

> > ### Comment · Reviewer_NNbz · 2021-08-23
> > **Reply to authors' response**
> >
> > After reading all the other reviews, I agree with Reviewers aknp and iyWx in that the lack of experimental results on real data constitutes a major weakness of this paper. Thus, I believe that it would be difficult for this paper to attract a wide audience. For this reason, I am downgrading my score to 6.

---

### Official Review · Reviewer_iyWx · 2021-07-15

**Rating:** 8
**Confidence:** 5

**Summary:**

The submitted paper addresses estimation of rank-1 matrices from observations (1.1) or (1.2) subject to rotationally invariant noise matrix W. The paper proposes approximate message passing (AMP) with long memory Onsager correction (3.2) in the square case or (3.10) in the rectangular case, inspired by an arXiv preprint [20]. AMP uses the conventional principal component analysis (PCA) estimate as the initial message (3.1) or (3.9). To eliminate dependencies between the initial message and the noise matrix W, non-trivial Onsager correction b_{t,1} in (3.3) or a_{t,1}, b_{t+1,1} given in (3.11) and (3.13) is proposed via state evolution while the other correction b_{t,t-j} in (3.3) or (3.12) and (3.14) is the same as in [20]. Theorems 1 and 2 claim state evolution recursion for the square and rectangular cases, respectively. The theoretical predictions are also verified via numerical simulations in artificial setting.

**Limitations And Societal Impact:**

 The limitation is that the paper only addressed the rank-1 case. From a practical point of view, the low rank case may be desired. Since real data was not treated, the paper provides no negative social impact.

**Main Review:**

Originality

 The main originality is in the proof strategy to handle dependencies between the initial PCA estimate and the noise matrix W. The strategy is different from in [43], where followed Bolthausen’s conditioning technique to analyze the conditional distribution of the data matrix X for given initial messages.

The proposed strategy is a two-phase approach: In the first phase, AMP uses a linear denoiser in (5.1) to obtain the conventional PCA estimate after a sufficiently large number of iterations. In the second phase, AMP switches the denoiser to a time-shifted nonlinear denoiser in (5.3). The proof is fully based on state evolution recursion derived in [20] under review (while I believe the correctness of the proof in [20]).

The first part in the proof is the convergence of AMP in the first phase toward the PCA estimate. The proof utilizes a standard strategy in convergence analysis, such as the existence of a fixed point (Lemma B.4) and a construction of a contractive mapping (Lemma B.5). I guess that the linear denoiser enables such a proof.

The second part is a proof of the consistency (Lemma B.8) between the actual AMP and the two-phase AMP in the second phase. This is an interesting step to justify the definition of the Onsager correction in the actual AMP.

--Why do Theorems 1 and 2 need the continuous differentiability of the denoiser? The assumption excludes practical functions, such as soft thresholding and ReLU. Conventional state evolution only postulates the Lipschitz continuity, which implies continuous differentiability “almost everywhere.”

--Why do not the authors need the non-negativity of the exponent T+1-(t-j) for 1/\alpha in (5.4)? In my understanding, this factor originates from the derivatives of the linear denoisers in the first phase for i\in\{t-j+1, …, T+1} in the case t-j+1\leq T+1. Thus, T+1-(t-j) must be non-negative. In the current definition, let j=1 to obtain T+1-(t-j)=T+2-t<0 for t>T+2.

--AMP in the first phase uses the true signal plus noise as the initial message. Why cannot AMP start with only noise, i.e. random initialization?

Quality

 As pointed out above, the proof strategy is novel, general, and may be applicable to the other message passing. The mathematical quality in the proof is high while I did not fully read the whole proof in the supplementary material. For a further improvement in terms of references, I would like to introduce a reference [R1]. The reference is a first paper (posted on arXiv 27 March 2020 and published on IEEE TIT 04 May 2021) that presented rigorous state evolution for message passing algorithms with long memory Onsager correction, as cited in both [20] and [34].

[R1] K. Takeuchi, "Bayes-Optimal Convolutional AMP," in IEEE Transactions on Information Theory, vol. 67, no. 7, pp. 4405-4428, July 2021, doi: 10.1109/TIT.2021.3077471.

Optional comment

--It would be better if the authors presented numerical simulations for not artificial but real problems.

Clarity

The paper is well written. Theorems 1 and 2 are presented concisely after defining AMP and state evolution recursion. The main idea in the proof strategy is clear. Nonetheless, a further improvement is possible.

--Clarify the significance of b_{t,1}, (3.11), and (3.13) for readers to understand the main difference between AMP in [20] and the proposed AMP. I reached my understanding mentioned in Originality after I considered several possibilities, such as differences between measurement models or the definitions of free cumulants in [20] and the submitted paper.

Significance

 The submitted paper is a first attempt (i.e. rank-1 case) for estimation of low-rank matrices from rotationally orthogonal measurements. In my understanding, the paper has solved the main challenge in that problem. A generalization to the low rank case is possible.


**Time Spent Reviewing:**

10

---

> ### Author Response · Authors · 2021-08-09
> **Response to Reviewer iyWx**
>
> Thanks very much for the detailed comments. We reply below to the points raised in the review, and will edit the revision accordingly.
>
>
> **Originality**
>
> *Comment: “Why do Theorems 1 and 2 need the continuous differentiability of the denoiser?”*
>
> **Response.** This is a great point. We actually do not need the continuous differentiability of the denoiser. Without the need to change the proof, the assumption on $u_t$ can be weakened to: (i) being differentiable almost everywhere, and (ii) satisfying a mild non-degeneracy condition (Assumption 4.2(d) in [20]). In this way, we can cover most of the choices of $u_t$ that are practically relevant (e.g., soft thresholding and ReLU). To keep the list of assumptions brief, in the submitted version of the paper we have used the stronger condition on $u_t$ (continuously differentiable and Lipschitz), but we will clarify this point in the revision.
>
> *Comment: “Why do not the authors need the non-negativity of the exponent T+1-(t-j) for 1/\alpha in (5.4)?”*
>
> **Response.** The reviewer is correct: the condition that the exponent $T+1-(t-j)$ of $1/\alpha$ in (5.4) is non-negative arises from the linear denoisers used in the first phase (up to iteration $T+1$). Since this term appears in the memory coefficient multiplying iterate $(t-j)$, it equals one unless the index $t-j$ corresponds to a first phase iterate. This notation allows us to write the memory coefficients corresponding to all the past iterates in a unified manner.
>
>
> *Comment: “Why cannot AMP start with only noise, i.e. random initialization?”*
>
> **Response.** If the signal has zero mean, then a random initialization would be asymptotically uncorrelated with the signal, leading to future AMP iterates also being uncorrelated. That is, the first phase would get stuck at a trivial fixed point and the iterates would not be guaranteed to converge to the principal eigenvector. To ensure that this does not happen, the first phase of the artificial AMP is initialized with a linear combination of the true signal and independent Gaussian noise. We note that this initialization is only used as a proof technique, and the actual algorithm uses the PCA initialization.
>
>
> **Quality**
>
> Thanks very much for pointing out the reference, we will add in the revision.
>
>
> **Optional comment**
>
> This is a very good point. As mentioned in the response to reviewer *bZze* above, generalizing the algorithm to estimate $k$ principal components and applying it to real data is an important direction for future research.
>
>
> **Clarity**
>
> Thank you for this comment, your intuition is correct. The key difference between our algorithm and the one in [20] is in the memory coefficient corresponding to the first pair of $u$ and $v$ iterates. The difference is precisely because one can interpret the PCA initialization as the result of a first AMP phase with linear denoiser. The effect of the first phase is reflected in the memory coefficient multiplying the initial iterates. We will clarify this point in the revision.

---

> > ### Comment · Reviewer_iyWx · 2021-08-23
> > **Response to authors' rebuttal**
> >
> > Thanks to the authors for their responses to my comments. I believe that their technical contributions on AMP should be presented at NeurIPS 2021. I would like to encourage the authors to tackle my optional comment in future research.

---

### Official Review · Reviewer_aknp · 2021-07-16

**Rating:** 6
**Confidence:** 2

**Summary:**

This paper considers the problem of estimating a signal from its noisy rank-1 measurement. PCA is a traditional method that can be used for this task. AMP is an alternative that can additionally leverage statistical priors on the unknown signal. AMP algorithms are traditionally analyzed using a scalar iteration known as state evolution (SE).

The focus of this paper is to theoretically establish SE for AMP under a more general noise matrix starting from an initialization using PCA. Having SE enables the analysis of the error behaviour of AMP in different settings. Empirical results focus on comparing AMP+PCA against PCA on simulated data.

Theoretical contributions: The paper develops a new AMP algorithm initialized using PCA. Theorem 1 and 2 are the main theoretical results in the paper that establish SE for the proposed algorithm for square-symmetric and rectangular matrices.

Empirical contribution: Validation of the superiority of AMP on simulated data over PCA.

**Ethical Concerns:**

No issues with this.

**Limitations And Societal Impact:**

No issues with this.

**Main Review:**

General comment: While I am familiar with AMP and SE, I am not an AMP researcher and thus have only loose familiarity with corresponding literature. My review thus should be weighted against others who have deeper expertise in AMP/SE. On the other hand, my review provides valid feedback from someone with expertise in optimization and ML.

Strengths: Generalizing the AMP and its analysis to rotationally invariant matrices can lead to new insights into AMP and its usage for statistical estimation from rank-1 measurements. The theorems show that one can establish SE for new AMP algorithms, which is also promising. Simulations highlight the benefit of AMP over PCA when leveraging structure on the signal.

Issues: (1) A persistent issue in the AMP literature is the lack of applications on real data analysis. This paper follows this trend and limits itself to validating the theory on synthetic experiments. In fact, this is a major issue that is often overlooked in theoretical papers, but which limits the significance of the results to data analysis. (2) Empirical validation limits comparisons to PCA. Why not compare to prior AMP methods for estimating from noisy rank-1 matrices (even when there is a mismatch in the noise distribution)? (3) How tolerant is the theoretical analysis to deviations from the assumptions? Which assumptions are more important than others? The assumptions in the manuscript are often presented in a scattered form within other assumptions or paragraphs of text, which reduces clarity.

Summary: For someone not deeply familiar with the theoretical work on AMP, this paper seems to be proposing a novel algorithmic extension and proving that SE is applicable to that extension. As with most prior work on AMP, the method is never used for actual data analysis, but validated on a contrived set of simulations. In short, I think this paper could be of value to AMP researchers, but its potential to broader ML and data science has not been demonstrated.

**Time Spent Reviewing:**

2 hours

---

> ### Author Response · Authors · 2021-08-09
> **Response to Reviewer aknp**
>
> Thanks very much for the comments. We respond below to the three issues raised in the review, and will edit the revision accordingly.
>
> --- **(1)** ---
>
> A recent paper by Zhong et al. (https://arxiv.org/abs/2012.11676) shows how the combination of PCA and AMP can be used for accurately estimating principal components (PCs) in applications such as high-dimensional genomics datasets. The authors use AMP to estimate the top-$k$ PCs assuming a Gaussian noise model, and they also mention that a rotationally invariant AMP will be more suited to applications where the noise has a correlation structure. Our work takes the first step towards this ambitious goal: we propose an AMP algorithm initialized with PCA to estimate the top principal component in the presence of rotationally invariant noise, and we give a rigorous characterization of its performance. This is a necessary prerequisite to tackle the problem of estimating the top-$k$ PCs, which would make AMP with PCA initialization directly usable in practical applications.
>
>
> --- **(2)** ---
>
> This is a great suggestion, thanks! In the final version, we will include in the plots the performance of the AMP with PCA initialization that assumes that the noise is Gaussian with the correct variance. In the following, we will refer to this last algorithm as “Gaussian PCA+AMP”. For symmetric noise matrices with Marcenko-Pastur spectrum (Figs. 1(a) and 2(a) of our paper), we have found that the performance of the proposed rotationally invariant PCA+AMP is consistently better than that of the Gaussian PCA+AMP. In fact, the Gaussian PCA+AMP does not improve on PCA alone up to moderate SNR values (i.e., up to $\alpha =3$), and it performs only modestly better than PCA alone at higher SNR.
>
> In general, the performance gap between the proposed rotationally invariant PCA+AMP and the existing Gaussian PCA+AMP will be significant unless the sequence of the free cumulants $\kappa_k$ decays very quickly. For the symmetric case with Marcenko-Pastur noise spectrum, the free cumulants are all equal (see line 266) and thus the performance gap is significant. If the eigenvalues of the noise matrix follow a uniform distribution, then the sequence of free cumulants decays rather fast and the performance gap is small. If the eigenvalues of the noise matrix follow a Beta distribution (as in the numerical experiments of [20]), then the sequence of free cumulants grows quickly, and the Gaussian PCA+AMP performs much worse than the proposed rotationally invariant PCA+AMP (since the latter adapts to the noise spectrum).
>
>
> --- **(3)** ---
>
> In the revision, we will provide a more detailed and unified discussion of the assumptions. We can divide those assumptions into the following three categories.
>
> *(i)* The basic assumption of the paper is that the noise matrix is rotationally invariant. This is a rather mild requirement as it allows for arbitrary distributions of the spectrum of the noise matrix.
>
> *(ii)* The assumption that the free cumulants are non-negative appears to be an artifact of the proof technique. We expect that it should be possible to relax it, and give experimental evidence that our results hold even in settings in which this assumption is not satisfied (uniform noise, see Figs. 1(b)-1(d) and Figs. 2(b)-2(d)).
>
> *(iii)* The paper also makes a few technical assumptions: convergence of the empirical distributions of the signal and of the eigenvalues of the noise matrix; boundedness of the $(2+\varepsilon)$-th moment of the signal; compact support of the spectrum of the noise matrix.  We regard these assumptions as minor, and we remark that they are quite standard in the literature. For the rectangular case, we also need the additional assumption that the law of the signal is zero mean and satisfies a log-Sobolev inequality, which is necessary to apply the framework in [11].

---

> > ### Comment · Reviewer_aknp · 2021-08-28
> > **Response to the Rebuttal**
> >
> > Thank you for carefully reading my feedback. My only remaining concern is the validation on real data, which, as I said above, is a broader problem in the AMP literature. However, based on the enthusiasm of other reviewers and the potential theoretical contribution to AMP, I am increasing my rating by a point.

---

### Official Review · Reviewer_bZzE · 2021-07-17

**Rating:** 6
**Confidence:** 3

**Summary:**

The paper studies the asymptotics of the AMP algorithms with PCA initialization.

**Limitations And Societal Impact:**

This is a theoretical paper. I think the main limitation is what are the possible applications to the theorems.

**Main Review:**

I am very sorry that I do not have enough time to spend on this paper especially to go through the long appendix. But I do have concerns about the significance of the contribution.
1) It is unclear to me why the paper only studies the rank-1 PCA rather than rank-k PCA. It is great to see that the symmetric condition can be relaxed and the results hold for the asymmetric matrix but the rank-1 condition is a very strong condition.
2) The $u_t$ function in (3.2) is assumed to be continuously differentiable and Lipschitz. This is a standard assumption in AMP literature. However, for some applications of this work like phase retrieval, the $u_t$ function may not be differentiable everywhere depending on the loss function. It would be great to extend the assumption on $u_t$ to tolerate some non-differentiable points.
3) It would be great to have a section to discuss which application requires a PCA initialization with an AMP algorithm. The only application I know of is phase retrieval. In general, only the non-convex problems require special initialization and hence PCA initialization might be required. Also, we need the AMP algorithm to perform well on the non-convex problem with PCA initialization. These conditions quite limit the potential applications.

In summary, it seems to be a borderline paper or slightly below average to me.

**Time Spent Reviewing:**

4

---

> ### Author Response · Authors · 2021-08-09
> **Response to Reviewer bZzE**
>
> Thanks very much for the comments. We address them below and will incorporate this discussion into the revision.
>
> --- **(1)** ---
>
> To extend the proposed PCA+AMP algorithm to estimate rank-$k$ signals, the memory coefficients in the AMP would need to be replaced by $k \times k$ matrices. This program has been carried out in the Gaussian noise setting [43], but determining the exact structure of these memory matrices in the rotationally invariant setting is an interesting open question. To address this, the key step is to generalize the standard AMP and its analysis in [20] to cover $n \times k$ matrix-valued iterates (rather than $n \times 1$ vector-valued iterates). We focus here on rigorously analyzing PCA+AMP for the rank-1 case, since it is a necessary (and already quite challenging) first step to address the rank-$k$ case.
>
> Let us also remark that, if there is sufficient separation between the SNRs (i.e., between the eigenvalues/singular values) of the $k$ spikes, one can sequentially estimate the spikes by applying the rank-1 algorithm $k$ times and subtracting the contribution of the previous estimates each time.
>
>
> --- **(2)** ---
>
> This is a great point. We actually do not need the continuous differentiability of the denoiser. Without the need to change the proof, the assumption on $u_t$ can be weakened to: (i) being differentiable almost everywhere, and (ii) satisfying a mild non-degeneracy condition (Assumption 4.2(d) in [20]). In this way, we can cover most of the choices of $u_t$ that are practically relevant (e.g., soft thresholding and ReLU). To keep the list of assumptions brief, in the submitted version of the paper we used the stronger condition on $u_t$ (continuously differentiable and Lipschitz), but we will clarify this point in the revision.
>
>
> --- **(3)** ---
>
> Thanks for this suggestion. PCA initialization is required for matrix estimation via AMP whenever the signal (i.e., spike) has zero mean. Community detection in a balanced stochastic block model is an example of such a rank-1 estimation problem. (Here, AMP is actually Bayes-optimal as shown in [16, 43].) Prior work on rotationally invariant AMP (such as [20]) assumes that an initialization correlated with the signal and independent of the data matrix is available, but this is impractical when the signal has zero mean. In the revision, we will clarify exactly when AMP requires PCA initialization.
>
> Let us also mention that phase retrieval does not fall under the low-rank matrix estimation model we study. Phase retrieval can indeed be viewed as a rank-1 matrix recovery problem from a measurement vector obtained via a linear transformation of the matrix.  (Algorithms such as PhaseLift use this approach.) In our setting, the observed data is just the rank-one signal plus a noise matrix -- there is no measurement operator/linear transformation. The AMP and the spectral estimator (PCA) are therefore substantially different from those for phase retrieval, even in the Gaussian setting. Obtaining an AMP for phase retrieval with rotationally invariant design matrices is an exciting direction for future research. We also note that Vector GAMP [50] is an alternative solution based on expectation propagation.

---

> > ### Comment · Reviewer_bZzE · 2021-09-04
> > **Upgrade by one point**
> >
> > Thanks for the response. I still have concerns about what are and how many applications can be benefited by this framework. Yet, based on the other reviewers and the potential theoretical contribution to AMP, I am increasing my rating to 6.

---

### Decision · Program_Chairs · 2021-09-27

**Decision:**

Accept (Poster)

**Comment:**

This paper received 5 reviews. The scores/confidences were 6/3, 6/2, 8/5, 6/3, and 10/4, implying that all the reviewers evaluated this paper positively. We can notice that there is a relatively large spread among these review scores, but I think that the review contents are more or less coherent across all the reviews: The major strength of this paper is certainly its high technical quality, as commented by most reviewers. On the other hand, the major weakness is that it does not succeed in demonstrating significance of the theoretical contribution in real applications, so that it would be difficult for this paper to attract a wide audience beyond those who have specific interest in AMP and related iterative inference methods. The five reviews weighed these points differently on the basis of their own expertise, resulting in the spread of the scores. I am thus happy to recommend acceptance of this paper for presentation at the NeurIPS conference. At the same time, I would like to encourage the authors to consider demonstrating the significance of their contribution in real-world settings.

Very minor points:
- Lines 107-109: One would have to assume that $O$ is independent of $\Lambda$.
- Lines 127-129: Similarly, one would have to assume that $O$ and $Q$ are independent and independent of $\Lambda$.